



**Changing characteristics of atmospheric CH₄ on the Tibetan Plateau,**
**records from 1994 to 2017 at Mount Waliguan station**
**Shuo Liu[1,2,3], Shuangxi Fang[3], Peng Liu[4], Miao Liang[5], Minrui Guo[6], Zhaozhong Feng[1,7]**
**[1]State Key Laboratory of Urban and Regional Ecology, Research Center for Eco-Environmental Sciences,**
**Chinese Academy of Sciences, Beijing, China**
**[2]College of Resources and Environment, University of Chinese Academy of Sciences, Beijing, China**
**[3]College of Environment, Zhejiang University of Technology, Hangzhou, China**
**[4]Mt. Waliguan background station, China Meteorological Administration (CMA), Qinghai, China**
**[5]Meteorological Observation Center (MOC), China Meteorological Administration (CMA), Beijing, China**
**[6]College of Global Change and Earth System Science, Beijing Normal University, Beijing, China**
**[7]Institute of Ecology, Key Laboratory of Agrometeorology of Jiangsu Province, School of Applied**
**Meteorology, Nanjing University of Information Science & Technology, Nanjing, China**
**Correspondence**:
Shuangxi Fang (fangsx@cma.gov.cn); Zhaozhong Feng (zhzhfeng201@hotmail.com)



**Abstract.** A 24-year long-term observation of atmospheric $CH_4$ was presented at
Mt.Waliguan (WLG) station, the only WMO/GAW global station in inland of Eurasia.
Overall, during 1994-2017, continuously increase of atmospheric $CH_4$ was observed at
WLG with yearly growth rate of $5.1 \pm 0.1$ ppb $yr^{-1}$, although near-zero and even
negative growth appeared in some particular periods, e.g., 1999-2000, and 2004-2006.
The average $CH_4$ mole fraction was only $1805.8 \pm 0.1$ ppb in 1995, but unprecedented
elevated ~100 ppb and reached a historic high of $1903.8 \pm 0.1$ ppb in 2016. The seasonal
averages of atmospheric $CH_4$ at WLG were ordered by summer, winter, autumn and
spring, and the correlation slopes of $\Delta CO/\Delta CH_4$ showed a maximum in summer and
minimum in winter, which was almost opposite to other sites in the northern
hemisphere, e.g., Mauna Loa, Jungfraujoch, and was caused by regional transport.
Strong potential sources at WLG were predominately identified in northeast (cities, e.g.,
Xining, Lanzhou) and southwest (the Northern India), and air masses from west and
northwest regions were accompanied with higher $CH_4$ mole fractions than that from
city regions.
What is interesting is that obviously changes appeared in different observing periods.
Generally, i) the amplitudes of diurnal or seasonal cycles were continuously increasing
over time, ii) the wind sectors with elevated $CH_4$ moved from ENE-…-SSE sectors in
early periods to NNE-…-E sectors (city regions) in later years, iii) the area of source
regions was increasing along with the years, and strong sources gradually shifted from
northeast to southwest, iv) the annual growth rates in recent years (e.g., 2013-2016)
were significantly larger than that in early periods (e.g., 1998-2012). We conclude that
the site was more and more affected by regional sources along with the time. Northern
India was possibly becoming the strongest source area to WLG rather than city regions
before. The case study in the Tibetan Plateau showed that the atmospheric $CH_4$
observed in Qinghai-Tibetan Plateau changed not as expected, the annual growth rate
was even larger than that in city regions in some period (e.g., $7.3 \pm 0.1$ ppb $yr^{-1}$ in 2013-





2016). It is unambiguous that the anomalously fluctuations of atmospheric $CH_4$ in this
region are a warning to the world, its increasingly annual growth rate may be a
dangerous signal to global climate change.



## 1 Introduction

Since the pre-industrial era, the emissions of greenhouse gases (GHGs) have increased continuously, and larger absolute increases were found in recent years with the concentration higher than ever now (WMO, 2019). The GHGs could perturb the infrared radiation balance, trap the heat in the atmosphere, which contributes to global warming, melting glaciers, extreme weather events and many other global climate changes (IPCC, 2014). The recent 30-years from 1983 to 2012 were the warmest of the last 800-years in the Northern Hemisphere, and half of the rising surface temperature was due to increased GHGs emissions (IPCC, 2014). As one of the most important GHGs, the global warming effect of methane ($CH_4$) is just after carbon dioxide ($CO_2$) (Etminan et al., 2016). It has an 8-12 years atmospheric lifetime (Battle et al., 1996), with the global warming potential of ~23 times greater than $CO_2$ over a 100 year horizon (Weber et al., 2019). About 17% of radiative forcing by long-lived greenhouse gases was contributed by $CH_4$ during 1750-2016 (Etminan et al., 2016). Since the beginning of the industrial era, the concentration of $CH_4$ is rapidly increased because of the influence of anthropogenic activities (Saunois et al., 2016). The result by the analyses of ice cores in Antarctica showed that the atmospheric concentration of $CH_4$ has reached unprecedented over the last 0.8 million years (IPCC, 2014).

In the beginning of 1990s, $CH_4$ concentration appeared a decreasing trend in global scale. However, high growth rates were found in 1998, which was possibly due to the higher global mean temperature (Dlugokencky et al., 1998; Nisbet et al., 2014). Subsequently, a low growth rate sustained over 1999 to 2006, except for special years (2002/2003) with El Niño events (Dlugokencky et al., 1998). The annual growth rates dropped from ~12 ppb $yr^{-1}$ to near 0 from late 1980s to 1999-2006 (Nisbet et al., 2019). But thereafter, the atmospheric $CH_4$ concentration keeps rising from 2007. During 2007-2013, the annual growth rate of methane was $5.7 \pm 1.2$ ppb $yr^{-1}$. After 2013, the atmospheric $CH_4$ grew even at rates not observed after 1980s, such as $12.7 \pm 0.5$ ppb $yr^{-1}$ in 2014 and $10.1 \pm 0.7$ ppb $yr^{-1}$ in 2015 (Nisbet et al., 2016, 2019). The overall



global growth rate was 7.1 ppb yr$^{-1}$ in recent 10 years (WMO, 2019). And the not
expected increase since 2007 would make it difficult to meet the targets of carbon
emission reduction in the future. The World Meteorological Organization/Global
Atmospheric Watch programme (WMO/GAW) annual greenhouse gas bulletin revealed
that globally averaged $CH_4$ mole fraction reached a new high with $1869 \pm 2$ ppb in 2018
(Rubino et al., 2019), which was ~259% of pre-industrial levels (~722 ppb around 1750
C.E.) (Etheridge et al., 1998; WMO, 2019).

The atmospheric $CH_4$ is mainly emitted from natural sources (about 40%, e.g.,

ruminants and wetlands) and anthropogenic sources (about 60%, e.g., paddies, cattle
ranch, coal mine, fossil fuel and biomass burning) (Hausmann et al., 2016; Saunois et
al., 2016). Studies from GAW observations indicated that the causes of recent increase
were likely attributed to anthropogenic emissions at mid-latitudes in the northern
hemisphere and the wetlands in the tropics (WMO, 2019). The rapid development of
population growth, economic expansion and countries urbanization has led to more and
more fossil fuel production and consumption (e.g., the large-scale exploitation of
natural gas, oil and coal) and biomass burning, consequently large amounts of
anthropogenic $CH_4$ were emitted around the world in recent years (Galloway, 1989;
Streets and Waldhoff, 2000; Wang et al., 2002; Lin et al., 2014; Hausmann et al., 2016).
The recent carbon isotope study revealed that biogenic emissions might also have
driven $CH_4$ increase, including microbial sources whether from rice, paddies,
ruminants, termites, enteric fermentation or all of these (Nisbet et al., 2016; Schaefer et
al., 2016; Wolf et al., 2017). 90% of $CH_4$ destruction in the atmosphere are mainly from
the reaction with hydroxyl radicals (OH) (Vaghjiani and Ravishankara, 1991; Bousquet
et al., 2011), an important oxidant in the troposphere (Logan et al., 1981). Therefore,
the interannual variability of OH or the decline oxidative capacity of the atmosphere
may also cause the recently increased $CH_4$ growth rates (Rigby et al., 2017; Turner et
al., 2017).

To get accurate understanding of atmospheric $CH_4$, a systematic observations



network would perform the best. Hundreds of $CH_4$ observation stations worldwide are
running under the framework of WMO/GAW. Since 1978, systematic measurements of
atmospheric $CH_4$ began around the world (Blake et al., 1982; Rasmussen and Khalil,
1984; Dlugokencky et al., 1994). On the northern slope of the Mauna Loa volcano,
Hawaii, there exists the first global station Mauna Loa (MLO), which was performed
about 3397m above sea level (a.s.l.) and far away from local sources and sinks. It has
the longest records of continuous atmospheric $CH_4$ observation (Keeling et al., 1976).
Later, many types of the sites were installed $CH_4$ observation system, such as the
Barrow (BRW), South Pole (polar site, SPO) (Dlugokencky et al., 1995), Cape Grim
(CGO) in Australia (coastal/island sites) (Pearman and Beardsmore, 1984),
Minamitorishima (MNM) in Japan (coastal/island sites) (Wada et al., 2007),
Jungfraujoch (JFJ) in Switzerland and Mount Waliguan in China (continental mountain
site) (Zhou et al., 2004; Loov et al., 2008). Even though, the exact causes of
significantly increased $CH_4$ emissions in past years are still remained unclear and
debated, especially for the anomalous periods with suddenly large growth, due to the
time and space sparsity of measurements and the crude model approaches, which
limited our understanding of the global variation of atmospheric $CH_4$ (Saunois et al.,
2019; Weber et al., 2019). As long as the reasons for rising $CH_4$ emissions contributed
by natural sources (e.g. wetlands), anthropogenic sources (e.g. fossil fuels), or climate
change feedbacks remain uncertainties, it will be impractical to predict $CH_4$ trends in
the future, and then to develop realistic management (Nisbet et al., 2019). Therefore, it
is essential to establish typical observing regions and perform long observations.

China has the largest anthropogenic $CH_4$ emissions in the world (Janssens-

Maenhout et al., 2019). Qinghai-Tibetan Plateau has an average altitude over 4000m
a.s.l., which has long been recognized as the roof of the world. By coincidence, the two
largest $CH_4$ source regions in the world (i.e. Eastern China and Northern India) trapped
the Tibetan Plateau in the middle (Zhang et al., 2011; Fu et al., 2012; Wilson and Smith,
2015). Under the characteristics of special geographical conditions, lower population



density, rarely industrial activities and high sensitivity to external disturbances, the
Tibetan Plateau is undoubtedly one of ideal regions to observe continual $CH_4$ signal
(Zhou et al., 2005; Fu et al., 2012; Zhang et al., 2013). Most of the previous studies
reported the short-term $CH_4$ variations in China and concluded the importance of long-
term observation (Cai et al., 2000; Zou et al., 2005; Wang et al., 2009; Fang et al., 2013),
which is of great value to enhance the understanding of the global carbon cycle (Yuan
et al., 2019). As the rapid development of China and India, the year to year difference
as well as the sources and sinks of $CH_4$ on the Tibetan Plateau might change
significantly over time. Since 1994, in-situ measurements of atmospheric $CH_4$ have
been launched at Mt.Waliguan (WLG) station. To study the long-term variations of
atmospheric $CH_4$ at the WLG and get a new insight of its characteristics in the inland
of the Eurasia, in this study, we evaluated the performance of a 24 year long-term in
situ observations of $CH_4$ at Mt.Waliguan baseline observatory, which is the longest time
observing records in China. Temporal patterns, annual variations, long-term trends, air
mass transports, spatial distribution of potential sources were analyzed. In addition, the
case studies combining atmospheric CO measurements and a separate analysis between
the Tibetan Plateau and the city regions were performed to constrain the contribution
of anthropogenic emissions.

**2 Methodology**
**2.1 Measurement site**
The Mt.Waliguan (WLG, 36.28° N, 100.09° E, 3816m a.s.l.) station is situated at the
edge of northeastern of the Tibetan (Qinghai-Xizang) Plateau, which was in remote
western China and isolated from populated and industrial regions (Fig. 1). WLG was
the only WMO/GAW global background station in Eurasia and running by the China
Meteorological Administration (CMA). The surrounding areas of the site are pristine
with sparse vegetation, naturally arid and semi-arid grasslands. Small farms with yak



and sheep are in the valley. Two adjacent large cities Xining (~2.2 million populations)
and Lanzhou are located about 90km northeast and 260km east of the station,
respectively. The Longyangxia hydroelectric station (~380 km$^2$) is located
approximately 13km south to southwest of the WLG. The predominant winds at WLG
are mainly from southwest and east in winter and summer, respectively (Zhou et al.,
2004; Zhang et al., 2011), which is controlled by Tibet Plateau monsoon.
Simultaneously, dial variations of vertical winds at WLG is influenced by mountain-
valley breezes, where upslope flow brings heated air masses from the boundary layer
to the site in daytime and downslope flow results in cool air masses transport from
mountain peak to the site. Under this unique location, the observation at WLG could
obtain essential information on CH$_4$ sources and sinks from Eurasia (Zhou et al., 2005;
Zhang et al., 2013).

**2.2  Instrumental setup**
Atmospheric CH$_4$ has been measured quasi-continuously using a HP 5890 gas
chromatograph (GC) equipped with a flame ionization detector (FID) since July 1994,
and an Agilent 6890N GC equipped with a FID since June 2008. Both of the systems
used the same sampling procedures. A Cavity Ring Down Spectroscopy system (Picarro
G1301) began in January 2009 and the instrument was upgraded to Picarro G2401 in
2015. Ambient air is delivered to the above systems at about 5 L/min by a KNF
Neuberger N2202 vacuum pump via a dedicated 0.95 cm o.d. sample line from an 80m
intake line attached to an 89m steel triangular tower located approximately 15m from
the main observatory. The residence time of the ambient air from the top of the tower
to the instrument is 30 s. The ambient air is first passed through a 7 mm stainless steel
membrane filter located upstream of the pump and then (after the pump) passed through
a pressure relief valve set at 1 atm to release excess air and pressure. The ambient air is
then dried to a dew point of approximately -60°C by passing it through a glass trap
submerged in a -70°C methanol bath. All standard gases supplied to the instruments are





from pressurized 37.5 L treated aluminum alloy cylinders fitted with high-purity, two-
stage gas regulators. Stainless steel tubing (0.32 cm o.d., 0.22 cm i.d.) is used for the
standard gas sample line and the ambient sample line after the cold trap. The automated
sampling module equipped with a VICI 8 ports valve is designed to sample from
separate gas streams (standard tanks and ambient air). According to the comparability
target of WMO/GAW program (WMO, 2019), methane mole fractions are referenced
to a Working High standard (WH) and a Working Low standard (WL). Additionally, a
calibrated cylinder filled with compressed ambient air is used as a Target gas (T) to
check the precision and stability of the system routinely. Diagram of the observing
system during different periods could be seen at Zhou et al. (2004) and Fang et al.
(2013). Here, we focus on the longest continuous measurements of $CH_4$ from August
1994 to May 2017 at WLG. Data gaps in limited periods are because of the malfunction
of instrument and the maintenance of the sampling system.

The records of CO in this study was initinially observed by an RGA-3 gas

chromatograph (GC) equiped with an HgO reduction detector (Trace Analytical Inc.)
since 1994. An automated sampling module was designed to sample from ambient air
and a series 9 standards. Deailed diagram of the system was described by Zhang et al.
(2011). Since 2010, the CO has been measured by the Cavity Ring Down Spectroscopy
instrument (Picarro G1302 and G2401 since 2015). The scale for all of the CO
measurement were further updated to WMO X2014A.

**2. 3 Data processing**
Most on-site $CH_4$ observations were unavoidably influenced by local sources and other
complex conditions (e.g., traffic transportation, various topography). As a result, the
records cannot fully represent the regional atmospheric $CH_4$ in well-mixed conditions
(Liu et al., 2019). To precisely get representative regional records, we exclude $CH_4$
measurements influenced by local sources adjacent to the site (e.g., agricultural fields,
cities, traffic emissions). The hourly $CH_4$ data were classified as Local/Regional events





through the meteorological approach, which was based on essential meteorological
information, similar to previous studies by Zhou et al. (2004) and Liu et al. (2019). In
this study, the $CH_4$ records associated with surface wind from selected sectors (i.e.
NNE-…-ENE in spring, NE-…-SE in summer, NE-…-ESE in autumn, and NE-ENE in
winter) were flagged as local representative. Subsequently, we further rejected portion
of daytime records to minimize the effect of human activities (e.g., rush hours),
including 9:00-13:00 LT (local time) in spring and summer, 9:00-14:00 LT in autumn,
10:00-17:00 LT in winter. Finally, we filtered $CH_4$ data into local events when the
surface wind speed was less than 1.5 m s$^{-1}$ to minimize the very local accumulation.
To understand the influence of local surface wind, the hourly $CH_4$ data was
calculated versus 16 horizontal wind directions (Fang et al., 2013). In this study, we
used the 'polarPlot' function located in the 'openair' package of the statistical software
R (R Core Team, 2020). It shows the bivariate (i.e. wind speed and wind direction)
polar plot of $CH_4$ concentrations, and the concentrations are calculated as a continuous
surface by modelling using smoothing techniques (Carslaw et al., 2006; Diederich,
2007). Also, conditional probability function (CPF) was used to detect the probability
of which wind directions are dominated by high $CH_4$ mole fractions (Uria-Tellaetxe and
Carslaw, 2014). In order to study the pollution transport pathways of air masses at
WLG, the cluster analysis of 3 days back trajectories was applied using the Hybrid
Single-Particle Lagrangian Integrated Trajectory (HYSPLIT) dispersion model
(Draxier and Hess, 1998; Rousseau et al., 2004) on the strength of gridded
meteorological data (2004-2017) from the National Oceanographic and Atmospheric
Administration's Air Resources Laboratory (NOAA-ARL). The trajectories in four
months, including January, April, July and October, were calculated to represent the
seasons of winter, spring, summer and autumn, respectively. The spatial source
distributions of annual $CH_4$ were analyzed using the Potential Source Contribution
Function (PSCF) approach, which computed the conditional probability of the
residence times of air parcels with greater concentration than threshold transport to the





exactly receptor site (Ashbaugh et al., 1985). In this study, PSCF value was calculated
in 0.5×0.5-degree grid cell $(i, j)$:
$$\mathrm{PSCF}_{ij} = m_{ij} / n_{ij} \qquad (1)$$
$n_{ij}$ represents the number of endpoints that terminate in the $ij$th grid cell, while the
number of trajectories with concentration exceed the threshold value was defined as $m_{ij}$
(Polissar et al., 1999). In order to reduce the abnormal influence of small $n_{ij}$ values in
some grid cells, $\mathrm{PSCF}_{ij}$ was further computed by an arbitrary weighting function $W_{ij}$ as
below.
$$W_{ij} = \begin{cases} 1.00 & 3n_{ave} < n_{ij} \\ 0.70 & 1.5n_{ave} < n_{ij} \le 3n_{ave} \\ 0.42 & n_{ave} < n_{ij} \le 1.5n_{ave} \\ 0.05 & n_{ij} \le n_{ave} \end{cases} \qquad (2)$$

$W_{ij}$ represents the weight of cell $(i, j)$, $n_{ij}$ is the number of trajectory endpoints that fall
in the $ij$th grid cell, while the $n_{ave}$ shows the mean number of the endpoints in all grid
cells.
In order to fill the data gaps so as to evaluate the long-term $CH_4$ trend, we applied
the curve fitting approach by Thoning et al. (1989). We also calculated the trend curve
that excluded the influence of seasonal variation, and then got the annual growth rates
of the average of the first derivative of the trend curve. The function consists of the
polynomial part and the annual harmonics part:
$$f(t) = a_0 + a_1 t + a_2 t^2 + \cdots + a_{(k-1)} t^{(k-1)} + \sum_{n=1}^{nh} c_n [\sin(2n \pi t) + \varphi_n] \qquad (3)$$
'k' represents the number of polynomial part. 'nh' is the number of harmonics part. We
applied k = 3 polynomial terms (a quadratic) for multi-year trends and nh = 4 yearly
harmonics for seasonal cycles in this study. The fast Fourier transform (FFT) was
utilized to smooth the fitting residuals (Press et al., 1992).
The significant difference test was applied by the 'scheirerRayHare' function in
the 'rcompanion' package of R software, which is a non-parametric test for two-way
ANOVA analysis. And the multiple comparison was used by Wilcoxon rank sum test





with R (R Core Team, 2020). For the correlation analysis between $CH_4$ and CO, we
obtained the detrended time series of $CH_4$ and CO from 2004-2017 based on the method
by Thoning et al. (1989). The detrended values are denoted as $\Delta CH_4$ and $\Delta CO$. To
obtain the correlation slopes of $\Delta CH_4$ and $\Delta CO$ accurately, a rolling linear regression
was applied to $\Delta CH_4$ and $\Delta CO$ time series by the 'roll_lm' function in 'roll' package of
R (R Core Team, 2020). We successively moved a 24-h time window by 1 h over the
whole time series. Similar to the study by Tohjima et al. (2014), we set 3 criteria to
achieve a better quality control of the slopes. When (i) the number of $CH_4$ record is less
than 5 in 24 hours, (ii) the coefficient variation of the correlation slope is more than
15%, (iii) the absolute value of the correlation slope is less than 0.8 ($|R| < 0.8$), the
correlation slopes were identified as statistically insignificant and inaccurately and were
rejected. In order to understand the year to year variations, we further analyzed the
different periods over 1994-2017. The entire $CH_4$ time series were divided according
to the significant stages or the critical time period of atmospheric $CH_4$ variations from
previously studies (Zhou et al., 2004; Fang et al., 2013; Zhang et al., 2013; Nisbet et
al., 2019; WMO, 2020), appropriately every five years a period. Unless special notes,
the average values in this study were presented with 95% confidence intervals (CIs).

**3   Results**
**3. 1   Diurnal variations**
Generally, distinct diurnal cycles were observed in four seasons during 1994-2017. The
$CH_4$ mole fraction increased from early morning and reached the maximum at noon and
a trough in late afternoon (Fig. 2f). However, differences also existed from different
seasons. In spring, the atmospheric $CH_4$ apparently increased from 9:00 to 13:00 LT
with the daily amplitude of 5.7 ±2.4 ppb. In summer, the elevated $CH_4$ also appeared
during 9:00-13:00 LT at noon, and the daily amplitude was 4.3 ± 2.6 ppb. In autumn,
the diurnal variation showed the mean amplitude of 4.5 ±2.4 ppb, significantly elevated



CH$_4$ reached at 9:00-13:00 LT with one peak at noon. In winter, largely increasing
presented in the daytime at 9:00-17:00 LT, with the largest amplitude of 6.2 ± 2.4 ppb
among four seasons. For the diurnal variation over the whole monitoring period, the
highest CH$_4$ mole fraction was observed in winter and the minimum value was found
in spring (Fig. 2f).

Different patterns for diurnal CH$_4$ cycles were also found over different periods.

In 1994-1997 and 1998-2002, the CH$_4$ mole fractions in winter were apparently higher
than the other seasons (Fig. 2a-b). But its value was the highest in summer during the
period of 2003-2007, 2008-2012 and 2013-2017 (Fig. 2c-e). The atmospheric CH$_4$
values in winter were gradually falling behind the other seasons, and the gaps among
different seasons were increasing, especially for summer. Before 2002, diurnal cycles
in four seasons were ambiguous (Fig. 2a-b), but significant diurnal variations appeared
afterwards (Fig. 2c-e). The peak to trough amplitude almost increased along with the
time in almost all seasons. For example in spring, the amplitude was 6.5 ± 3.0, 4.7 ±
1.8, 5.6 ± 2.6, 6.2 ± 2.4 and 6.8 ± 3.4 ppb over 1994-1997, 1998-2002, 2003-2007,
2008-2012 and 2013-2017, respectively (Table S1).

**3. 2   The impact of local surface winds**
As observed by the previous short-term variations, the atmospheric CH$_4$ at WLG was
significantly influenced by local surface wind from northeast to southeast sectors (Fig.
3f). Slight differences were also found among seasons. In spring, the atmospheric CH$_4$
was enhanced by 2.5-6.5 ppb compared to the seasonal average (1839.7 ± 1.4 ppb)
when the wind was originating from NNE-NE-ENE-E sectors. In summer and autumn,
the wind from NNE-NE-ENE-E-ESE induced higher CH$_4$ mole fractions, with
enhancement of about 3-9.5 ppb and 4 to 18 ppb, respectively. In winter, the CH$_4$ mole
fractions significantly elevated from the wind sectors that same as those found in spring,
with value of 7-21 ppb than seasonal average (1854.5 ± 4.8 ppb). Relatively, the
amplitude of enhancements in winter and autumn were larger than those in spring and



summer.
What interesting is that wind sectors elevating $CH_4$ mole fractions vary in different
periods. The early periods (i.e. 1994-1997 and 1998-2002) were different from the
recent periods (2003-2007, 2008-2012 and 2013-2017). The elevated $CH_4$ was
predominately from about ENE-E-ESE-SE-SSE sectors in early years (Fig. 3a-b), but
evolved to NNE-NE-ENE-E sectors in later years (Fig. 3c-e). Furthermore, the
amplitude of enhancements was almost increasing continuously along with the time.
For example, in autumn, the maximum $CH_4$ mole fractions were from E, ENE, ENE,
NE and ENE sectors in 1994-1997, 1998-2002, 2003-2007, 2008-2012 and 2013-2017,
with the successively increasing enhancements of 8.6, 12.1, 14.7, 16.8 and 19.7 ppb,
respectively.
We applied the CPF to hourly $CH_4$ and CO data by considering intervals of entire
data percentiles including 0-20, 20-40, 40-60, 60-80 and 80-100 to draw the CPF polar
plot. It is clear that different sources only affected $CH_4$ mole fractions on different
percentile range. For example, for most wind speed-directions the CPF probability of
$CH_4$ being greater than the $60^{th}$ percentile was tending to zero (Fig. 4). And it is apparent
that most sources contributed to the less than $60^{th}$ percentiles of $CH_4$ mole fraction (e.g.
40-60) (Fig. 4). The specific sources were prominent for specific percentile ranges. The
wind from the southwest and southeast was important on the cases of the higher
percentiles, resulted in the highest $CH_4$ mole fractions of 1849-1872 ppb for 60-80
percentile and 1872-2031 ppb for 80-100 percentile (Fig. 4). It's more obvious that the
CO showed gradually shifted sources with the increase of percentile ranges. The areas
where the CPF probabilities were higher is to the NW-SW sectors when the percentages
ranged from 0 to $40^{th}$. Nevertheless, when the percentages were larger than $60^{th}$, the
high probability areas completely moved to NE-SE sectors (Fig. 4).



### 3.3 Air mass pathways and potential source distributions


### 3.3.1 Air mass transports


Figure 5 illustrates the cluster analysis to the 3-day back trajectories to WLG during
2004-2017. In spring, the majority of the air masses were from west and northwest
regions, which accounted for about 24% (cluster 3) and 44% (cluster 5) of total
trajectories (Fig. 5a). These air masses were also accompanied with higher $CH_4$ mole
fractions than those from east and northeast regions, i.e. cluster 1 (13.3% of total) and
cluster 4 (11.69%) (Table 1). The largest enhancement was ~18 ppb (relative to spring
average) by cluster 3. In summer, 45% of air masses (cluster 1) were from eastern
regions. But the high $CH_4$ mole fractions were on cluster 2 and cluster 5 from northwest
and west regions, although low percentages were found (cluster 2: 26%; cluster 5: 7%)
(Fig. 5b). The highest $CH_4$ mole fraction was associated with cluster 2, with ~9 ppb
larger than the average in summer. In autumn, large proportion of air masses was
originating from west and southwest station, such as cluster 2 (49%) and cluster 3 (32%)
(Fig. 5c). The highest $CH_4$ was from cluster 3 with enhancement of ~4 ppb than the
seasonal average. Similar to autumn, the air masses were primarily from northwest in
winter, including northwest cluster 3 (59%) and southwest cluster 1 (34%) regions (Fig.
5d). The highest $CH_4$ mole fractions was on cluster 1with the enhancement of ~7 ppb
than the average value.

### 3.3.2 Spatial distribution of potential source regions


In this study, the potential sources were analyzed over different periods, i.e. 2004-2007,
2008-2012 and 2013-2017. Generally, the strong potential sources were located at the
northeast to southeast of the station, especially in summer, but the source regions
differed in various seasons as well as years (Fig. 6). The regions of potential source in
spring (Fig. 6a-c) and winter (Fig. 6j-l) was obviously larger than that in summer (Fig.





6d-f) and autumn (Fig. 6g-i). There were also trends for the $CH_4$ source regions along
with years: i) the area of potential source regions was increasing with the years, and ii)
the location of strong potential sources changed along with the time. For example in
autumn and winter, the strength of $CH_4$ sources were very strong in the southeast to
northeast during 2004-2007 (Fig. 6g & i), and then weaked in 2008-2012 (Fig. 6h & k),
and finally (i.e. 2013-2017), almost vanished in eastern regions but moved to southwest
with very large distribution area (Fig. 6i & l).

**3. 4   Extracting the well-mixed ambient methane**
To precisely understand characteristics of atmospheric $CH_4$, e.g., seasonal cycle or
long-term trend, it is vital to identify the $CH_4$ records that were influenced by local
sources and sinks. In this study, we analyzed hourly $CH_4$ measurements during 1994-
2017, and 47.3% of $CH_4$ data were classified as regional representative, with the
average $CH_4$ mole fraction of $1847.9 \pm 0.3$ ppb. The local representative data was
obviously larger than regional events, with an average value of $1858.2 \pm 0.4$ ppb (Table
2). The proportion of regional events increased slightly before 2012, but significantly
reduced in recent years (e.g., 2013-2017). The filtered regional/local time series was
shown in Figure 7. It can be seen that the $CH_4$ mole fractions obviously increased from
1994 to 2017. The atmospheric $CH_4$ showed strong growth and displayed large
fluctuation at WLG (Fig. 7). In 1995, the average $CH_4$ mole fraction was only 1805.8
$\pm 0.1$ ppb, however, the average value increased 98 ppb by the year of 2016 (1903.8 $\pm$
0.1 ppb) (Table 3).

**3. 5   Correlation analysis between $CH_4$ and CO**
Because part of $CH_4$ and CO in atmosphere are from the same anthropogenic sources
(e.g., fossil fuel combustion), we calculated the regression slopes of $\Delta CO/\Delta CH_4$ from
2004 to 2017 (Fig. S1). Figure 8 presents the average seasonal cycles of the $\Delta CO/\Delta CH_4$



slopes. Generally, the slopes were larger in summer and lower in winter during the
observing period, except for 2004-2007 with the high slope in autumn. Additionally,
the regression slopes increased along with the time, which showed the maximum in
2013-2017 and the minimum in 2004-2007. For the year to year variations, the
$\Delta CO/\Delta CH_4$ slopes showed large fluctuations from 2004 to 2017 at WLG (Fig. 9). The
slopes showed decreasing trend during 2004-2007 but then increased from 2007 to
2010, and again decreased after 2010. In spring and summer, increasing trends appeared
again after 2014. The slopes in summer were almost the largest but the lowest in winter.

**3.6   Variation of long-term records**
**3.6.1   Seasonal cycles**
In order to further investigate the characteristics of atmospheric $CH_4$, we divided the
$CH_4$ observations into two main regions according to the above analysis, including
geographical conditions, the effect of surface winds, the long-range transports and the
potential source distributions. The first region was covered the northeast to southeast
(NNE-…SE) of WLG, which was denoted as City Regions (CR). The second region
was located south to west (S-…-W) of the station and was well known Tibet (Qinghai-
Xizang) Plateau (TP) (Fig. S2). Accordingly, the hourly $CH_4$ records when the surface
wind coming from these sectors were divided into two subsets (i.e. TP and CR). The
long-term variations between the two regions as well as the total regional time series
(Total) were compared and analyzed to explore new sight of atmospheric $CH_4$ variation
at WLG.
Overall, at WLG, the seasonal averages of $CH_4$ were ordered by summer (1850.0
± 0.3 ppb), winter (1847.4 ± 0.3 ppb), autumn (1844.4 ± 0.3 ppb) and spring (1841.2 ±
0.3 ppb), except during 1994-1997 with the maximum in winter and minimum in
autumn (Fig. 10). Seasonal averages in CR were significantly different to that in TP and
also the entire regional data (Total). The seasonal average in TP was mostly higher than



that in CR, except for wintertime (Table S2). The atmospheric $CH_4$ in August was
mostly the maximum and the April was the minimum for the total regional time series
(Total), with the seasonal amplitude of 13.4 ppb. The peak to trough amplitude in CR
(~15 ppb) was higher than that in TP (~13 ppb) during 1994-2016. Additionally,
seasonal amplitudes indicated different trends between CR and TP. For CR, the seasonal
amplitude was firstly dropped and then increased along with time, which were similar
to the variation of total regional events (Total). But for TP, the amplitude displayed a
continuously increasing trend, with values of about 15.9, 19.3, 21.6, 23.4 and 22.4 ppb
in 1994-1997, 1998-2002, 2003-2007, 2008-2012 and 2013-2016, respectively.

### 3.6.2 Long-term trend
In the 1990s, the $CH_4$ growth rates were very low and even negative at WLG.
Subsequently, during 2002-2006, a steady period was found with a near-zero growth
rates. After 2007, the atmospheric $CH_4$ was raised significantly (Fig. 11a). In the year
of 1997/1998, 2000/2001, 2007/2008 and 2011/2012, larger amplitude of the growth
rates was found and strong growth appeared (Fig. 11b). The growth rates fluctuated
evenly with both positive and negative values before 2009. But almost all of the growth
rates showed a positive value after 2009. From 1990s to 2010s, three apparently
developing stages (i.e. highlighted green, blue and red blocks) were presented, the $CH_4$
mole fraction slightly decreased in 1998-2000 (green color), and then go through a
relative steady period in 2003-2006 (blue color), finally increased steadily after 2007
(red color) (Fig. 11).

The overall annual growth rates were $5.1 \pm 0.1$ ppb $yr^{-1}$ over 1994-2016 at WLG

(Table 4). However, the periodic annual growth rats were $4.6 \pm 0.1$, $2.6 \pm 0.2$, $5.3 \pm 0.2$,
$7.6 \pm 0.2$ and $5.7 \pm 0.1$ ppb $yr^{-1}$ in 1994-1997, 1998-2002, 2003-2007, 2008-2012 and
2013-2016, respectively. The $CH_4$ growth rate in CR was significantly different from
that in TP (Fig. S3). In 1994-1997, 2003-2007 and 2013-2016, the growth rates in TP
were obviously larger than that in CR (Table 4). But in 2003-2007 and 2008-2012, the


CR showed higher annual growth rates. In addition, for the entire observing period (i.e.
1994-2016), the growth rates in both TP ($5.2 \pm 0.1$ ppb yr$^{-1}$) and CR ($5.0 \pm 0.1$ ppb yr$^{-1}$
) were similar to the overall annual growth rates (Fig. S3).

**3. 7   Case study for air mass transport**
As described above, the northeast and southeast city regions might act as strong
regional sources influencing the atmospheric CH$_4$ at WLG. Therefore, to analyze the
effect of long-distance transport of emissions from cities, we further excluded the
regional data by air mass transport, and the rest of regional records were denoted as
'TR'. We applied the monthly cluster analysis to hourly trajectories over 2005-2007,
2008-2012 and 2013-2017. The cluster results from city regions were presented in detail
in Fig. S4 and Table S3.

The proportion of trajectories from cities was 40.3%, 32.5% and 6.8% in 2005-

2007, 2008-2012 and 2013-2017, respectively. And about 22.8%, 35.6% and 38.4% of
the regional records were associated with air masses transport from city regions in
2005-2007, 2008-2012 and 2013-2017, respectively (Table 5). The average CH$_4$ value
when air masses transport from city regions ($1863.0 \pm 0.3$ ppb) was obviously higher
than other sectors ($1850.6 \pm 0.2$ ppb) (Fig. S5). However, after excluding the CH$_4$
records when air trajectories transport from city regions, the growth rates of TR in 2008-
2012 ($10.1 \pm 0.1$ ppb yr$^{-1}$) and 2013-2017 ($6.3 \pm 0.1$ ppb yr$^{-1}$) (Table 5) were higher
than the original regional data series (i.e. $7.6 \pm 0.2$ and $5.7 \pm 0.1$ ppb yr$^{-1}$) (Table 4).
And the overall growth rate for TR in 2005-2016 or 1994-2016 was still similar to
original data series (Total), no significant difference was found (Fig. S5).


## 4  Discussion

### 4.1  Anthropogenic emission on temporal patterns

In the early years, the daily cycles of atmospheric $CH_4$ at WLG were not distinct and the amplitude was small (Fig. 2a-b), which were similar to the previous studies by Zhang et al. (2013) and Fang et al. (2013). The ambiguous diurnal patterns indicated that the local sources were weak at WLG in the past. The apparent diurnal cycles after 2000 may be attributed to the intense activities by human (e.g., grazing, fuel burning), which was aggravated in daytime and weak in nighttime (Fang et al., 2013). The meteorological conditions (e.g., diffusion and transport) could contribute to the increasing $CH_4$ amplitudes. The WLG was remote from the populate center, the good diffusion condition in daytime may bring high $CH_4$ mole fractions to the site. The increasing amplitude (Table S1) and $CH_4$ mole fractions over time suggested that the WLG was affected by increasingly local sources (e.g., human activities) (Zhou et al., 2004). The maximum was found in summer in recent years (Fig. 2e) could also be ascribed to the transport of anthropogenic sources by the meteorological factors. In summer, the intensely herd or graze activities around WLG might enhance the regional $CH_4$ emissions and hence contribute to the higher $CH_4$ mole fractions. The higher $CH_4$ mole fraction in winter in past years (Fig. 2a) was probably because of the way of heating (e.g., large biomass burning) as well as the adverse diffusion conditions in cold weather.

The previous study by Zhou et al. (2004) found that higher $CH_4$ mole fractions appeared when the winds come from the ENE-E-ESE-SE sectors at WLG during 1991-2002, which was similar to our study in similar period (Fig. 3a-b). Causes of the elevated $CH_4$ from these wind sectors could be attributed to the large plantation of highland barley as well as high population density in those areas (Fang et al., 2013). Two largest cities Xining and Lanzhou are situated in northeast and east of WLG, respectively, which could emit large amount of $CH_4$ by human activities. Besides,





previous studies on black carbon (BC) and carbon monoxide (CO) indicated that the
emissions from the Yellow River Canyon industrial area, ~500km away from northeast
of WLG may also donate to the high CH$_4$ values originating from ENE and NE sectors
(Zhou et al., 2003). In summer, the prevailing wind directions were from NE-…-ESE
sectors (~46%) (Fig. S6), and the CH$_4$ mole fractions were also higher in the related
sectors. However, in the autumn and winter, although the prevailing wind and high wind
speed were from SSW-…-W sectors (~ 40-50%) (Fig. S6), the high CH$_4$ mole fractions
were from almost the opposite wind sectors of NNE-….-ESE (Fig. 3f), which indicated
that strong local sources were distributed from northeast to southeast (city regions), and
even covered the emissions of natural sources. As time goes on, the wind sectors with
high CH$_4$ mole fractions changed and concentrated on ESE to ENE sectors, and the
amplitudes of enhancements were increasing, which further implied the effect of
stronger emissions from anthropogenic sources in city regions in recent years.

**4. 2   Pollutant sources regions**
**4. 2. 1   Sources regions**
The air masses from east and northeast regions passed over the cities of Xining and
Lanzhou (capital of Gansu province), which is the populated center and industrial area
(Fig. 5). However, the highest CH$_4$ values was not observed when air mass was from
these sectors. Instead, high CH$_4$ mole fractions were frequently observed when air mass
from the northwest to southwest (Table 1). It was possibly due to that the air masses
from west and northwest had passed through the northwest of Qinghai province and the
central area of Xinjiang Uygur Autonomous Region (XUAR), where located Ge'ermu
urban area (the second largest city of Qinghai) with rapid industrial development,
natural gas and petroleum resources exploitation and large crops residue burning
(Zhang et al., 2013). Similar to the CPF percentile analysis (Fig. 4), the southwest or
northwest region away from the site may be also strong source regions.


Most potential source identified in northwestern regions (Fig. 6) was possibly due
to $CH_4$ emissions from the northwest Gansu province, the northwest Qinghai province
and the southeast of XUAR. The different source distribution by seasons could be
attributed to the effect of westerlies or the southeast monsoons (Zhou et al., 2004). The
obviously increasing source region was clear evidence for the strong effect of the
expansion of human activity. Moreover, the pattern of source region moved from the
east to the southwest, especially in autumn and winter, indicated that the southwest
away from the WLG, e.g., Northern India, were gradually becoming a strong $CH_4$
source region. India has abundant cattle as well as extensive large-scale coal mining,
large amount of $CH_4$ emissions may transport from northern India to the northeastern
Tibetan Plateau (Fig. 6i & l). The air mass transport result (Fig. 5d) also support the
result that the southwest air masses (cluster 1) contributed the highest $CH_4$ mole
fractions. The studies of atmospheric Hg at WLG by Fu et al. (2012) also revealed this
phenomenon.

**4. 2. 2   Different sources between $CH_4$ and CO**
The percentile polar plot clearly showed the specific distribution of different $CH_4$ mole
fractions (Fig. 4). The result revealed that most areas around WLG contributed to low
$CH_4$ mole fractions, the southeast and southwest of the site exist two strong source
regions. It is of great possible that the anthropogenic emission from cities (e.g.,
Lanzhou, Chengdu, etc.) was the only cause for high values in the southeast, and the
southwest region away from WLG was possibly due to sources from other countries,
such as India. Unlikely to the $CH_4$, the high CO mole fractions were consistent from
east regions (urbanized area) (Fig. 4), indicating strong anthropogenic sources in city
regions (i.e. Xining and Lanzhou) (Zhang et al., 2011).
The seasonal cycles of $\Delta CO/\Delta CH_4$ slopes (high in summer and low in winter) (Fig.
8) may primarily be due to the effect of monsoons and air mass transport. In summer,
air masses arriving at WLG were predominantly transported from the northeast to east





city regions (e.g., Xining, Lanzhou) with the largest CO mole fractions. In contrast, the
air masses were mainly from the southwest in winter, which carried strong $CH_4$
emissions but few CO emissions (Zhang et al., 2011) (Table 1). Hence the opposite two
air mass transport lead to a peak in summer and a trough in winter (Fig. 8). Moreover,
we could see apparently regional polarization in the concentration ratio of $CH_4$ and CO
(Fig. S7), implying the different strong source distribution between $CH_4$ and CO at
WLG. The cluster results (Fig. 5b & d) and the potential sources analysis (Fig. 6f & l)
also support this seasonal variation. Tohjima et al. (2014) found an opposite variation
at Hateruma Island, which showed low slope values in summer. It could be attributed
to different local sources and sinks, suggested the special topography condition and
local source distribution around WLG. The large $\Delta CO/\Delta CH_4$ fluctuations (Fig. 9) over
the study period was likely because of the anomaly years of different $CH_4$ or CO mole
fractions as well as source regions. In 2007, large increase of $\Delta CH_4$ appeared, and from
2010 to 2013, the $\Delta CO$ decreased significantly (Fig. S1). Before 2010, large air masses
and potential source regions were identified in eastern regions (city regions) with the
highest CO emission (Fig. 5; Fig. 6). After 2010, the southwest regions showed high
contributions, with the highest $CH_4$ emission but relatively lower CO emission.
Therefore, obviously variation of the slopes presented with almost an increase in 2007-
2010 and a decrease after 2010 (Fig. 9).

**4. 3  Long-term variations**
**4. 3. 1  Seasonal cycles**
The seasonal variations with maximum in summer and minimum in spring at WLG
(Fig. 10) were consistent with the previously short-term studies, e.g., Zhang et al.
(2013) in 2002-2006. However, it was almost opposite to observation in the adjacent
stations such as Lin'an, Shangdianzi and Longfengshan in China (Fang et al., 2013,
2016), as well as most global stations in the northern hemisphere, e.g., MLO, BRW and





JFJ (Dlugokencky et al., 1995; Loov et al., 2008). We further compared similar
WMO/GAW global stations in the north hemisphere, including MLO (19.54° N, -
155.58° E, 3397m a.s.l.) (Dlugokencky et al., 1995, 2019a), JFJ (46.55° N, 7.99° E,
3580m a.s.l.) (Zellweger et al., 2016), MNM (24.29° N, 153.98° E, 7.1m a.s.l.)
(Matsueda et al., 2004; Tsutsumi et al., 2006), as well as the marine boundary layer
(MBL) from NOAA/ESRL lab at similar latitude (Dlugokencky et al., 2019b). It can be
seen that the stations in the northern hemisphere (i.e. MLO, JFJ and MNM) and the
MBL showed an opposite trend to WLG with the minimum in summer and maximum
in winter or spring (Fig. 12). And the seasonal amplitude at WLG (~14 ppb) was lower
than many other sites in the northern hemisphere, about 35-70 ppb, e.g., MLO, BRW,
bialystok in Poland, ochsenkopf in Germany and beromunster in switzerland
(Dlugokencky et al., 1995; Thompson et al., 2009; Popa et al., 2010; Satar et al., 2016).
MBL also showed larger amplitude than that in WLG (Fig. 12). The low amplitude at
WLG was possibly because of high elevation of continental mountain sites (e.g., WLG,
JFJ), where were relatively less affected by local influences than the coastal/island sites
(Yuan et al., 2019). The peak in summer at WLG may be attributed to larger grazing
and human activities than other seasons. The $CH_4$ emissions from yaks and other
ruminants in Tibetan Plateau (alpine pasture) were very strong in summer, preceded
only by paddy emission (Fang et al., 2013; Zhang et al., 2013). Furthermore, the
photochemical capacities were very weak (high altitude) and the dynamic transport by
air flow from polluted northeast/southeast region was also strong in summer at WLG,
which all induced high $CH_4$ mole fractions in summer and consequently an opposite
trend with other sites (Ma et al., 2002; Xiong et al., 2009).

**4. 3. 2   Long-term trends in different observing periods**
The entirely fluctuant trend of atmospheric $CH_4$ in 1994-2016 at WLG (Fig. 11) was
similar to the global trend reported by quite a few studies (Bergamaschi et al., 2013;
Rigby et al., 2017; Nisbet et al., 2019). The previous study by Zhou et al. (2004) showed



the $CH_4$ annual increase of 4.5 ppb $yr^{-1}$ in 1992-2001, which was similar to that in 1994-
1997 (4.6 ± 0.1 ppb $yr^{-1}$) and 1997-2002 (2.6 ± 0.2 ppb $yr^{-1}$) in our study (Table 4).
Tohjima et al. (2002) found similar growth rates that the $CH_4$ at Cape Ochi-ishi and
Hateruma Island in1995-2000 was increased about 4.5 and 4.7 ppb $yr^{-1}$, respectively. In
early 1990s, the $CH_4$ trend at WLG is very low, which was similar to the global growth
rates (Fig. 11b). The levels of OH radicals may have controlled the decrease or increase
of $CH_4$ in the atmosphere during this period (Dlugokencky et al., 1998; Rigby et al.,
2017; Turner et al., 2017). The growth rates were high in 1998 (Fig. 11b), which may
have been due to the high temperatures, large biomass burning and weak destruction
(Cunnold et al., 2002; Lelieveld et al., 2004; Simmonds et al., 2005).
The continuously larger $CH_4$ growth rates after 2007 at WLG (Fig. 11), e.g., 7.6 ±
0.2 ppb $yr^{-1}$ in 2008-2012, 5.7 ± 0.1 ppb $yr^{-1}$ in 2013-2016 (Table 4), were similar to the
recent study by Nisbet et al. (2016) and (2019), which showed the global $CH_4$ increased
by 5.7 ± 1.2 ppb $yr^{-1}$ in 2007-2014, and the much higher of 12.7 ± 0.5 ppb $yr^{-1}$ in 2014,
10.1 ± 0.7 ppb $yr^{-1}$ in 2015, 7.0 ± 0.7 ppb $yr^{-1}$ in 2016 and 7.7 ± 0.7 in 2017 ppb $yr^{-1}$.
The average growth rate in the northern hemisphere was 7.3 ± 1.3 ppb in 2007 and 8.1
± 1.6 ppb in 2008 (Dlugokencky et al., 2009), which was also similar to the observation
at WLG (Table 4). After 2007, most sites in the northern hemisphere displayed large
$CH_4$ growth rates. Fang et al. (2013) showed that the annual growth rate of $CH_4$ was
9.4 ± 0.2 ppb $yr^{-1}$ in 2009-2011 at WLG, which were a little higher than our study in
similar period. The adjacent stations in China also revealed high $CH_4$ growth rates, e.g.,
8.0 ± 1.2 ppb $yr^{-1}$ at Lin'an in 2009-2011, 7.9 ± 0.9 ppb $yr^{-1}$ at Longfengshan in 2009-
2011, and 10 ± 0.1 ppb $yr^{-1}$ at Shangdianzi in 2009-2013 (Fang et al., 2013, 2016),
which was higher than the similar period in 2008-2012 or 2013-2016 at WLG (Table
4). The $CH_4$ measurements in other countries, e.g., Beromünster tall tower station also
showed high growth rate of 9.66 ppb $yr^{-1}$ in 2012-2014 (Satar et al., 2016). The very
warm temperatures, large biomass burning and the climatic anomaly e.g., El Niño, La
Niña event, were likely enhanced the $CH_4$ emissions after 2007 (Dlugokencky et al.,





2009). Additionally, the anomalously sharply increasing or decreasing years (e.g.,
2007/2008) may have a significant influence on the overall $CH_4$ trend (Fig. 11), and
these frequent anomalies appeared in most long-term observation stations, e.g., MLO
in USA (Dlugokencky et al., 2009), Mount Zugspitze in Germany (Yuan et al., 2019),
which were also possibly attributed to climatic forces, such as the exception during the
El Niño oscillation, forest fires, volcanic eruptions, and extreme weather events
(Keeling et al., 1995; Dlugokencky et al., 2009; Keenan et al., 2016; Nisbet et al., 2019).
The study by Satar et al. (2016) at Beromünster, Switzerland explained that the short-
term spikes were possibly related to emissions from agricultural activities, while the
longer lasting peaks were because of air mass transport and mixing.

It is well established that the human activities are mainly responsible for the recent

rapid $CH_4$ growth rates and anomalies. The analysis from Emissions Database for
Global Atmospheric Research (EDGAR) showed the $CH_4$ emission by per sector in
China (Fig. S8) (Crippa et al., 2019). During the observing period, the waste, the oil
and natural gas and open burning continuously emitted large amount of $CH_4$ into the
air. After 2000, the $CH_4$ emission from solid fuel increased greatly in China. After 2003,
the $CH_4$ emitted from rice cultivation also increased continuously (Fig. S8). The
increased emissions from these sectors may greatly contribute to the $CH_4$ increase at
WLG, as well as the other regions in China. In addition, the recent studies reveled that
China's coal sector may have dominated the clearly positive trend in recent years, which
contributed the highest proportion of the anthropogenic $CH_4$ emissions (~33%)
(Janssens-Maenhout et al., 2019; Miller et al., 2019). In 2010-2015, China's coal
production increased obviously (from 3400 to 4000 million metric tons), but emissions
trends of $CH_4$ by rice, agriculture, ruminants, waste, and oil/gas have grown slightly
and even remained flat (EIA, USA). The isotopic evidence suggests that the significant
increase of biogenic emissions was the dominant factor of $CH_4$ rise, especially in the
tropical wetlands with strong rainfall anomalies, or the agricultural sources such as rice
paddies and ruminants, fossil fuel emissions have not been the main cause (Nisbet et



al., 2016). The study by Chen et al. (2013) illustrated that the warming (0.2 °C per
decade) in the Tibetan Plateau resulted in substantial emission of $CH_4$ due to the
permafrost thawing and glaciers melting. However, up to now, the specific causes of
such distinct variability around the years, e.g., the spikes or near-zero $CH_4$ growth rates,
are not yet determined.

### 681    4.3.3  Annual growth rate in Qinghai-Tibetan Plateau

Although similar annual growth rates were found among the City Regions (CR), the
Tibet Plateau (TP) and total regional records (Total) in the entire observing period
(1994-2016) (Fig. 11), significant differences were found in short-term periods (Fig.
S3). In 2013-2016 (Table 4), the TP showed larger growth rate than that in CR, implying
stronger $CH_4$ source in the Tibetan Plateau in recent years. The seasonal amplitude in
the Tibetan Plateau was continuously increasing, which also revealed that the Tibetan
Plateau was intensively affected by strong regional sources. Without air mass transport
from the city regions, the significantly increased annual growth rate (TR) in 2008-2012
($10.1 \pm 0.1$ ppb $yr^{-1}$) and 2013-2016 ($6.3 \pm 0.1$ ppb $yr^{-1}$) (Fig. S5 and Table 5) suggested
that there were possibly other strong $CH_4$ sources around WLG not from cities.
Northern India and eastern China were obviously the largest two source regions of $CH_4$
at WLG (Fig. S9) (Crippa et al., 2019). Since the Tibetan Plateau was coincidently
trapped in the middle of the largest increased source areas, the atmospheric $CH_4$ at
WLG was very likely dominated by long-distance transport from these two regions.
Although $CH_4$ emissions increased slowly during 1994-2002, and even negative trend
appeared in southeast China (Fig. S9), significantly increased emissions appeared in
both southeast and southwest Asia after 2007. The rapidly increased $CH_4$ would
probably make it difficult to meet the goals of carbon emission reduction in the future.
Especially on the scenario of quick increasing $CH_4$ on the Qinghai-Tibetan Plateau due
to the emission from two largest source regions. In view of the integrated eco-
environmental change processes and unique topography in the Qinghai-Tibetan



Plateau, it may provide us one of the last precious regions to study global climate
changes (Chen et al., 2013). The anomalously year to year fluctuations of atmospheric
$CH_4$ in Tibetan Plateau were unquestionably a warning or alarm to the world, and the
unprecedented annual growth rate might be a dangerous signal to global climate change.

**5 Conclusion**
The atmospheric $CH_4$ at Mt.Waliguan increased continuously during 1994-2017.
Although near-zero and even negative growth appeared in some particular periods, e.g.,
1999-2000, and 2004-2006, the overall trend of $CH_4$ was increased rapidly, especially
in recent decade. Obvious diurnal cycle was found with the peak at noon and a trough
at late afternoon. Due to the unique geophysical locations and transport pathway, the
seasonal averages of $CH_4$ at WLG displayed an opposite trend with sites in the northern
hemisphere, with summer maximum and spring minimum. Large amount of air masses
was from west and northwest regions of WLG, which accompanied with higher $CH_4$
mole fractions than that from city regions. The Northern India possibly became a strong
source of $CH_4$ to WLG rather than city regions before.
As time goes by, the temporal patterns (e.g., seasonal amplitude), the annual
variations, the long-term trends or potential source distribution of $CH_4$ at WLG are all
changing. Thus, the long-term verification is extremely important to accurately
understand $CH_4$ variations. The case study in Qinghai-Tibetan Plateau revealed
unprecedented annual growth rates of $CH_4$. In recent years, the Tibetan Plateau even
showed larger growth rate than that in city regions. Tibetan Plateau was with the highest
average altitude and was almost impervious to strong human activities. There is no
doubt that the anomalously variation and the unprecedented annual growth rate of
atmospheric $CH_4$ in this region might be a dangerous signal to global climate change.





*Data availability.* The gridded meteorological data (2004-2017) from NOAA-ARL was
available at ftp://arlftp.arlhq.noaa.gov/pub/archives/gdas1/. The data from MLO, JFJ,
MNM station was downloaded from World Data Centre for Greenhouse Gases
(WDCGG) at https://gaw.kishou.go.jp/. The MBL data was available at
ftp://aftp.cmdl.noaa.gov/data/trace_gases/CH$_4$/flask/surface/. The geographical
distribution of annual emission data by Emissions Database for Global Atmospheric
Research (EDGAR) was from website
https://edgar.jrc.ec.europa.eu/overview.php?v=50_GHG.


*Author contributions.* SL, SF and ZF designed the research. SL performed the data
processing with assistance of SF and MG. The station were monitored, maintained ML
and PL, and they collected, preprocessed, and provided the hourly observational
dataset. SL and SF finished the manuscript with contributions from all the co-authors.


*Competing interests.* The authors declare that they have no conflict of interest.


*Acknowledgments.* This study was funded by the National Key Research and
Development Program of China (2017YFC0209700). We also thanks to the staff who
have contributed to the system installation and maintenance at the Waliguan in past
decades.



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



**Table 1.** The statistics for cluster analysis result for both CH$_4$ and CO at WLG station. The clusters
from urban areas are highlighted with face bold.

| | Cluster | Number | Average CH$_4$ mole fraction |
|---|---|---|---|
| Spring | **1** | **1243** | **1853.4 ± 2.7** |
| | 2 | 685 | 1852.6 ± 3.4 |
| | 3 | 2231 | 1877.6 ± 2.5 |
| | **4** | **1093** | **1850.5 ± 2.2** |
| | 5 | 4108 | 1860.8 ± 1.5 |
| Summer | **1** | **3981** | **1869.9 ± 1.2** |
| | 2 | 2244 | 1878.5 ± 2.6 |
| | **3** | **1040** | **1866.3 ± 2.6** |
| | 4 | 916 | 1857.8 ± 2.4 |
| | 5 | 578 | 1876.3 ± 5.1 |
| Autumn | **1** | **1133** | **1870.1 ± 3.7** |
| | 2 | 4235 | 1868.8 ± 1.3 |
| | 3 | 2745 | 1873.6 ± 1.6 |
| | 4 | 550 | 1865.2 ± 3.6 |
| Winter | 1 | 3066 | 1879.8 ± 2.1 |
| | **2** | **601** | **1872.6 ± 3.4** |
| | 3 | 5261 | 1865.8 ± 1.0 |






**Table 2.** The statistics of filtered $CH_4$ data series over different periods during 1994-2017 at WLG
station.

| Year | Regional representative | | | Local representative | | |
|------|-------|----------------|------------|-------|----------------|------------|
| | Hours | Percentage (%) | Mean (ppb) | Hours | Percentage (%) | Mean (ppb) |
| 1994-1997 | 6481 | 46.9 | 1801.7 ± 0.5 | 7346 | 53.1 | 1806.1 ± 0.5 |
| 1998-2002 | 5332 | 47.6 | 1832.6 ± 0.7 | 5877 | 52.4 | 1837.9 ± 0.6 |
| 2003-2007 | 12421 | 49.2 | 1832.2 ± 0.3 | 12850 | 50.8 | 1839.2 ± 0.3 |
| 2008-2012 | 11314 | 49.2 | 1856.4 ± 0.4 | 11664 | 50.8 | 1867.0 ± 0.4 |
| 2013-2017 | 10140 | 43.6 | 1894.9 ± 0.5 | 13121 | 56.4 | 1907.4 ± 0.5 |
| 1994-2017 | 45688 | 47.3 | 1847.9 ± 0.3 | 50858 | 52.7 | 1858.2 ± 0.4 |




**Table 3.** Yearly average $CH_4$ mole fractions at WLG station.

| Year | Mean (ppb) | year | Mean (ppb) |
|---|---|---|---|
| 1995 | 1805.8 ± 0.1 | 2006 | 1835.6 ± 0.2 |
| 1996 | 1804.6 ± 0.2 | 2007 | 1839.9 ± 0.5 |
| 1997 | 1806.8 ± 0.2 | 2008 | 1865.9 ± 0.3 |
| 1998 | 1827.0 ± 0.1 | 2009 | 1849.1 ± 0.1 |
| 1999 | 1820.2 ± 0.1 | 2010 | 1857.9 ± 0.2 |
| 2000 | 1820.0 ± 0.2 | 2011 | 1872.6 ± 0.4 |
| 2001 | 1849.2 ± 0.4 | 2012 | 1881.2 ± 0.3 |
| 2002 | 1835.5 ± 0.2 | 2013 | 1896.3 ± 0.2 |
| 2003 | 1842.5 ± 0.3 | 2014 | 1890.4 ± 0.1 |
| 2004 | 1836.7 ± 0.1 | 2015 | 1905.6 ± 0.3 |
| 2005 | 1837.4 ± 0.1 | 2016 | 1903.8 ± 0.1 |






**Table 4.** Annual growth rates of atmospheric $CH_4$ in the City Regions (CR), the Tibet Plateau (TP),
and original total regional records (Total) during 1994-2016 at WLG station.

|       | 1994-1997 | 1998-2002 | 2003-2007 | 2008-2012 | 2013-2016 | 1994-2016 |
|-------|-----------|-----------|-----------|-----------|-----------|-----------|
| CR    | 2.8 ± 0.1 | 3.6 ± 0.2 | 5.6 ± 0.2 | 7.1 ± 0.2 | 5.5 ± 0.1 | 5.0 ± 0.1 |
| TP    | 3.4 ± 0.1 | 3.0 ± 0.2 | 6.8 ± 0.2 | 5.6 ± 0.2 | 7.3 ± 0.1 | 5.2 ± 0.1 |
| Total | 4.6 ± 0.1 | 2.6 ± 0.2 | 5.3 ± 0.2 | 7.6 ± 0.2 | 5.7 ± 0.1 | 5.1 ± 0.1 |






**Table 5.** The statistics of $CH_4$ data without air mass transport from city regions (TR) over different
periods during 2005-2016 at WLG station.

| | Transport regions | Hours | Percentage (%) | Average (ppb) | Updated growth rate (ppb yr-1) |
|---|---|---|---|---|---|
| 2005-2007 | TR | 6922 | 77.2 | 1824.9 ± 0.2 | 2.7 ± 0.2 |
| | City | 2041 | 22.8 | 1835.9 ± 0.5 | - |
| 2008-2012 | TR | 7060 | 64.4 | 1853.7 ± 0.2 | 10.1 ± 0.1 |
| | City | 4254 | 35.6 | 1861.0 ± 0.3 | - |
| 2013-2016 | TR | 4152 | 61.6 | 1888.2 ± 0.3 | 6.3 ± 0.1 |
| | City | 2591 | 38.4 | 1888.5 ± 0.5 | - |
| 2005-2016 | TR | 18134 | 67.1 | 1850.6 ± 0.2 | 7.0 ± 0.1 |
| | City | 8886 | 32.9 | 1863.2 ± 0.3 | - |


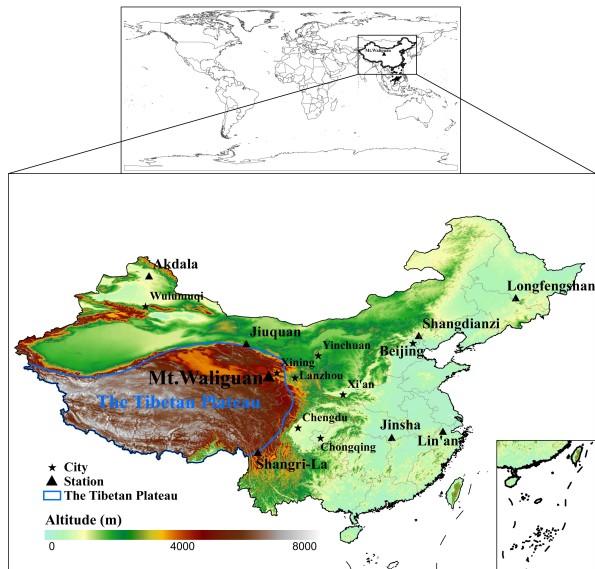

**Figure 1.** Location of Mt.Waliguan WMO/GAW global station as well as other regional stations in China. The gradient color indicates altitude. The digital elevation model (DEM) was downloaded from Geospatial Data Cloud site, Computer Network Information Center, Chinese Academy of Sciences (http://www.gscloud.cn), and then processed by ArcGis software. The China map was derived from © National basic Geomatics Center of China (http://www.ngcc.cn/ngcc/). The world map was obtained from © OpenStreetMap (https://www.openstreetmap.org/). And the other shpfile file data and entire map were created by ArcGis software.





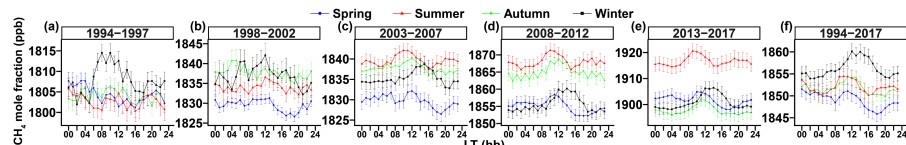

**Figure 2.** Diurnal $CH_4$ cycles in different periods from 1994 to 2017 at WLG station. The lines with

different colors represent various seasons. Error bars indicate the 95% confidence intervals.





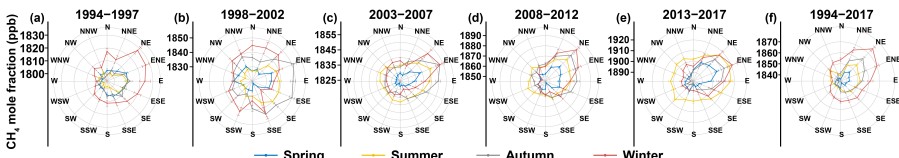

**Figure 3.** Wind-rose distribution of average hourly CH$_4$ records from 16 horizontal wind directions over different periods in 1994-2017 at WLG station. The different colors represent the CH$_4$ data in different seasons. Error bars in all directions indicate 95% confidence intervals.

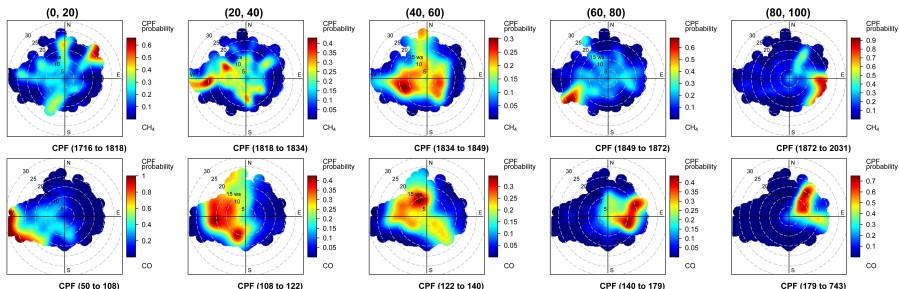

**Figure 4.** The polar plot of the distribution of CH$_4$ and CO concentration probability in different

percentile ranges at WLG station. The analysis was based on the conditional probability functions

(CPF) by Ashbaugh et al. (1985). The top plots show the measurements of CH$_4$ from 1994 to 2017.

The bottom plots show the CO measurements in 2004-2017. 'ws' means the wind speed. The values

in the bottom of each panel show the range of concentration in relevant percentile range. Gradient

colors represent the levels of CPF probability in different percentile ranges.

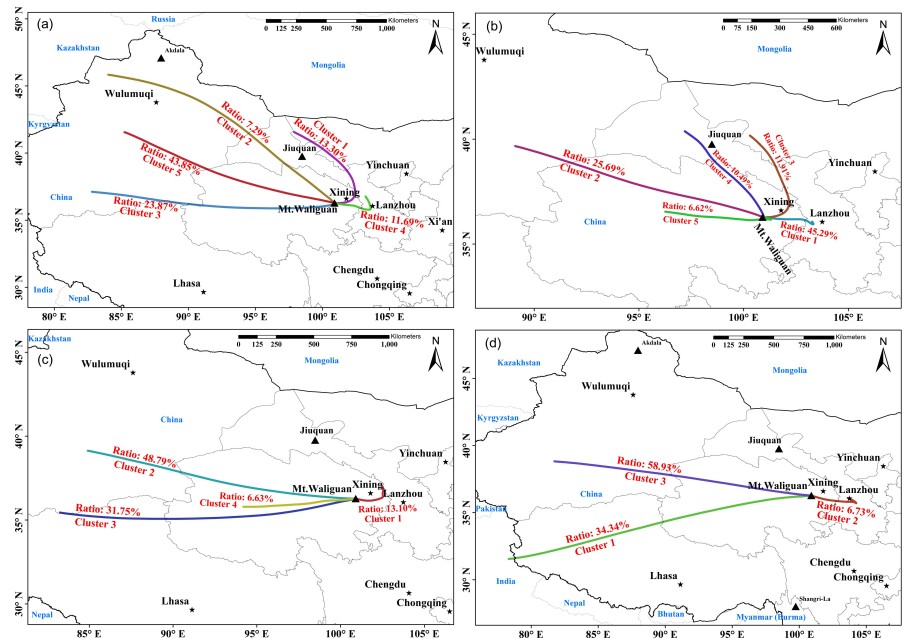

**Figure 5.** Cluster analysis to the 72-h back trajectories in different seasons during 2004-2017 ending at WLG station. The (a), (b), (c) and (d) represents spring, summer, autumn and winter, respectively. The lines with different colors denote different cluster analysis results. The proportion of trajectories on each cluster is also marked.

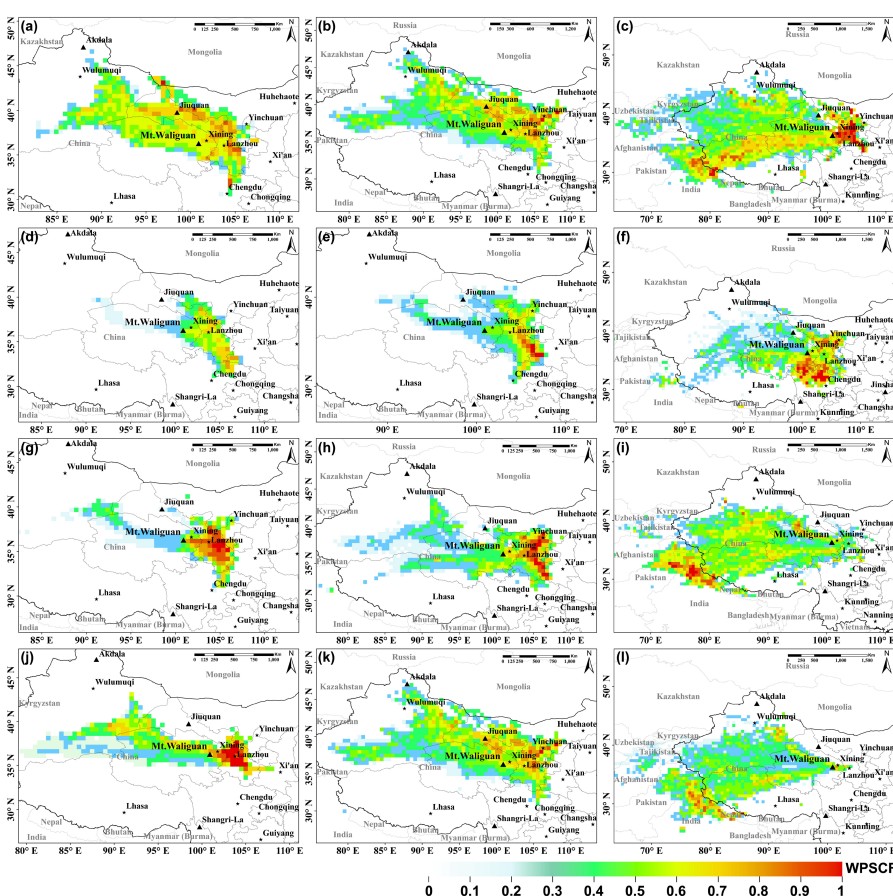

**Figure 6.** Geographical distribution of weighted potential source of CH$_4$ in different periods over

1994-2017 at WLG station. The gradient color shows strong levels of potential source regions in

different seasons, i.e. spring (a, b, c), summer (d, e, f), autumn (g, h, i) and winter (j, k, l), and

different periods, i.e. 2004-2007 (a, d, g, j), 2008-2012 (b, e, h, k) and 2013-2017 (c, f, i, l).



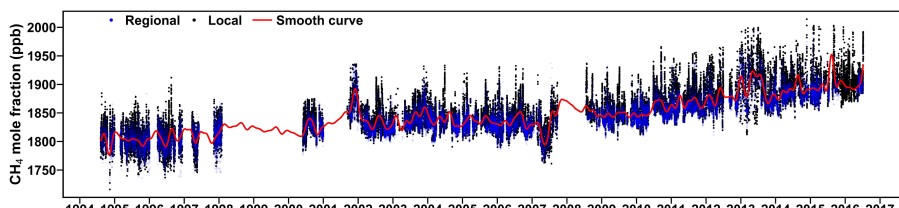

**Figure 7.** Filtered hourly $CH_4$ data series from 1994 to 2017 at WLG station. The blue points with

transparency represent regional events. The black points are the filtered local events. The red lines

are calculated smooth values to the regional data by curve-fitting routine of Thoning et al. (1989).





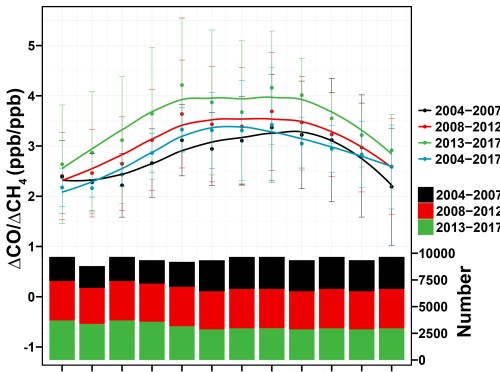

**Figure 8.** Average seasonal variation of $\Delta CO/\Delta CH_4$ slopes in different periods over 2004-2017 at WLG station. The error bars show the standard deviation of the monthly averages. The vertical bars are the monthly numbers of data in different periods.





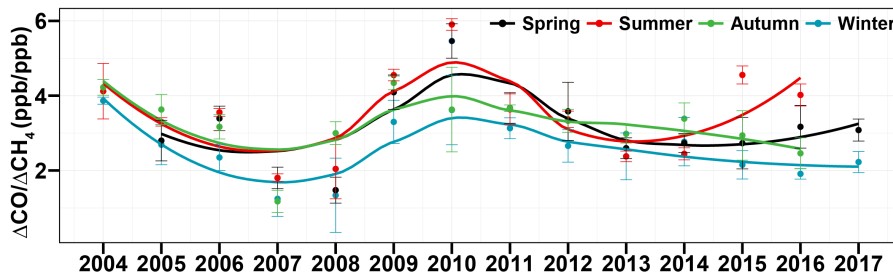

**Figure 9.** Long-term trend of $\Delta CO/\Delta CH_4$ slopes over 2004-2017 at WLG station.

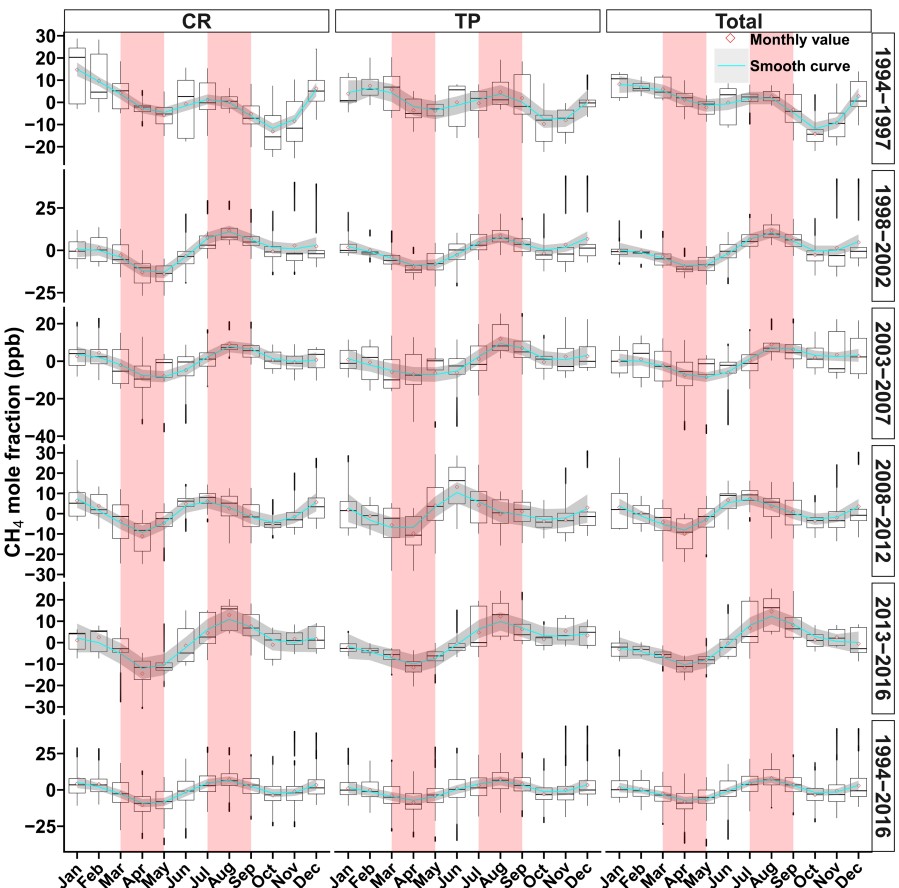

**Figure 10.** Monthly variations of regional CH$_4$ mole fractions from 1994 to 2016 at WLG station.

The 'CR', 'TP' and 'Total' represents the measurements from the City Regions, the Tibet Plateau

and the original total regional records, respectively. The box respectively shows the 25$^{th}$ percentile,

the median and the 75$^{th}$ percentile from bottom to top. The bottom and the top whisker respectively

reaches the minimum and 1.5 times the IQR (interquartile range). The black points are identified as

outliers. The red squares are the averages. The cyan lines are the smooth curve of averages using

the method of loess (Local Polynomial Regression Fitting). The gray bands are the 95% confidence

interval of smooth curve.





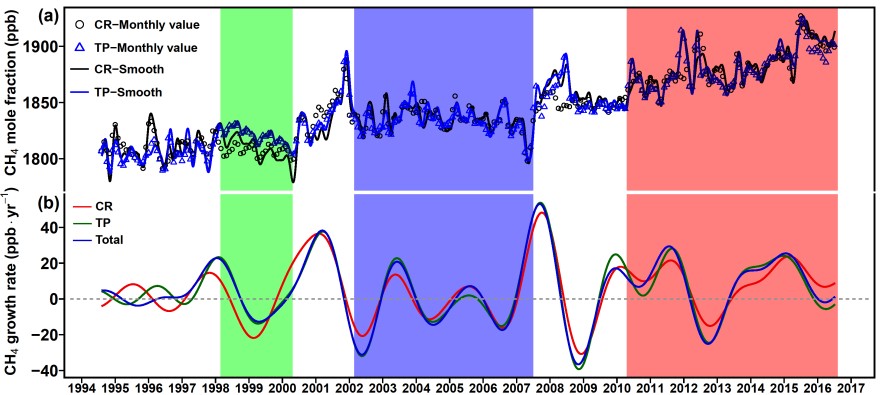


**Figure 11.** The top panel (a) shows the smooth curve and monthly values of $CH_4$ mole fraction in
the City Regions (CR) and the Tibet Plateau (TP) during 1994-2016 at WLG station. The bottom
panel (b) is the annual growth rates of atmospheric $CH_4$ records from CR, TP as well as the total
regional time series (Total). The growth rates are calculated from the first derivative of trend curves.
The smooth curve and the trend is calculated by the method of Thoning et al. (1989).



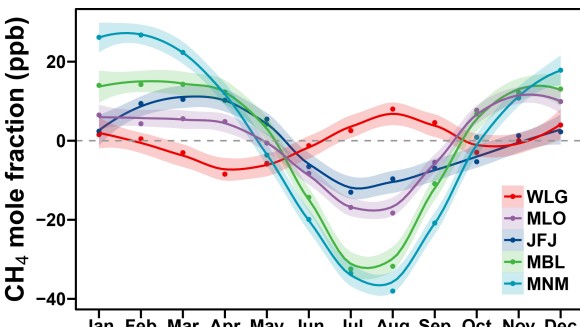

**Figure 12.** The seasonal cycles of atmospheric CH$_4$ observed over 1994-2017 at WMO/GAW global stations of Mauna Loa (MLO), Jungfraujoch (JFJ), Minamitorishima (MNM) and Mt.Waliguan (WLG) in the northern hemisphere. The data of other sites (except WLG) are from WDCGG. The data in the marine boundary layer (MBL) are from NOAA / ESRL lab at the similar latitude to WLG.