# Peer review of "Changing characteristics of atmospheric CH₄ on the Tibetan Plateau, records from 1994 to 2017 at Mount Waliguan station"

_Atmospheric Chemistry and Physics, 2020_

## Referee Comment (RC1) · Anonymous Referee #1 · 19 Aug 2020

review of manuscript acp-2020-481

Title: Changing characteristics of atmospheric CH4 on the Tibetan Plateau, records from 1994 to 2017 at Mount Waliguan station Authors: Shuo Liu et al.

General comments: The manuscript presents an analysis of 24 years of continuous in-situ ground-based methane observations at the remote Mount Waliguan station in China. Such records are very essential for monitoring atmospheric variability and trends of this important greenhouse gas. It is challenging to keep those observations up and running for decades. Thus, I fully support publication and release of these data. However, the analysis of the data is very descriptive and lacks conclusive statements.

There is hardly any novel insight into the CH4 cycle, which is currently drawn from the analysis. Several filters are applied to the data but it is difficult to follow how the filters are applied and which filtered dataset is used in which analysis. The paper may profit from merging chapter 3 (Results) and chapter 4 (Discussion). Moreover, the paper contains quite a number of linguistic flaws. Proofreading by an English native speaker is necessary. Some language mistakes are listed below. The paper is within the scope of 'Atmospheric Chemistry and Physics' but – to my mind – requires major revisions prior to its acceptance for ACP. See my specific comments below.

Specific comments:

Abstract is rather long, you may consider shorten it.

Lines 18-19: write "A 24-year long-term record of atmospheric methane (CH4) measured in-situ at the Mt. Waliguan station, the only ..., is presented."

Line 20: "... 1994-2017 ...": why don't you use more recent data (e.g. until end of 2019)?

Line 20: "continuously" need to be "continuous"

Lines 20-22: "continuous increase" but "even negative growth trend ... in particular periods" is a contradiction.

Lines 24-25: "but unprecendeted elevated  $\sim$ 100 ppb" sounds awkward

Line 24: "... historic high of 1903.8  $\pm$  0.1 ppb in 2016 ..."; what about 2017?

Line 26: what does " $\Delta CO/\Delta CH4$ " mean? Why do you look at CO to CH4 ratios?

Line 27: add "elevated" that is reads "... opposite to other elevated sites ..."

Line 30: "(the Northern India)" -> "(Northern India)"

Line 35: delete "What is interesting" I do not believe that you want to mention uninteresting things in the abstract of your manuscript.
Line 47: "anomalously" -> "anomalous"

Lines 47-49: If this statement is presented prominently in the abstract, it also needs to be discussed in the manuscript. Moreover, when looking at Fig. 11 (lower panel), I do not really see an increasing growth rate (but only a mostly positive growth rate).

Line 60: write "CH4 has an 8-12 years lifetime ...."

Line 64: "... CH4 rapidly increased ...."

Lines 65-66: write "Results from ice core analyses in Antarctica showed ...."

Line 67: write "... has reached a level unprecedented over ..."

Line 68: awkward English

Line 71: delete "special"

Line 73: write "... (Nisbet, et al., 2019) followed by a renewed CH4 increase since then".

Line 78: don't start a sentence with "And ...."

Lines 78-79: write "unexpected" instead of "not expected"; why the increase is "unexpected"?

Line 84: explain "C.E."

Line 85: write "Atmospheric CH4 is mainly ...."

Lines 105-106: write "Systematic observations are a prerequisite to get an accurate understanding of spatial and temporal behavior of atmospheric CH4 concentrations."

Line 113: awkward English

Lines 114-118: There are many more stations continuously measuring CH4 levels in the atmosphere; why did you select these ones?

ACPD
Lines 118-127: this is largely a repetition, merge with paragraph on the previous page (lines 85-104).

Lines 146-147: write "... which is the longest record in China."

Lines 157: write "WLG is the only ...."

Line 158: write "... and is run by the China Meteorological Administration ..."

Line 166: "Tibetan Plateau"

Line 167: "dial variations ... are influenced ...."

Line 221: write "... were flagged as locally influenced."; if the data are locally influenced they are poorly representative.

Lines 220-225: add percentages of rejection according to the individual filters.

Line 224: "We filtered CH4 data into local events ..." sounds strange, how can you filter data into events?

Line 226: write "hourly CH4 data was binned into 16 horizontal wind direction classes ...."; is this done for all data or the regionally representative data only?

Lines 235-236: elaborate on HYSPLIT: what's the spatial resolution of the model, what is the height above sea level of WLG in the model?

Line 239: write "The trajectories for January, April, ..."

Line 285: "appropriately every five years a period" sounds strange.

Results and Discussion chapters (Chapters 3 and 4): I strongly encourage the authors to merge these two chapters. Results should be discussed in stronger conjunction with existing literature, preferably from Asian sites; I don't see the rationale why some findings are compared with conclusions from non-elevated sites in Europe.

Lines 297-298: awkward English
End of Chapter 3.1: interpretation/conclusion missing.

Line 315: "As observed by the previous short-term observations"; I don't understand this statement.

Line 326: Delete "What interesting is"

Line 345: Write "It's obvious that CO showed ...."

Line 348: write "when percentages ranged from 0 to 40."; write "When data exceeded the 60% percentile, the high area probability areas ..."

End of Chapter 3.2: interpretation/conclusion missing.

Line 365: awkward English

Lines 368-369: awkward English

End of Chapter 3.3: interpretation/conclusion missing.

Line 389: write "regionally representative"

Line 390: write ""locally influenced"h

Line 391: "data was ... larger than .... events"; awkward English

Line 394: Does it hold true for all data or regionally representative data?

Line 396 (reference to Fig. 7): I suggest to move Fig. 7 above and make it to Fig. 2; show first the whole dataset before you analysis of the data

Line 402: what is deltaCO and deltaCH4? I assume it is excess CO and excess CH4, i.e. data above baseline (looks like when looking at Fig. S1); if so, how was the background determined? Please elaborate; show also the CO time series.

Line 411: incomplete sentence.

End of Chapter 3.5: what do we learn from the different deltaCO to deltaCH4 ratios?
Lines 415-419: this is a different filtering than the one described above, right? Move these lines to Chapter 2.3.

Line 426: write "... the seasonal averages of regionally representative CH4 were ...."; do these numbers relate to regionally representative data?

End of Chapter 3.6.1: interpretation/conclusion missing.

Line 442: write "... growth rates were very small or even negative ...."

Lines 466.468: How is this filtering exactly done? Please explain. How many data (in %) are remaining?

Line 481: "no significant trend was found ..." what does that mean? How do you interpret this finding?

Line 495: Write "WLG was increasingly affected by local sources  $\ldots$ " which ones are these?

Line 500: "... higher CH4 ... in past years"; which years, be more explicit.

Line 505: this statement is trivial as Zhou et al. used the same data, right?

Line 509: "... which could emit large amount of CH4 by human activities"; write "large amounts"; statement is very vague, can you be more specific?

Line 512: write "... may also contribute to the high CH4 ..."; statement is very vague, can you be more specific?

Chapter 4.1 misses quantitative statements; what's new compared to existing literature?

Line 530: "It was possibly due to  $\ldots$ "; very vague statement, can you be more conclusive?

Line 536: "the southwest or northwest region ... may be also strong source regions"; very vague statement.

**ACPD**
Line 540-542: awkward English; no conclusive statement.

Lines 544-545: add reference.

Line 556: "It is of great possible ...." awkward English

Line 573: Hateruma Island is at sea level, why do you compare WLG with this station?

Line 582: awkward English

Chapter 4.2 is very descriptive and lacks conclusions

Lines 592-597: move in front of line 591.

Line 602: start station names with upper case letters; why do compare with these stations, which have different characteristics.

Line 611: why is the photochemical capacity weak? I would expect a high photochemical activity.

Lines 630-631: numbers were already given in lines 454 ff.; repetition won't be needed if Results and Discussions are merged.

Lines 644-645: why do you compare with a Swiss site? You may compare with composite numbers, e.g. from the WDCGG data summary report (most recent version is #43); downloadable on the WDCGG webpage.

Lines 656-657: are the conclusions drawn for the Swiss site also true for WLG?

Lines 658-680: this paragraph may better fit into the introduction.

Lines 691-692: "suggested that there were possibly other strong CH4 sources ...": which ones? The following lines do not provide any answer.

Line 702: "due to the emission from the two largest source regions": be more specific.

Lines 704-705: I don't understand what wants to be said here.
Line 705: "anomalously" -> "anomalous"

Lines 706-707: how about trends in other regions in the world? This statement is the same as in the abstract and I still don't understand it. This is rather a statement that fits to a synthesis analysis as it is e.g. done in the annual WMO GHG bulletin.

Lines 722-723: "... the long-term verification is extremely important to ... understand CH4 variations ..."; did the understanding improve based on the present analysis? Which are the lessons-learnt?

Lines 725-726: "Tibetan Plateau was with the highest average altitude ...." awkward English.

Line 727: "anomalously" -> "anomalous"

Data availability statement: very strangely, this paragraph only refers to the data from the other stations but nothing is said how to access WLG data. The data used in the analysis doesn't seem to be freely available since WDCGG only contains daily and monthly CH4 averages from WLG; where can anybody access hourly CH4 data from WLG?

---

## Referee Comment (RC2) · Anonymous Referee #2 · 25 Aug 2020

General comments: In this manuscript, a 24-year long-term observation of atmospheric CH4 at Waliguan WMO/GAW global station in the Tibetan Plateau was studied. This report is very meaningful and the analysis of the paper is very comprehensive. The CH4 variations and its related potential causes during the long-term observation have been analyzed in details, which would help the scientific community to understand carbon cycle and formulate more informed carbon reduction policy. Given the importance and value of the long-term measurements of CH4 in the Tibetan Plateau plus that this manuscript is well drafted, I would recommend accepting this paper after minor modifications listed as below. Specific comments: 1) Line 98: add 'About' before 90%. 2) Line 233-238: How frequent is the backward trajectory computed? Hourly? Please

specify the numbers of trajectories are determined. 3) Some expression is not professional, e.g. "long-distance transport" (Lines 465), can be expressed by "long-range transport", and so on. Please polish the whole report. 4) There are several places in results and discussion where too many details are given, which make the text a little difficult to follow. Results and discussion are suggested to be merged. 5)Lines 685-686: Suggest to give more discussions about the larger growth rates in the Tibetan Plateau than that of city region. 6) Line 1057: delete 'CO'. 7) Why are the points not on the line in figure 8 and 9?

---

## Author Comment (AC2) · 19 Sep 2020

**Point by point response to the reviewer #2:**

First of all, we would like to thank the reviewers for the positive and constructive comments. The line number below is indicated based on the clean version. The followings are response to the questions of reviewer #2.

**Comments from the reviewer:**

**-Reviewer 2**

General comments: In this manuscript, a 24-year long-term observation of atmospheric CH4 at Waliguan WMO/GAW global station in the Tibetan Plateau was studied. This report is very meaningful and the analysis of the paper is very comprehensive. The CH4 variations and its related potential causes during the long-term observation have been analyzed in details, which would help the scientific community to understand carbon cycle and formulate more informed carbon reduction policy. Given the importance and value of the long-term measurements of CH4 in the Tibetan Plateau plus that this manuscript is well drafted, I would recommend accepting this paper after minor modifications listed as below.

**Response:** Thank you very much for your positive comments. We revised the manuscript and answered the questions point by point.

Specific comments:

Line 98: add 'About' before 90%.

**Response:** We have added it (Line 82).

Line 233-238: How frequent is the backward trajectory computed? Hourly? Please specify the numbers of trajectories are determined.

**Response**: Yes, it's hourly. We computed the 3-day back trajectories coincident with hourly  $CH_4$  mole fractions. The number of trajectories was determined by the valid hourly  $CH_4$  observations. We added the descriptions (Line 229-230). The related information, e.g. numbers, and average mole fractions, was listed in Table 3.

Some expression is not professional, e.g. "long-distance transport" (Lines 465), can be expressed by "long-range transport", and so on. Please polish the whole report.

**Response:** Thank you for your suggestion. We have revised all the descriptions in the manuscript.

There are several places in results and discussion where too many details are given, which make the text a little difficult to follow. Results and discussion are suggested to be merged.

**Response:** Thank you very much for your constructive suggestion. We merged 'Results' and 'Discussion' and rewrote the Chapter. The interpretation/conclusion was described in conjunction with results.

Lines 685-686: Suggest to give more discussions about the larger growth rates in the Tibetan Plateau than that of city region.

**Response:** We rewrote the Chapter 3.7. The interpretation/conclusion was described in conjunction with results. More discussions were added (Line 685-702).

"These results suggested that there were possibly other strong CH4 sources at the WLG that were not from cities and the southwest region (Northern India) was the most likely contributor. The PSCF analysis also supported this result (Fig .7). At present, Northern India and Eastern China were the two largest sources of CH4 at the WLG (Fig. S10) (Crippa et al., 2019). Since the Tibetan Plateau was coincidently trapped in the middle of them, the atmospheric CH4 at WLG was very likely dominated by long-range transport from these two regions. Although CH4 emissions increased slowly during 1994-2002, a negative trend appeared (Fig. S10), significantly increased emissions were found in both southeast and southwest Asia after 2007. Chen et al. (2013) illustrated that the warming (0.2 °C per decade) in the Tibetan Plateau resulted in substantial emissions of CH4 due to the thawed permafrost and melted glaciers. The rapid increase of CH4 would probably make it difficult to meet the goals of carbon emission reduction in the future. This would be especially true with the scenario of quickly increasing CH4 on the Qinghai-Tibetan Plateau due to the emissions from the two largest source regions of Northern India and Eastern China. The large growth rate of atmospheric CH4 in the TP revealed that i) the atmospheric CH4 at the WLG was not predominantly influenced by eastern cities in recent years and ii) large amounts of CH4 were transported from the Tibetan Plateau to WLG in recent years."

Line 1057: delete 'CO'.

**Response:** We are sorry for the mistake. The average CO mole fractions were added in Table 3.

Why are the points not on the line in figure 8 and 9?

**Response:** Thank you for your question. The lines are the smoothed curves of the points using the method of 'LOESS' Curve Fitting (Local Polynomial Regression), hence they do not just connect the points.

---

## Author Response (AR1)

**Point by point response to the reviewers:**

First of all, we would like to thank the reviewers for the positive and constructive comments. The line number below is indicated based on the clean version. The followings are response to the questions of reviewers.

**Comments from the reviewers:**

**-Reviewer #1**

General comments: The manuscript presents an analysis of 24 years of continuous in-situ ground-based methane observations at the remote Mount Waliguan station in China. Such records are very essential for monitoring atmospheric variability and trends of this important greenhouse gas. It is challenging to keep those observations up and running for decades. Thus, I fully support publication and release of these data. However, the analysis of the data is very descriptive and lacks conclusive statements. There is hardly any novel insight into the $CH_4$ cycle, which is currently drawn from the analysis. Several filters are applied to the data but it is difficult to follow how the filters are applied and which filtered dataset is used in which analysis. The paper may profit from merging chapter 3 (Results) and chapter 4 (Discussion). Moreover, the paper contains quite a number of linguistic flaws. Proofreading by an English native speaker is necessary. Some language mistakes are listed below. The paper is within the scope of 'Atmospheric Chemistry and Physics' but – to my mind – requires major revisions prior to its acceptance for ACP. See my specific comments below.

> **Response:** Thank you very much for your good comments. Based on the comments, our paper has improved a lot. We revised the manuscript and answered the questions point by point.
>
> Overall, i) according to your suggestion, we extended our records from 2017 to 2019, to increase the value and novelty of our study. We updated all our data and updated the respective part in our manuscript including the filtered time series (Table 1, Table 2, and Figure 2), the diurnal variations (Figure 3 and Table S1), the seasonal cycles (Figure 10, Figure 11, and Table S2), and the long-term trends (Table 4, Figure 12, and Figure S6). Please refer to the revised manuscript for details.

ii) Different data filtering methods were further described and the descriptions of which dataset is used were added. For the meteorological approach, we revised the descriptions as below (Line 197-204, Line 283-285).

"In this study, the $CH_4$ records associated with local surface winds from selected sectors, i.e. NNE-…-ENE in spring, NE-…-SE in summer, NE-…-ESE in autumn, and NE-ENE in winter, were flagged as locally influenced (27.0%). Subsequently, we rejected portion of daytime records to minimize the effect of human activities (16.9%), e.g. 9:00-13:00 LT (local time) in summer, and 10:00-17:00 LT in winter. Finally, we filtered $CH_4$ data into locally influenced when the surface wind speeds were less than 1.5 m s$^{-1}$ to minimize the very local accumulation (9.2%)."

"To precisely understand the characteristics of atmospheric $CH_4$, including seasonal cycles and long-term trends, it is vital to identify the $CH_4$ records in well-mixed air without local contaminations."

For the analysis between the city regions and the Tibetan plateau, the data filtering methods are described as below (Line 205-215).

"In order to investigate the influence of anthropogenic emissions from cities and remote area as Tibetan plateau, we divided the $CH_4$ data into two main influencing regions according to the analysis, including the geographical conditions, the effect of surface winds, the long-range transports, and the potential source distributions. The first region covers the northeast and southeast (NNE-…SE) of the WLG, which is denoted as City Regions (CR). The second region is located the south to west (S-…-W) of the station and is well known Tibetan (Qinghai-Xizang) Plateau (TP) (Fig. S1). Accordingly, the hourly $CH_4$ records when the surface winds coming from these sectors were divided into two subsets (i.e. TP and CR). The long-term variations between the two regions as well as the total regional time series were further compared and analyzed."

For the analysis of the long-range transport of emissions from cities, the data filtering methods are described as below (Line 665-674).

"As described above, the northeast and southeast city regions might have acted as strong regional sources influencing the atmospheric $CH_4$ at the WLG. Therefore, to analyze the effect of long-range transport of emissions from cities, the regionally representative data was further excluded by air mass transport, and the remaining regional records were denoted as 'TR'. First, the monthly cluster analysis was applied to hourly trajectories over 2005-2007, 2008-2012 and 2013-2017. Then, based on the cluster analysis, the clusters were divided into two groups, i.e. from city regions (red clusters in Fig. S8), and other (black clusters in Fig. S8). Finally, the regionally representative data were accordingly classified as two groups based on the cluster results (cities or other). The statistical results were presented in detail in Figure S8 and Table S3."

For other analysis, i.e. diurnal variation, local surface wind, air mass pathways, and potential source distributions, we used total hourly $CH_4$ data.

iii) We merged chapter 3 (Results) and chapter 4 (Discussion). The interpretation/conclusion was described in conjunction with results, and more discussions were added. Based on the long-term measurements, we concluded the main findings compared to the existing studies (Line 27-34). Generally, the characteristics of $CH_4$ varied in different observing periods: i) the diurnal cycle has been becoming apparent and the amplitudes of the diurnal or seasonal cycles increased over time, ii) the wind sectors with elevated $CH_4$ mole fractions switched from ENE-…-SSE sectors in early periods to NNE-…-E sectors in later years, iii) the area of source regions increased as the years progressed and strong sources shifted from northeast (city regions) to southwest (Northern India), iv) the annual growth rates in recent years (e.g. 2008-2019) were significantly larger than that in early periods (e.g. 1994-2007).

iv) We have asked the language company to help the English improvement.

Specific comments:

Abstract is rather long, you may consider shorten it.

**Response:** Thank you for your suggestion. We have shortened the abstract (Line

18-34) as below.

"A 26-year long-term record of atmospheric methane ($CH_4$) measured in-situ at the Mount Waliguan (WLG) station, the only WMO/GAW global station in inland Eurasia, is presented. Overall, a nearly continuous increase of atmospheric $CH_4$ was observed at the WLG with a yearly growth rate of $5.1 \pm 0.1$ ppb yr$^{-1}$ during 1994-2019, except for some particular periods with near-zero or negative values, e.g. 1999-2000, and 2004-2006. The average $CH_4$ mole fraction was only $1799.0 \pm 0.4$ ppb in 1994 but increased about 133 ppb and reached a historic level of $1932.0 \pm 0.1$ ppb in 2019. The case study in the Tibetan Plateau showed that the atmospheric $CH_4$ increased rapidly. During some special period, it is even larger than that of city regions (e.g. $6.7 \pm 0.2$ ppb yr$^{-1}$ in 2003-2007). Generally, the characteristics of $CH_4$ varied in different observing periods: i) the diurnal cycle has been becoming apparent and the amplitudes of the diurnal or seasonal cycles increased over time, ii) the wind sectors with elevated $CH_4$ mole fractions switched from ENE-…-SSE sectors in early periods to NNE-…-E sectors in later years, iii) the area of source regions increased as the years progressed and strong sources shifted from northeast (city regions) to southwest (Northern India), iv) the annual growth rates in recent years (e.g. 2008-2019) were significantly larger than that in early periods (e.g. 1994-2007)."

Lines 18-19: write "A 24-year long-term record of atmospheric methane ($CH_4$) measured in-situ at the Mt. Waliguan station, the only …, is presented."

**Response:** Revised the sentence (Line 18-19) as you suggested.

Line 20: "… 1994-2017 …": why don't you use more recent data (e.g. until end of 2019)?

**Response:** Thank you for your question. According to your suggestion, we extended our records from 2017 to 2019, to increase the value of our study. We updated all our data and updated the respective part in the manuscript including the filtered time series (Table 1, Table 2, and Figure 2), diurnal variations (Figure 3

and Table S1), seasonal cycles (Figure 10, Figure 11, and Table S2), and long-term trends (Table 4, Figure 12, and Figure S6). Please refer to the revised manuscript for details.

Line 20: "continuously" need to be "continuous"

**Response:** We have corrected it (Line 20).

Lines 20-22: "continuous increase" but "even negative growth trend … in particular periods" is a contradiction.

**Response:** Thank you for the correction. We rewrote it as "a nearly continuous increase" in the revised draft (Line 20).

Lines 24-25: "but unprecendeted elevated ~100 ppb" sounds awkward

**Response:** We updated the data to 2019 and changed it to "but increased about 133 ppb" (Line 24).

Line 24: "… historic high of 1903.8 $\pm$ 0.1 ppb in 2016…"; what about 2017?

**Response:** We have updated it to 2019. "… reached a historic level of $1932.0 \pm 0.1$ ppb in 2019" (Line 24).

Line 26: what does "$\triangle CO/\triangle CH_4$" mean? Why do you look at CO to $CH_4$ ratios?

**Response:** Thank you for your question. $\Delta CH_4$ and $\Delta CO$ are the detrended time series of $CH_4$ and $CO$ based on the method of Thoning et al. (1989). These are results of the original data minus the trend curve. $\Delta CO/\Delta CH_4$ is the ratio of the $\Delta CO$ and $\Delta CH_4$. Because part of the $CH_4$ and $CO$ in atmosphere are from the same anthropogenic sources (e.g. fossil fuel combustion), the long-term trend of $\Delta CO/\Delta CH_4$ is helpful to understand the variation of the sources/sinks. We added the descriptions about $\Delta CH_4$ and $\Delta CO$ (Line 263-266, Line 464-467) as below.

"The detrended values are denoted as $\Delta CH_4$ and $\Delta CO$, which are the original data points minus the trend curve. To accurately obtain the correlation slopes of

ΔCO and ΔCH$_4$, i.e. ΔCO/ΔCH$_4$, a rolling linear regression was applied to the ΔCH$_4$ and ΔCO time series by the 'roll_lm' function in 'roll' package of R"

"Because parts of the CH$_4$ and CO in atmosphere were from the same anthropogenic sources (e.g. fossil fuel combustion), the long-term trend of ΔCO/ΔCH$_4$ was helpful to understand the variation in the sources/sinks in many studies (Buchholz et al., 2016; Niwa et al., 2014; Tohjima et al., 2014; Wada et al., 2011)."

Line 27: add "elevated" that is reads "… opposite to other elevated sites …"

**Response:** We have revised the abstract and deleted the sentence.

Line 30: "(the Northern India)" -> "(Northern India)"

**Response:** We revised it (Line 32).

Line 35: delete "What is interesting" I do not believe that you want to mention uninteresting things in the abstract of your manuscript.

**Response:** We deleted it (Line 27).

Line 47: "anomalously" -> "anomalous"

**Response:** We rewrote the abstract and deleted the related sentence.

Lines 47-49: If this statement is presented prominently in the abstract, it also needs to be discussed in the manuscript. Moreover, when looking at Fig. 11 (lower panel), I do not really see an increasing growth rate (but only a mostly positive growth rate).

**Response:** Thank you for your comment. Yes, the description is inappropriate here, it should be "larger growth rate" and this is rather a statement that fits to a synthesis analysis. We rewrote the abstract and deleted the related descriptions.

The mostly positive values will lead to larger average growth rate, but the increasing growth rate is not distinct in Fig. 11. We listed the periodic growth rates in Table 4 as below. The growth rates during 2008-2019 are larger than that during

1994-2002.

**Table 4.** Annual growth rates of atmospheric $CH_4$ in the City Regions (CR), the Tibetan Plateau (TP), and total regional records during 1994-2019 at the WLG station.

|  | 1994-1997 | 1998-2002 | 2003-2007 | 2008-2012 | 2013-2019 | 1994-2019 |
|---|---|---|---|---|---|---|
| CR | $3.0 \pm 0.1$ | $3.6 \pm 0.2$ | $5.3 \pm 0.2$ | $7.0 \pm 0.2$ | $6.2 \pm 0.1$ | $5.2 \pm 0.1$ |
| TP | $3.4 \pm 0.1$ | $3.0 \pm 0.2$ | $6.7 \pm 0.2$ | $5.7 \pm 0.2$ | $5.7 \pm 0.1$ | $5.1 \pm 0.1$ |
| **Total** | **$4.9 \pm 0.1$** | **$2.5 \pm 0.2$** | $4.9 \pm 0.1$ | **$7.7 \pm 0.1$** | **$5.5 \pm 0.1$** | $5.1 \pm 0.1$ |

Line 60: write "$CH_4$ has an 8-12 years lifetime …"

**Response:** We have revised it (Line 45).

Line 64: "… $CH_4$ rapidly increased …"

**Response:** We have revised it (Line 49).

Lines 65-66: write "Results from ice core analyses in Antarctica showed …"

**Response:** We revised the sentence (Line 50-51).

Line 67: write "… has reached a level unprecedented over …"

**Response:** We have revised the sentence (Line 52).

Line 68: awkward English

**Response:** We have revised the sentence (Line 53-55) as below.

"At the beginning of the 1990s, the $CH_4$ mole fraction showed a decreasing trend in. Consequently, the reverse trend has been observed since 1998 due to the higher global mean temperature (Dlugokencky et al., 1998; Nisbet et al., 2014)."

Line 71: delete "special"

**Response:** We have deleted it.

Line 73: write "… (Nisbet, et al., 2019) followed by a renewed $CH_4$ increase since then".

**Response:** We revised the sentence (Line 58).

Line 78: don't start a sentence with "And …"

**Response:** We have deleted it.

Lines 78-79: write "unexpected" instead of "not expected"; why the increase is "unexpected"?

**Response:** We have deleted the related sentence because it is inappropriate here (Line 62). Because the growth rate of $CH_4$ is originally very low and lasts a long time before 2007, but increases strongly and continuously from 2007 to now. Therefore, we used the 'unexpected increase'.

Line 84: explain "C.E."

**Response:** We added "Common Era" (Line 66).

Line 85: write "Atmospheric $CH_4$ is mainly …"

**Response:** We have revised it (Line 68).

Lines 105-106: write "Systematic observations are a prerequisite to get an accurate understanding of spatial and temporal behavior of atmospheric $CH_4$ concentrations."

**Response:** We revised the sentence (Line 94-95).

Line 113: awkward English

**Response:** We revised the sentence (Line 101-107) as below.

"Recently, other stations have been installed for $CH_4$ observation, such as the Barrow (BRW), South Pole (SPO) (polar site), Cape Grim (CGO) and Minamitorishima (MNM) (coastal/island sites), Jungfraujoch (JFJ) and Mount Waliguan (continental mountain site). Hundreds of $CH_4$ observation stations worldwide are currently running under the framework of the WMO/GAW."

Lines 114-118: There are many more stations continuously measuring $CH_4$ levels in the atmosphere; why did you select these ones?

> **Response:** Thank you for your question. We select these stations because i) they just represent different types of global stations, ii) parts of the stations have similar altitude or latitude to Waliguan, e.g. MLO, JFJ, and MNM.

Lines 118-127: this is largely a repetition, merge with paragraph on the previous page (lines 85-104).

> **Response:** We have moved and merged the sentences (Line 87-93).

Lines 146-147: write "… which is the longest record in China."

> **Response:** We have revised it (Line 126).

Lines 157: write "WLG is the only …"

> **Response:** We have revised it (Line 136).

Line 158: write "… and is run by the China Meteorological Administration …"

> **Response:** We have revised it (Line 137).

Line 166: "Tibetan Plateau"

> **Response:** We have revised it (Line 145).

Line 167: "dial variations … are influenced …"

> **Response:** We have revised it (Line 146).

Line 221: write "… were flagged as locally influenced."; if the data are locally influenced they are poorly representative.

> **Response:** Thank you for your comments. We have revised it (Line 200). Yes, the data are locally influenced means they are poorly representative. They represent the $CH_4$ that are strongly affected by local contaminations or not in well-mixed air. We excluded them to get the regionally representative data. Finally, based on the meteorological method, about 64% of the CH$_4$ data was classified as regionally representative over 1994-2019 at the WLG (Line 198-204, Table 1).

Lines 220-225: add percentages of rejection according to the individual filters.

**Response:** We have calculated and added the percentage of each filter (Line 198-204). Overall, about 27.0% by wind sectors, 16.9% from daytime records, and 9.2% by wind speed.

Line 224: "We filtered CH$_4$ data into local events …" sounds strange, how can you filter data into events?

**Response:** We have changed "local events" to "locally influenced". The locally influenced data cannot represent the CH$_4$ in well-mixed air.

Line 226: write "hourly CH$_4$ data was binned into 16 horizontal wind direction classes …"; is this done for all data or the regionally representative data only?

**Response:** We have revised the sentence (Line 216-217). This is done for all data.

Lines 235-236: elaborate on HYSPLIT: what's the spatial resolution of the model, what is the height above sea level of WLG in the model?

**Response:** The spatial resolution of the model is 0.5×0.5 degree and the height is 10 km a.s.l. We added the descriptions in the paper (Line 229-230).

"The spatial resolution of the model is 0.5×0.5 degree and the model height is 10 km a.s.l."

Line 239: write "The trajectories for January, April, …"

**Response:** We have revised the sentence (Line 231).

Line 285: "appropriately every five years a period" sounds strange.

**Response:** We have revised the sentence (Line 274-278) as bellow.

"The entire CH$_4$ time series were divided into five observing periods, i.e. 1994-

1997, 1998-2002, 2003-2007, 2008-2012, and 2013-2019, according to the significant stages or the critical time period of atmospheric $CH_4$ variations from previous studies."

Results and Discussion chapters (Chapters 3 and 4): I strongly encourage the authors to merge these two chapters. Results should be discussed in stronger conjunction with existing literature, preferably from Asian sites; I don't see the rationale why some findings are compared with conclusions from non-elevated sites in Europe.

**Response:** Thank you very much for your constructive suggestions. We have merged the section of 'Results' and 'Discussion'.

We added more discussions with existing long-term studies from Asian sites, e.g. Yonagunijima (YON) and Ryori (RYO) in Japan, Sinhagad (SNG) and Cape Rama station (CRI) over India, Ulaan Uul (UUM) in Mongolia, and Tae-ahn Peninsula (TAP) in Korea. Some added descriptions are as below.

"Tohjima et al. (2014) found an opposite variation at the Hateruma Island, which showed small slope values in the summer. Wada et al. (2011) analyzed more than 10-year seasonal variation of the $\Delta CO/\Delta CH_4$ ratios at three monitoring stations, i.e. MNM, Yonagunijima (YON), and Ryori (RYO) in Japan, which also showed an opposite trend to that of the WLG. It was because these sites were considerably affected by the Asian continental source regions, where had enhanced emissions of $CH_4$ in the summer (rice paddies) and CO in the winter (fuels combustion)." (Line 474-480)

"For other regional sites in the Asia, Guha et al. (2018) studied seasonal variability at the Sinhagad (SNG) and Cape Rama station (CRI) over India, which also showed an opposite trend to the WLG due to the strong impact of monsoon dynamics. Ahmed et al. (2015) found that the seasonal $CH_4$ showed a maximum in the winter and a minimum in the spring at two urban sites of Guro (GR) and Nowon (NW), in Seoul, Korea over 2004-2013. Kim et al. (2015) investigated the decadal variation (1991-2013) of $CH_4$ at the East Asian sites, e.g. Ulaan Uul (UUM) in Mongolia and Tae-ahn Peninsula (TAP) in Korea, which revealed again an opposite seasonal trend to that of the WLG." (Line 521-529)

"The seasonal amplitude at the WLG (~14 ppb) was significantly lower than many other sites in the Northern Hemisphere, by about 35-70 ppb. Such sites included MLO in America, BRW in North Pole, UUM in Mongolia, TAP in Korea, Ny-Ålesund in Norway, Bialystok in Poland, Ochsenkopf in Germany, and Beromunster in Switzerland (Dlugokencky et al., 1995; Kim et al., 2015; Morimoto et al., 2017; Thompson et al., 2009; Popa et al., 2010; Satar et al., 2016). MBL also showed a larger amplitude than WLG (Fig. 11). The study at the SNG and CRI over India showed a much larger amplitude close to 200 ppb (Guha et al., 2018)." (Line 546-553)

"Tohjima et al. (2002) found that the $CH_4$ levels at the Cape Ochi-ishi and Hateruma Island in1995-2000 respectively increased by 4.5 and 4.7 ppb $yr^{-1}$, which were also similar to that of the WLG. Tsutsumi et al. (2006) analyzed the trend of hourly $CH_4$ data from 1998 to 2004 on the YON, which showed a similar increase (~3.0 ppb $yr^{-1}$) to the WLG. The study at the GR and NW in Seoul, Korea, presented almost an identical trend of 2 ppb $yr^{-1}$ between 2004 and 2013 (Ahmed et al., 2015), which was lower than that of the WLG in similar period." (Line 586-592)

"The growth rate of $CH_4$ observed at the Ny-Ålesund, Svalbard, increased from $0.3 \pm 0.2$ ppb $yr^{-1}$ during 2000-2005 to $5.5 \pm 0.2$ ppb $yr^{-1}$ during 2005-2014, which had a similar variation but with a little lower growth rates than that of the WLG (Morimoto et al., 2017). The study suggested that the temporal pause in 2000-2005 was ascribed to the reductions of $CH_4$ emissions from the microbial and fossil fuel sectors, while the increase in 2005-2014 was due to an increase in microbial release." (Line 598-604)

Additionally, we also compared the result with adjacent stations in China, e.g. Shangdianzi, Lin'an. For other sites with similar latitude or altitude in the Northern Hemisphere, e.g. Mauna Loa and Jungfraujoch, are also compared.

Lines 297-298: awkward English

**Response:** We have revised the sentence (Line 308-309) as bellow.

"In winter, a large increase of CH$_4$ was found during 9:00-17:00 LT, with the largest peak to trough amplitude of 7.1 ± 2.9 ppb."

End of Chapter 3.1: interpretation/conclusion missing.

**Response:** We merged the section of 'Results' and 'Discussion' and rewrote the Chapter 3.1 (Chapter 3.2 now). The interpretation/conclusion was described in conjunction with results (Line 310-333). Please refer to the revised manuscript.

Line 315: "As observed by the previous short-term observations"; I don't understand this statement.

**Response:** We have changed to "Similar to the previous studies…" (Line 319).

Line 326: Delete "What interesting is"

**Response:** We have deleted it.

Line 345: Write "It's obvious that CO showed …"

**Response:** We have revised it (Line 391).

Line 348: write "when percentages ranged from 0 to 40."; write "When data exceeded the 60% percentile, the high area probability areas …"

**Response:** We have revised the sentence (Line 394-395).

End of Chapter 3.2: interpretation/conclusion missing.

**Response:** We merged the section of 'Results' and 'Discussion' and rewrote the Chapter 3.2 (Chapter 3.3 now). The interpretation/conclusion was described in conjunction with results (Line 345-365, Line 374-375, Line 383-391, and Line 395-399). Please refer to the revised manuscript.

Line 365, 368-369: awkward English

**Response:** We have revised the sentence (Line 414-419) as below.

"Cluster 3 showed the highest $CH_4$ mole fraction, with an enhancement of ~4 ppb relative to the seasonal average. In winter, the air masses primarily came from northwest and southwest regions, e.g. cluster 3 (59%), and cluster 1 (34%) (Fig. 6d) and the cluster 1 brought the highest $CH_4$ mole fractions with an enhancement of ~7 ppb over the seasonal average."

End of Chapter 3.3: interpretation/conclusion missing.

**Response:** We merged the section of 'Results' and 'Discussion', and rewrote the Chapter 3.3. The interpretation/conclusion was described in conjunction with results (Line 420-433, Line 451-461).

Line 389: write "regionally representative"

**Response:** We have revised all the descriptions in the manuscript.

Line 390: write ""locally influenced"

**Response:** We have revised all the descriptions in the manuscript.

Line 391: "data was … larger than … events"; awkward English

**Response:** We have revised the sentence (Line 288-289) as below.

"The average of the locally influenced data (1868.2 ± 0.3 ppb) was larger than that of the regionally representative records (Table 1)."

Line 394: Does it hold true for all data or regionally representative data?

**Response:** It hold true for both of them. The average of regionally representative data and all data both showed increasing trend over 1994-2019.

Line 396 (reference to Fig. 7): I suggest to move Fig. 7 above and make it to Fig. 2; show first the whole dataset before you analysis of the data

**Response:** Thank you very much for your suggestion. We changed Fig. 7 to Fig. 2 and moved Chapter 3.4 above as Chapter 3.1.

Line 402: what is deltaCO and deltaCH$_4$? I assume it is excess CO and excess CH$_4$, i.e. data above baseline (looks like when looking at Fig. S1); if so, how was the background determined? Please elaborate; show also the CO time series.

**Response:** Thank you for your questions. Similar to the question you posed on L26, $\Delta$CH$_4$ and $\Delta$CO are the detrended time series of CH$_4$ and CO from 2004-2017 based on the method of Thoning et al. (1989).

The detrended value is the original data points minus the trend curve. The trend value is the polynomial part of the function plus the long-term filter of the residuals. We have added the descriptions in the manuscript (Line 261-266).

The $\Delta$CH$_4$ and $\Delta$CO time series are showed in Figure S4. The original hourly CO time series was also added in the paper (Figure S3).

[Figure]

**Figure. S3.** The hourly CO data from 2004 to 2017 at the WLG station.

Line 411: incomplete sentence.

**Response:** We deleted the sentence because there is a repetition of above descriptions.

End of Chapter 3.5: what do we learn from the different deltaCO to deltaCH$_4$ ratios?

**Response:** The strong seasonal variation of $\Delta$CO/$\Delta$CH$_4$ also revealed that the WLG was affected by different anthropogenic sources, e.g. sources from cities, and sources from the Tibetan Plateau, during a year, especially in the summer and winter. The long-term trend of the slopes implied that the source emission types (CO sources or $CH_4$ sources) around the WLG might have been changing with human activities, like straw burning, in the early years, or coal mining in recent years. We revised the Chapter 3.5 and added the descriptions (Line 498-504). Please refer to the revised manuscript.

Lines 415-419: this is a different filtering than the one described above, right? Move these lines to Chapter 2.3.

**Response:** Yes, this is a different data filtering method. We are using this method to further investigate the influence of anthropogenic emissions from cities and remote area as Tibetan plateau. We moved the paragraph to Chapter 2.3 (Line 205-215).

Line 426: write "… the seasonal averages of regionally representative $CH_4$ were …"; do these numbers relate to regionally representative data?

**Response:** We have revised it. Yes, the seasonal averages are calculated based on regionally representative data.

End of Chapter 3.6.1: interpretation/conclusion missing.

**Response:** We merged the section of 'Results' and 'Discussion' and rewrote the Chapter 3.6.1. The interpretation/conclusion was described in conjunction with results (see Chapter 3.6.1, Line 518-562).

Line 442: write "… growth rates were very small or even negative …"

**Response:** We have revised it (Line 567).

Lines 466.468: How is this filtering exactly done? Please explain. How many data (in %) are remaining?

**Response:** The regionally representative data are further filtered based on the hourly trajectories of cluster analysis.

First, the monthly cluster analysis was applied to hourly trajectories over 2005-2007, 2008-2012 and 2013-2017. Then, based on the cluster analysis, the clusters were divided into two groups, i.e. from city regions (red clusters in Fig. S8), and other (black clusters in Fig. S8). Finally, the regionally representative data were accordingly classified as two groups based on the cluster results (cities or other). We added the descriptions in the manuscript (Line 669-674).

Figure S8, Table S3, and Table 5 showed the filtered results in detail. Eventually, 67.1% of regionally representative data are remaining.

[Figure]

**Figure S8.** The monthly cluster analysis of hourly trajectories during 2005-2017 at the WLG station. The red clusters represent air masses transport from city regions.

**Table S3.** The statistics of the excluding trajectories over different observing periods during 2004-2017 at the WLG station.

|  |  | Jan | Feb | Mar | Apr | May | Jun | Jul | Aug | Sep | Oct | Nov | Dec | Sum |
|---|---|---|---|---|---|---|---|---|---|---|---|---|---|---|
| 2005-2007 | Total | 764 | 1193 | 938 | 1190 | 882 | 1139 | 921 | 1431 | 1427 | 843 | 1272 | 911 | 12911 |
|  | Reject | 157 | 111 | 51 | 371 | 545 | 767 | 632 | 1325 | 911 | 181 | 133 | 19 | 5203 |
| 2008-2012 | Total | 3060 | 2081 | 1764 | 1811 | 1136 | 2250 | 1919 | 2826 | 2791 | 1527 | 1868 | 1079 | 24112 |
|  | Reject | 41 | 228 | 194 | 467 | 675 | 1681 | 1428 | 1727 | 1298 | 90 | 0 | 0 | 7829 |
| 2013-2017 | Total | 2110 | 1738 | 2112 | 1882 | 1269 | 1259 | 461 | 1816 | 2186 | 1659 | 1844 | 1111 | 19447 |
|  | Reject | 0 | 1 | 0 | 55 | 81 | 0 | 90 | 875 | 138 | 71 | 0 | 12 | 1323 |

**Table 5.** The statistics of CH$_4$ data without air mass transport from city regions (TR) over different periods during 2005-2016 at WLG station.

|  | Transport regions | Hours | Percentage (%) | Average (ppb) | Updated growth rate (ppb yr-1) |
|---|---|---|---|---|---|
| 2005-2007 | TR | 6922 | 77.2 | 1824.9 ± 0.2 | 2.7 ± 0.2 |
|  | City | 2041 | 22.8 | 1835.9 ± 0.5 | - |

| 2008-2012 | TR | 7060 | 64.4 | 1853.7 ± 0.2 | 10.1 ± 0.1 |
|---|---|---|---|---|---|
| | City | 4254 | 35.6 | 1861.0 ± 0.3 | - |
| 2013-2016 | TR | 4152 | 61.6 | 1888.2 ± 0.3 | 6.3 ± 0.1 |
| | City | 2591 | 38.4 | 1888.5 ± 0.5 | - |
| **2005-2016** | TR | 18134 | **67.1** | 1850.6 ± 0.2 | 7.0 ± 0.1 |
| | City | 8886 | 32.9 | 1863.2 ± 0.3 | - |

Line 481: "no significant trend was found …" what does that mean? How do you interpret this finding?

**Response:** It means that the $CH_4$ from the Tibetan Plateau has similar growth rates to that from city regions. We revised the description (Line 581-582) as below. "Similar growth rates were found between the CR and the TP during 1994-2019 (Fig. S6)."

The finding implied that i) the atmospheric $CH_4$ at the WLG was not predominantly influenced by eastern cities in recent years and ii) large amounts of $CH_4$ were transported from the Tibetan Plateau to WLG in recent years (Line 699-702). Through the analysis of spatial distribution of emissions (Fig. 7, Fig. S10), the emission from southwest to south regions, i.e. Northern India, increased strongly after 2007, which may have contributed a lot to the high growth rates in the Tibetan Plateau in recent years.

Line 495: Write "WLG was increasingly affected by local sources …" which ones are these?

**Response:** We revised the sentence (Line 331-333).

"The increasing amplitude and $CH_4$ mole fractions suggested that the WLG was increasingly affected by local and regional anthropogenic sources, such as gas exploitation, and grazing."

Line 500: "… higher $CH_4$ … in past years"; which years, be more explicit.

**Response:** We revised the descriptions, "In 1994-2002, the $CH_4$ mole fractions …" (Line 310).

Line 505: this statement is trivial as Zhou et al. used the same data, right?

**Response:** Zhou et al. (2004) used the data from 1991 to 2002. However, we are using much longer observation records than them. The compared period is also not absolutely the same. Through the 26-year measurements, we found different results, which provided a broader view of changing characteristics of $CH_4$ to understand $CH_4$ cycle.

In this chapter 3.3, compared to Zhou et al. (2004), we found that wind sectors with elevated $CH_4$ mole fractions changed over time (Line 366-375). The elevated $CH_4$ was predominately from the ENE-..-SSE sectors in the early years (Fig. 4a-b) but evolved to the NNE-NE-ENE-E sectors in later years (Fig. 4c-e).

Line 509: "… which could emit large amount of $CH_4$ by human activities"; write "large amounts"; statement is very vague, can you be more specific?

**Response:** We have revised the sentence as below (Line 353-359).

"The two largest cities of Xining (with population of ~2.2 million) and Lanzhou (with population of ~4 million) are also situated in the northeast and east of the WLG, respectively. The heavy human activities from anthropogenic fossil combustion, landfills, and livestock, could have also emitted large amounts of $CH_4$. Based on the data from the Emissions Database for Global Atmospheric Research (EDGAR), the increase of $CH_4$ emission was 500 kg yr$^{-1}$ in these two regions throughout 1994-2015 (Crippa et al., 2019)."

Line 512: write "… may also contribute to the high $CH_4$ …"; statement is very vague, can you be more specific?

**Response:** We have revised the sentence as below (Line 359-365).

"Also, the Yellow River Canyon industrial area (YRC), which is ~500 km northeast of the WLG, may also have contributed to the high $CH_4$ values (Zhou et al., 2003). With the rapid development of land-use, water utilization, and agriculture sources in the YRC, large $CH_4$ emissions could have easily transported to the WLG. Previous studies on black carbon (BC) and carbon monoxide (CO)

also revealed that the high CH$_4$ values at the WLG in winter were a result of transport from the YRC (Tang et al., 1999; Zhou et al., 2003)."

Chapter 4.1 misses quantitative statements; what's new compared to existing literature?

**Response:** We merged Chapter 4.1 with the 'Results' and added more descriptions (see Chapter 3.2 and 3.3). The interpretation/conclusion was added in conjunction with results.

Compared to previous studies, we found that i) the diurnal variations were ambiguous before 2002, but significant diurnal cycles appeared afterward, ii) the highest CH$_4$ mole fraction was found in winter over 1994-2002, but in summer during the periods of 2003-2007, 2008-2012, and 2013-2019, iii) the wind sectors with high CH$_4$ mole fractions changed and concentrated on ESE-ENE sectors, and iv) the amplitude of enhancements was increasing.

Line 530: "It was possibly due to …"; very vague statement, can you be more conclusive?

**Response:** We have revised the statement (Line 424-433).

"The higher values seen in the northwest to southwest airflow were because the air masses had passed through the northwest of the Qinghai province and the central area of the Xinjiang Uygur Autonomous Region (XUAR). This is where the Ge'ermu urban area (the second largest city of Qinghai) was located and where there was with rapid industrial development, natural gas and petroleum resource exploitation, and residue burning of large crops, hence the CH$_4$ emissions were strong (Fang et al., 2013; Zhang et al., 2013). This result was also similar to the CPF percentile analysis (Fig. 5), as time went by, the southwest or northwest regions that were farther away from the site became the strongest source regions to the WLG."

Line 536: "the southwest or northwest region … may be also strong source regions"; very vague statement.

**Response:** We have revised it as "This result was also similar to the CPF percentile analysis (Fig. 5), as time went by, the southwest or northwest regions that were farther away from the site became the strongest source regions to the WLG." (Line 431-433).

Line 540-542: awkward English; no conclusive statement.

**Response:** We have revised the sentence (Line 451-453) as below.

"More $CH_4$ sources appeared at the WLG along with the progression of time, which could have been attributed to the influence of human expansion."

Lines 544-545: add reference.

**Response:** We added the reference (Fu et al, 2012) (Line 457).

Line 556: "It is of great possible …" awkward English

**Response:** We have deleted it.

Line 573: Hateruma Island is at sea level, why do you compare WLG with this station?

**Response:** We compared WLG with Hateruma Island (Tohjima et al, 2014), because the studies they reported about the long-term slope of $\Delta CO/\Delta CH_4$, and we used similar method by Tohjima et al. (2014) to calculate $\Delta CO/\Delta CH_4$. In addition, as an island station, it could well capture the $CH_4$ signal that is unaffected by human activities.

Line 582: awkward English

**Response:** We have merged Chapter 4.2 with 'Results', the statement is a repetition and we deleted it.

Chapter 4.2 is very descriptive and lacks conclusions

**Response:** We merged the section of 'Results' and 'Discussion' and added interpretation/conclusion in conjunction with result (see Chapter 3.4.1 and 3.4.2).

Lines 592-597: move in front of line 591.

**Response:** We have moved the sentence (Line 530-535).

Line 602: start station names with upper case letters; why do compare with these stations, which have different characteristics.

**Response:** We revised them (Line 546-550). The seasonal amplitude at WLG was lower than many other sites in the Northern Hemisphere. We compared these stations because they are typical stations with large seasonal amplitude. We also added more discussions to compare the study with Asian sites.

"The seasonal amplitude at the WLG (~14 ppb) was significantly lower than many other sites in the Northern Hemisphere, by about 35-70 ppb. Such sites included MLO in America, BRW in North Pole, UUM in Mongolia, TAP in Korea, Ny-Ålesund in Norway, Bialystok in Poland, Ochsenkopf in Germany, and Beromunster in Switzerland (Dlugokencky et al., 1995; Kim et al., 2015; Morimoto et al., 2017; Thompson et al., 2009; Popa et al., 2010; Satar et al., 2016). MBL also showed a larger amplitude than WLG (Fig. 11). The study at the SNG and CRI over India showed a much larger amplitude close to 200 ppb (Guha et al., 2018)."

In order to fully discuss the characteristic of $CH_4$ at WLG, based on the existing studies, we compared WLG with both adjacent stations around China and other stations with similar altitude or latitude in the Northern Hemisphere, e.g. Asian sites, European sites, and American site.

Line 611: why is the photochemical capacity weak? I would expect a high photochemical activity.

**Response:** Yes, the photochemical capacity is high in summer. We have revised the statement (Line 539-542). But as a plateau station with lower VOC and $O_3$, the oxidizing capacity at WLG is far below the lower altitude area.

Lines 630-631: numbers were already given in lines 454 ff.; repetition won't be needed if Results and Discussions are merged.

**Response:** Thank you for your good suggestion. We have merged 'Results' and 'Discussion'.

Lines 644-645: why do you compare with a Swiss site? You may compare with composite numbers, e.g. from the WDCGG data summary report (most recent version is #43); downloadable on the WDCGG webpage.

**Response:** Thank you for your comments. We have compared the data from WDCGG sum43 as well as WMO GHG Bulletin, e.g. (Nisbet et al., 2016; 2019), (WMO, 2019; 2020) (Line 605-613). The comparisons of other sites with similar altitude or latitude are also listed, aimed to fully discuss the long-term $CH_4$ growth in the world. We also added more discussions about the Asian sites (Line 586-592 and 598-604).

Lines 656-657: are the conclusions drawn for the Swiss site also true for WLG?

**Response:** To our knowledge, until now, the reasons of the anomalous spikes or strong growth of atmospheric $CH_4$ have not yet been determined. The conclusion raised by the study at Beromünster, Swiss is also a possible reason. However, we are trying to provide more information to figure the cause for these spikes.

We analyzed the long-term variation between city regions and Tibetan Plateau, the $CH_4$ emission from different sectors in China, and source changes over different periods. We also discussed the potential reasons provided by other studies (Line 631-645). We concluded that i) the emission from solid fuel (e.g. coal) and rice cultivation may contribute to the anomalous increase (Line 649-653), ii) large emissions from Northern India in recent years may have contributed to the anomalous increase at the WLG (455-461), iii) the warming in the Tibetan Plateau is also an important factor (Line 693-695).

Lines 658-680: this paragraph may better fit into the introduction.

**Response:** Thank you for your comments. The paragraph describes the $CH_4$

emissions based on the data from EDGAR. It could fit into the introduction. But based on previous studies, we already generally described $CH_4$ sources using one paragraph in the introduction (69-93).

This paragraph particularly describes the $CH_4$ emissions from different sectors during 1995-2015 in China, which showed a similar period to our study. The data could be used to discuss the potential reasons for $CH_4$ growth at the WLG. Hence, we merged the paragraph with 'Results' (Chapter 3.6.2).

Lines 691-692: "suggested that there were possibly other strong $CH_4$ sources …": which ones? The following lines do not provide any answer.

**Response:** Based on our study, it's the strong sources from Northern India. India with abundant cattle as well as an extensive large-scale coal mining operation possibly contributed large amounts of $CH_4$ to move from northern India to the northeastern Tibetan Plateau (Fig. 7i & l). We have revised the sentence as below (Line 685-687).

"These results suggested that there were possibly other strong $CH_4$ sources at the WLG that were not from cities and the southwest region (Northern India) was the most likely contributor."

Line 702: "due to the emission from the two largest source regions": be more specific.

**Response:** We have revised the sentence as below (Line 697-699).

"This would be especially true with the scenario of quickly increasing $CH_4$ on the Qinghai-Tibetan Plateau due to the emissions from the two largest source regions of Northern India and Eastern China."

Lines 704-705: I don't understand what wants to be said here.

**Response:** We would like to show the importance of studying carbon cycle in the Tibetan Plateau. We merged the 'Results' and 'Discussion'. The sentence is not appropriate here. Hence, we deleted it and added the descriptions as below (Line 696-702).

"The rapid increase of $CH_4$ would probably make it difficult to meet the goals of carbon emission reduction in the future. This would be especially true with the scenario of quickly increasing $CH_4$ on the Qinghai-Tibetan Plateau due to the emissions from the two largest source regions of Northern India and Eastern China. The large growth rate of atmospheric $CH_4$ in the TP revealed that i) the atmospheric $CH_4$ at the WLG was not predominantly influenced by eastern cities in recent years and ii) large amounts of $CH_4$ were transported from the Tibetan Plateau to WLG in recent years."

Line 705: "anomalously" -> "anomalous"

**Response:** We have revised it.

Lines 706-707: how about trends in other regions in the world? This statement is the same as in the abstract and I still don't understand it. This is rather a statement that fits to a synthesis analysis as it is e.g. done in the annual WMO GHG bulletin.

**Response:** The trend at WLG is similar to other stations in the Northern Hemisphere, and also the global level reported by WMO and WDCGG sum43, especially after 2007, with high growth rate.

Thank you for your comment. Yes, the statements are not appropriate here, and they fit to a synthesis analysis better. We rewrote the abstract and conclusion and revised the related sentence (Line 715-719).

"Additionally, the Tibetan Plateau was intensively affected by strong sources over time, which showed a larger growth rate than that of the city regions in some periods. The anomalous variation and unprecedented growth rate of the atmospheric $CH_4$ in this region revealed that it was urgent to control $CH_4$ emissions."

Lines 722-723: "… the long-term verification is extremely important to … understand $CH_4$ variations …"; did the understanding improve based on the present analysis? Which are the lessons-learnt?

**Response:** We have revised the conclusion and deleted the related sentence.

The 26-year measurements provided a broader view of changing characteristics of $CH_4$ at WLG and improved our understanding of the future trend, such as three developing stages of $CH_4$ (Fig. 12a), rather than a limited view on $CH_4$ growth, e.g. steady or negative growth in 1994-2006. We also discussed the potential reasons for the increasingly long-term trend (Chapter 3.6.2 and 3.7).

In this study, we found new characteristics of $CH_4$ at WLG. Generally, the characteristics of $CH_4$ varied in different observing periods: i) the diurnal cycle has been becoming apparent and the amplitudes of the diurnal or seasonal cycles increased over time, ii) the wind sectors with elevated $CH_4$ mole fractions switched from ENE-…-SSE sectors in early periods to NNE-…-E sectors in later years, iii) the area of source regions increased as the years progressed and strong sources shifted from northeast (city regions) to southwest (Northern India), iv) the annual growth rates in recent years (e.g. 2008-2019) were significantly larger than that in early periods (e.g. 1994-2007).

Lines 725-726: "Tibetan Plateau was with the highest average altitude …" awkward English.

**Response:** We have deleted the description.

Line 727: "anomalously" -> "anomalous"

**Response:** We have revised it (Line 717).

Data availability statement: very strangely, this paragraph only refers to the data from the other stations but nothing is said how to access WLG data. The data used in the analysis doesn't seem to be freely available since WDCGG only contains daily and monthly $CH_4$ averages from WLG; where can anybody access hourly $CH_4$ data from WLG?

**Response:** Thank you for your question. We will upload the data as supplementary material.

**-Reviewer 2**

General comments: In this manuscript, a 24-year long-term observation of atmospheric $CH_4$ at Waliguan WMO/GAW global station in the Tibetan Plateau was studied. This report is very meaningful and the analysis of the paper is very comprehensive. The $CH_4$ variations and its related potential causes during the long-term observation have been analyzed in details, which would help the scientific community to understand carbon cycle and formulate more informed carbon reduction policy. Given the importance and value of the long-term measurements of $CH_4$ in the Tibetan Plateau plus that this manuscript is well drafted, I would recommend accepting this paper after minor modifications listed as below.

> **Response:** Thank you very much for your positive comments. We revised the manuscript and answered the questions point by point.

Specific comments:

Line 98: add 'About' before 90%.

> **Response:** We have added it (Line 82).

Line 233-238: How frequent is the backward trajectory computed? Hourly? Please specify the numbers of trajectories are determined.

> **Response**: Yes, it's hourly. We computed the 3-day back trajectories coincident with hourly $CH_4$ mole fractions. The number of trajectories was determined by the valid hourly $CH_4$ observations. We added the descriptions (Line 229-230). The related information, e.g. numbers, and average mole fractions, was listed in Table 3.

Some expression is not professional, e.g. "long-distance transport" (Lines 465), can be expressed by "long-range transport", and so on. Please polish the whole report.

> **Response:** Thank you for your suggestion. We have revised all the descriptions in the manuscript.

There are several places in results and discussion where too many details are given, which make the text a little difficult to follow. Results and discussion are suggested to be merged.

**Response:** Thank you very much for your constructive suggestion. We merged 'Results' and 'Discussion' and rewrote the Chapter. The interpretation/conclusion was described in conjunction with results.

Lines 685-686: Suggest to give more discussions about the larger growth rates in the Tibetan Plateau than that of city region.

**Response:** We rewrote the Chapter 3.7. The interpretation/conclusion was described in conjunction with results. More discussions were added (Line 685-702).

"These results suggested that there were possibly other strong $CH_4$ sources at the WLG that were not from cities and the southwest region (Northern India) was the most likely contributor. The PSCF analysis also supported this result (Fig .7). At present, Northern India and Eastern China were the two largest sources of $CH_4$ at the WLG (Fig. S10) (Crippa et al., 2019). Since the Tibetan Plateau was coincidently trapped in the middle of them, the atmospheric $CH_4$ at WLG was very likely dominated by long-range transport from these two regions. Although $CH_4$ emissions increased slowly during 1994-2002, a negative trend appeared (Fig. S10), significantly increased emissions were found in both southeast and southwest Asia after 2007. Chen et al. (2013) illustrated that the warming (0.2 °C per decade) in the Tibetan Plateau resulted in substantial emissions of $CH_4$ due to the thawed permafrost and melted glaciers. The rapid increase of $CH_4$ would probably make it difficult to meet the goals of carbon emission reduction in the future. This would be especially true with the scenario of quickly increasing $CH_4$ on the Qinghai-Tibetan Plateau due to the emissions from the two largest source regions of Northern India and Eastern China. The large growth rate of atmospheric $CH_4$ in the TP revealed that i) the atmospheric $CH_4$ at the WLG was not predominantly influenced by eastern cities in recent years and ii) large amounts of $CH_4$ were transported from the Tibetan Plateau to WLG in recent years."

Line 1057: delete 'CO'.

Response: We are sorry for the mistake. The average CO mole fractions were added in Table 3.

Why are the points not on the line in figure 8 and 9?

Response: Thank you for your question. The lines are the smoothed curves of the points using the method of 'LOESS' Curve Fitting (Local Polynomial Regression), hence they do not just connect the points.

**Changing characteristics of atmospheric $CH_4$ in the Tibetan Plateau:**

**records from 1994 to 2019 at the Mount Waliguan station**

**Shuo Liu[1,2,3], Shuangxi Fang[3], Peng Liu[4], Miao Liang[5], Minrui Guo[6], Zhaozhong Feng[1,7]**

**[1]State Key Laboratory of Urban and Regional Ecology, Research Center for Eco-Environmental Sciences,**

**Chinese Academy of Sciences, Beijing, China**

**[2]College of Resources and Environment, University of Chinese Academy of Sciences, Beijing, China**

**[3]College of Environment, Zhejiang University of Technology, Hangzhou, China**

**[4]Mt. Waliguan background station, China Meteorological Administration (CMA), Qinghai, China**

**[5]Meteorological Observation Center (MOC), China Meteorological Administration (CMA), Beijing, China**

**[6]College of Global Change and Earth System Science, Beijing Normal University, Beijing, China**

**[7] Key Laboratory of Agrometeorology of Jiangsu Province, Institute of Ecology, School**

**of Applied Meteorology, Nanjing University of Information Science & Technology, Nanjing, China**

**Correspondence**:

Shuangxi Fang (fangsx@cma.gov.cn); Zhaozhong Feng (zhzhfeng201@hotmail.com)

**Abstract.** A 26-year long-term  record of atmospheric methane (CH₄)

measured in-situ at the ount Waliguan (WLG) station, the only

WMO/GAW global station in inland  Eurasia, is presented. Overall, a nearly continuous increase of atmospheric CH₄ was observed at the WLG with a yearly growth rate of $5.1 \pm 0.1$ ppb yr$^{-1}$ during 1994-2019, except for some particular periods with near-zero or negative values, e.g. 1999-2000, and 2004-2006. The average CH₄ mole fraction was only 1805.8 $\pm$ 0.1 ppb in 1995, but  increased about 133 ppb and reached a historic  level of 1932.8 $\pm$ 0.1 ppb in 2019.

The case study in the Tibetan Plateau showed that the atmospheric CH₄

increased rapidly. During some special period, it is even larger than that of city regions (e.g. $6.7 \pm 0.2$ ppb yr$^{-1}$ in 2003-2007). Generally, the characteristics of CH₄ varied in different observing periods

.

i) the diurnal cycle has been becoming apparent and the amplitudes of the diurnal or seasonal cycles increased over time, ii) the wind sectors with elevated CH₄ mole fractions switched from ENE-…-

SSE sectors in early periods to NNE-…-E sectors  in later years, iii) the area of source regions  increasing as the years progressed and strong sources  shifted from northeast (city regions) to southwest (Northern India), iv) the annual growth rates in recent years (e.g. 2013 2019) were significantly  larger than that in early periods (e.g. 1994-

2007).

**Introduction**

Since the pre-industrial era, the emission of greenhouse gases (GHGs) ha increased continuously and large  increases  have been found in recent years with the concentration now being higher than ever  (WMO, 2019). The GHGs perturb the infrared radiation balance, which trap the heat in the atmosphere and contributes to global warming, melting glaciers, extreme weather events and many other global climate changes (IPCC, 2014). The recent 30-year span from 1983

to 2012 wa the warmest of the last 800 years in the Northern Hemisphere, and half of the rising surface temperature was due to increased GHGs emissions (IPCC,

2014). As one of the most important GHGs, methane (CH$_4$) has a global warming effect that is just less than carbon dioxide (CO$_2$) (Etminan et al.,

2016). CH$_4$ has an 8-12 year lifetime (Battle et al., 1996), with  a global warming potential  ~23 times greater than CO$_2$ over a 100-year horizon (Weber et al., 2019). CH$_4$ also contributed to about 17% of the radiative forcing caused by long-lived greenhouse gases was from CH$_4$  1750 to

2016 (Etminan et al., 2016). Since the beginning of the industrial era, the concentration of CH$_4$  rapidly increased because of the influence of anthropogenic activities (Saunois et al., 2016). Results from ice cores analyses in

Antarctica showed that the atmospheric concentration of CH$_4$  reached a level unprecedented  over the last 0.8 million years (IPCC, 2014).

At the beginning of the 1990s, the CH$_4$ mole fraction showed a decreasing trend.

Consequently, the reverse trend has been observed since 1998 due to the higher global mean temperature

(Dlugokencky et al., 1998; Nisbet et al., 2014). However, a low growth rate was sustained over 1999 to

2006, except for  years  with El Niño events (2002/2003)

(Dlugokencky et al., 1998). The annual growth rate dropped from ~12 ppb yr$^{-1}$ to near

0 from the late 1980s to  2006 , followed by a renewed CH$_4$

increase since then (Nisbet et al., 2019). During 2007-2013, the annual growth rate of methane was 5.7 ± 1.2 ppb yr$^{-1}$. After 2013, the atmospheric CH$_4$ grew  at rates not observed

, such as 12.7 ± 0.5 ppb yr$^{-1}$ in 2014 and 10.1 ± 0.7 ppb yr$^{-1}$ in 2015 (Nisbet et al., 2016, 2019). In the most recent decade, the  global mean growth rate was

7.1 ppb yr$^{-1}$  (WMO, 2019).

The World Meteorological Organization/Global Atmospheric Watch Programme (WMO/GAW) annual greenhouse gas bulletin revealed that the global average mole fraction of CH$_4$ reached a new high with 1869 ± 2 ppb in 2018 (Rubino et al.,

2019), which was ~259% of pre-industrial levels (~722 ppb around 1750 C.E.

(Common Era)) (Etheridge et al., 1998; WMO, 2019).

Atmospheric CH$_4$ is mainly emitted from natural sources (e.g. about 40%

from ruminants and wetlands) and anthropogenic sources (e.g. about 60% from paddies, cattle ranch, coal mine, fossil fuel and biomass burning) (Hausmann et al., 2016; Saunois et al., 2016). Observations from the GAW

indicated that the causes of the recent increase were likely attributed to anthropogenic emissions at mid-latitudes in the Northern Hemisphere and the wetlands in the tropics (WMO, 2019). The rapid development of population growth, economic expansion, and urbanization of countries has led to more and more fossil fuel production and consumption (e.g., the large-scale exploitation of natural gas, oil, and coal) and biomass burning, Consequently, large amounts of anthropogenic CH$_4$ were emitted around the world in recent years (Galloway, 1989; Streets and Waldhoff, 2000; Wang et al., 2002; Lin et al., 2014; Hausmann et al., 2016). The recent carbon isotope study revealed that biogenic emissions might also have driven CH$_4$ increase, including microbial sources  from rice, paddies, ruminants, termites, enteric fermentation, or a combination of these (Nisbet et al., 2016; Schaefer et al., 2016; Wolf et al., 2017).

About 90% of the CH$_4$ destruction in the atmosphere is  from the reaction with hydroxyl radicals (·OH) (Vaghjiani and Ravishankara, 1991; Bousquet et al.,

2011), an important oxidant in the troposphere (Logan et al., 1981). Therefore, the inter- annual  variation of ·OH or the decline of the oxidative capacity of the atmosphere may also cause the recent increase in the CH$_4$ growth rates (Rigby et al.,

2017; Turner et al., 2017). T the exact causes of significant increase in the CH$_4$ emissions in past years  remained  debated, especially for the anomalous periods with a suddenly large growth, due to the sparse time and space measurements and the crude model s, which limited our understanding of the global variation of atmospheric CH$_4$ (Saunois et al., 2019; Weber et al., 2019). As

a consequence, it will be impractical to predict CH$_4$

trends in the future, and then to develop realistic management (Nisbet et al., 2019).

Systematic observations are a prerequisite to get an accurate understanding of the spatial and temporal behaviors of atmospheric CH$_4$ concentrations.

Since 1978, systematic measurements of atmospheric CH$_4$ have been taken around the world (Blake et al., 1982; Rasmussen and Khalil, 1984; Dlugokencky et al., 1994). On the northern slope of the Mauna Loa volcano, Hawaii, there exists the first global station, Mauna

Loa (MLO), which has performed at about 3397m above sea level (a.s.l.) and far away from local sources and sinks. It has the longest record of continuous atmospheric CH$_4$

observation (Keeling et al., 1976). Recently,  other stations have been  installed for CH$_4$ observation, such as the Barrow (BRW), South Pole (SPO) (polar site) (Dlugokencky et al., 1995), Cape Grim (CGO) and Minamitorishima (MNM) in Australia (coastal/island sites) (Pearman and

Beardsmore, 1984; Wada et al., 2007), Minamitorishima (MNM) in Japan (coastal/island sites) (Wada et al., 2007), Jungfraujoch (JFJ) in Switzerland and Mount

Waliguan in China (continental mountain site) (Zhou et al., 2004; Loov et al., 2008).

Hundreds of CH$_4$ observation stations worldwide are currently running under the framework of the WMO/GAW. Even though, the exact causes of significantly increased

CH$_4$ emissions in past years are still remained unclear and debated, especially for the anomalous periods with suddenly large growth, due to the time and space sparsity of measurements and the crude model approaches, which limited our understanding of the global variation of atmospheric CH$_4$ (Saunois et al., 2019; Weber et al., 2019). As long as the reasons for rising CH$_4$ emissions contributed by natural sources (e.g. wetlands), anthropogenic sources (e.g. fossil fuels), or climate change feedbacks remain uncertainties, it will be impractical to predict CH$_4$ trends in the future, and then to develop realistic management (Nisbet et al., 2019). Therefore, it is essential to establish typical observing regions and perform long observations.

China has the largest anthropogenic CH$_4$ emissions in the world (Janssens-

Maenhout et al., 2019). The. Qinghai-Tibetan Plateau withhas an average altitude of over 4000m a.s.l., which has long been recognized as the roof of the world. By coincidence, the two largest CH$_4$ source regions in the world (i.e.i.e. Eastern China and

Northern India) are trapped by the Tibetan Plateau in betweenthe middle (Zhang et al.,

2011; Fu et al., 2012; Wilson and Smith, 2015). Under the characteristics of special geographical conditions, lower population density, rarely industrial activities, and high sensitivity to external disturbances, the Tibetan Plateau is undoubtedly one of the ideal regions to observe a continual CH$_4$ signal (Zhou et al., 2005; Fu et al., 2012; Zhang et al., 2013). Most of the previous studies reported on the short-term CH$_4$ variations in

China and concluded that the importance of long-term observation (Cai et al., 2000;

Zou et al., 2005; Wang et al., 2009; Fang et al., 2013), which is of great value to enhance the understanding of the global carbon cycle ((Cai et al., 2000; Zou et al., 2005; Wang et al., 2009; Fang et al., 2013); Yuan et al., 2019). As the rapid development of China and India continues, the characteristics of CH$_4$ on the Tibetan Plateau, might change significantly over time. Since 1994, in-situ measurements of atmospheric CH$_4$ have been launched at the

Mt. Waliguan (WLG) station. To study the long-term variations of the atmospheric CH$_4$

and get a new insight  on its characteristics in the inland of  Eurasia, the performance of a  26-year   in situ observations  at the Mt. Waliguan baseline observatory were evaluated, which is the longest  records in China. Temporal patterns, annual variations, long- term trends, air mass transports, and the spatial distribution of potential sources were analyzed. In addition, the case studies combining atmospheric carbon monoxide (CO)

measurements and a separate analysis between the Tibetan Plateau and the city regions were performed to constrain the contribution of anthropogenic emissions.

2  **Methodology**

2. 1  **Measurement site**

The Mt. Waliguan (WLG, 36.28° N, 100.09° E, 3816m a.s.l.) station is situated at the edge of northeast Tibetan (Qinghai-Xizang) Plateau, which  is in remote western China and isolated from populated and industrial regions (Fig. 1). WLG

is the only WMO/GAW global background station in Eurasia and is  run by the China Meteorological Administration (CMA). The surrounding  of the site are pristine with sparse vegetation, naturally arid and semi-arid grasslands. Small farms with yak and sheep are in the valley. Two adjacent large cities Xining (~2.2 million populations) and Lanzhou, are located about 90 km northeast and 260 km east of the station, respectively. The Longyangxia hydroelectric station (~380 km$^2$) is located approximately 13 km south to southwest of the  WLG. The predominant winds at the WLG are mainly from southwest and east in winter and summer, respectively (Zhou et al., 2004; Zhang et al., 2011), which is controlled by Tibetan Plateau monsoon.

Simultaneously, diurnalal variations of vertical winds at the WLG is are influenced by mountain-valley breezes, where upslope flow brings heated air masses from the boundary layer to the site in daytime and downslope flow results in cool air masses transport from the mountain peak to the site. Under this unique location, the observation at the WLG could can obtain essential information on CH$_4$ sources and sinks from

Eurasia (Zhou et al., 2005; Zhang et al., 2013).

**2. 2  Instrumental setup**

Atmospheric CH$_4$ has been measured quasi-continuously using a HP 5890 gas chromatograph (GC) equipped with a flame ionization detector (FID) since July 1994, and an Agilent 6890N GC equipped with a FID since June 2008. Both of the systems used the same sampling procedures. A Cavity Ring Down Spectroscopy system (Picarro

G1301) began in January 2009 and the instrument was upgraded to Picarro G2401 in

2015. Ambient air is delivered to the above systems at about 5 L/min by a KNF

Neuberger N2202 vacuum pump via a dedicated 0.95 cm o.d. sample line from an 80m intake line attached to an 89m steel triangular tower located approximately 15m from the main observatory. The residence time of the ambient air from the top of the tower to the instrument is 30 s. The ambient air is first passed through a 7 mm stainless steel membrane filter located upstream of the pump and then (after the pump) passed through a pressure relief valve set at 1 atm to release excess air and pressure. The ambient air is then dried to a dew point of approximately -60°C by passing it through a glass trap submerged in a -70°C methanol bath. All standard gases supplied to the instruments are from pressurized 37.5 L treated aluminum alloy cylinders fitted with high-purity, two- stage gas regulators. Stainless steel tubing (0.32 cm o.d., 0.22 cm i.d.) is used for the standard gas sample line and the ambient sample line after the cold trap. The An automated sampling module equipped with a VICI 8 ports valve is designed to sample from separate gas streams (standard tanks and ambient air). According to the comparability target of WMO/GAW program (WMO, 2019), methane mole fractions are referenced to a Working High standard (WH) and a Working Low standard (WL).

Additionally, a calibrated cylinder filled with compressed ambient air is used as a Target gas (T) to check the precision and stability of the system routinely. Diagram of the observing system during different periods could be seen at Zhou et al. (2004) and Fang et al. (2013). Here, we focus on the longest continuous measurements of $CH_4$ from

August 1994 to  December 2019 at WLG. Data gaps in limited periods are because of the malfunction of instrument and the maintenance of the sampling system.

The records of CO in this study was initinially observed by an RGA-3 gas chromatograph (GC) equiped with an HgO reduction detector (Trace Analytical Inc.)

since 1994. An automated sampling module was designed to sample from ambient air and a series 9 standards. Deailed diagram of the system was described by Zhang et al.

(2011). Since 2010, the CO has been measured by the Cavity Ring Down Spectroscopy instrument (Picarro G1302 and G2401 since 2015). The scale for all of the CO

measurement were further updated to WMO X2014A.

2. 3  **Data processing**

Most on-site $CH_4$ observations were unavoidably influenced by local sources and other complex conditions (e.g. traffic transportation, various topography). As a result, the records cannot fully represent the regional atmospheric $CH_4$ in well-mixed conditions (Liu et al., 2019). To  get regionally representative  records, we excluded the $CH_4$  data influenced by local sources adjacent to the site (e.g. agricultural fields, cities, traffic emissions). The hourly $CH_4$ data were classified as Local/Regional representative through the meteorological approach, which was based on essential meteorological information, similar to previous studies by Zhou et al. (2004) and Liu et al. (2019). In this study, the $CH_4$ records associated with local surface wind from selected sectors, i.e. NNE-…-ENE in spring, NE-…-SE in summer, NE-…-ESE in autumn, and NE-ENE in winter, were flagged as locally influenced (27.0%). Subsequently, we  rejected portion of daytime records to minimize the effect of human activities (16.9%), e.g.

9:00-13:00 LT (local time) in  summer, and 10:00-17:00 LT in winter. Finally, we filtered CH4 data into locally influenced when the surface wind speed we less than 1.5 m s⁻¹ to minimize the very local accumulation (9.2%).

In order to  investigate the influence of anthropogenic emissions from cities and remote area as Tibetan plateau, we divided the CH4 data into two main influencing regions according to the  analysis, including the geographical conditions, the effect of surface winds, the long-range transports, and the potential source distributions. The first region covered the northeast and southeast (NNE-…SE) of the WLG, which is denoted as City Regions (CR). The second region is located the south to west (S-…-W) of the station and is well known Tibet (Qinghai-Xizang) Plateau (TP) (Fig. S1).

Accordingly, the hourly CH4 records when the surface wind coming from these sector were divided into two subsets (i.e. TP and CR). The long-term variations between the two regions as well as the total regional time series  were further compared and analyzed

To understand the influence of local surface wind,  hourly CH4 data  was binned  into 16 horizontal wind direction classe (Fang et al., 2013).

[revised manuscript text omitted]

In order to understand the year to year variations, we  analyzed the $CH_4$ variation over different periods  in 1994-2019. The entire $CH_4$ time series were divided into five observing periods, i.e. 1994-1997, 1998-2002, 2003-2007, 2008-2012, and 2013-

2019, according to the significant stages or the critical time period of atmospheric $CH_4$

variations from_ ly studies (Zhou et al., 2004; Fang et al., 2013; Zhang et al.,

2013; Nisbet et al., 2019; WMO, 2020). Unless special notes, the average values in this study  are presented with 95% confidence intervals (CIs).

**3   Results and discussion**

**3. 1   Extracting the regional atmospheric methane**

To precisely understand the characteristics of atmospheric $CH_4$, including seasonal cycles and long-term trends, it is vital to identify the $CH_4$ records in well-mixed air without local contaminations (Liu et al., 2019). In this study, hourly $CH_4$ measurements between 1994 and 2019 were analyzed, resulting in 64.0% of the $CH_4$ data being classified as regionally representative, with an average $CH_4$ mole fraction of 1865.8 ±

0.4 ppb. The average of the locally influenced data (1868.2 ± 0.3 ppb) was larger than that of the regionally representative records (Table 1). The filtered regional/local time series was shown in Figure 2. It can be seen that the $CH_4$ mole fractions increased from

1994 to 2019. The atmospheric $CH_4$ showed a strong growth and displayed large fluctuation. In 1994, the average $CH_4$ mole fraction was only 1799.0 ± 0.4 ppb, however, the average value increased 133 ppb by the year 2019 (1932.0 ± 0.1 ppb)

(Table 2). Compared with the global average mole fractions in recent years, i.e. 1853 ±

2 ppb in 2016, 1859 ± 2 ppb in 2017, and 1869 ± 2 ppb in 2018 (WMO, 2019; 2020), the atmospheric $CH_4$ mole fractions at the WLG were significantly higher. These results indicated that the WLG was affected by strong $CH_4$ sources in recent years, which were possibly due to the influence of the two largest source regions of Northern India and

Eastern China (Fang et al., 2013; Zhou et al., 2004).

3. 2   **Diurnal variations**

Distinct diurnal cycles were observed in four seasons during 1994-

2019 at the WLG. The $CH_4$ mole fraction increased from early morning, reached the maximum at noon, and had a trough in the late afternoon (Fig. 3f).

However, differences also existed in four seasons. In spring and summer, the atmospheric $CH_4$  increased from 9:00 to 13:00 LT at noon, with the daily amplitude of 5.8 ±2.8 and 4.4 ± 3.4 ppb, respectively (Table S1).

In autumn, the diurnal variation showed an amplitude of

4.3 ± 3.1 ppb,  with one peak at noon. In winter, a large increasing was found during

9:00-17:00 LT, with the largest  peak to trough amplitude of 7.1 ± 2.9

ppb

Different patterns for diurnal $CH_4$ cycles were also found over different periods. In

1994-2002, the $CH_4$ mole fractions in the winter were  higher than the other seasons (Fig. 3a-b), which was likely due to the manner of heating (e.g. large biomass burning) as well as the adverse diffusion conditions in cold weather (Fang et al., 2013). However, the highest value was found in summer during the period of 2003-2007, 2008-2012, and 2013-2019 (Fig. 3c-e), which could be ascribed to the transport of anthropogenic emissions (e.g. southeast cities) by the meteorological factors (Zhang et al., 2013). Additionally, in summer, the intense herding or grazing activities around the WLG have enhanced the regional $CH_4$

emissions and hence contributed to the higher $CH_4$ mole fractions (Zhou et al.,

2004).

Similar to the previous studies by Zhang et al. (2013) and Fang et al. (2013), diurnal cycles  were ambiguous before 2002 (Fig. 3a-b), but significant diurnal variations appeared afterward (Fig. 3c-e), which indicated that the local sources were weak at the WLG in the past. The apparent diurnal cycles after 2002 may be attributed to the intense activities by humans (e.g. grazing, burning fuel), which were enhanced in the daytime and weakened at nighttime (Fang et al., 2013). The peak to trough amplitude almost increased  in almost all seasons. For example, in spring, the amplitude was 6.5 ± 3.1, 4.7 ± 2.1, 5.6 ± 2.7, 7.0 ± 3.1,4 and 6.9 ± 3.14 ppb over the periods of 1994-1997, 1998-2002, 2003-2007, 2008-2012, and 2013-2019, respectively (Table S1). The meteorological conditions (e.g. diffusion and transport) could have also contributed to the increasing $CH_4$ amplitude. The WLG was remote from the populated center, therefore, the good diffusion conditions in the daytime may have brought high anthropogenic emitted $CH_4$ to the site. The increasing amplitude and $CH_4$ mole fractions suggested that the WLG was increasingly affected by local and regional anthropogenic sources, such as gas exploitation, and grazing (Zhou et al., 2004).

**3. 3  The impact of local surface winds**

Similar to the  short-term study by Zhou et al. (2004), the atmospheric $CH_4$ at the WLG was significantly influenced by local surface wind from the northeast and southeast sectors (Fig. 4f). Slight differences were also found among seasons. In the spring, when the wind was originating from the NNE-NE-ENE-E sectors, the atmospheric $CH_4$ was enhanced by 2.5-6.5 ppb as compared to the seasonal average (1839.7 ± 1.4 ppb) . In the summer and autumn, the wind from the NNE-NE-ENE-E-ESE produced higher $CH_4$ mole fractions, with an enhancement of  3-9.5 ppb and 4 -18 ppb, respectively. In  winter, similar to that in spring, the $CH_4$ mole fractions significantly increaseed  from NNE-…-E sectors, with a value of 7-21 ppb higher than the seasonal average (1854.5 ± 4.8 ppb). In summer, the prevailing winds were from the NE-…-ESE sectors (~46%) (Fig. S2), and the CH$_4$ mole fractions were also higher in the related sectors. However, in autumn and winter, although the prevailing wind and the high wind speed were from SSW-…-W sectors (~ 40-50%) (Fig. S2), the high CH$_4$ mole fractions were from the opposite wind sectors of NNE-…..-ESE (Fig. 4f), suggesting that strong local sources were distributed from northeast to southeast of the WLG (city regions), and even covered the emission of natural sources. Causes of the elevated CH$_4$ from these sectors could be attributed to the large plantations of highland barley as well as the high population density in those areas (Fang et al., 2013). The two largest cities of Xining (with population of ~2.2 million) and Lanzhou (with population of ~4 million) are also situated in the northeast and east of the WLG, respectively. The heavy human activities from anthropogenic fossil combustion, landfills, and livestock, could have also emitted large amounts of CH$_4$. Based on the data from the Emissions Database for Global Atmospheric Research (EDGAR), the increase of CH$_4$ emission was 500 kg yr$^{-1}$ in these two regions throughout 1994-2015 (Crippa et al., 2019). Also, the Yellow River Canyon industrial area (YRC), which is ~500 km northeast of the WLG, may also have contributed to the high CH$_4$ values (Zhou et al., 2003). With the rapid development of land-use, water utilization, and agriculture sources in the YRC, large CH$_4$ emissions could have easily transported to the WLG. Previous studies on black carbon (BC) and carbon monoxide (CO) also revealed that the high CH$_4$ values at the WLG in winter were a result of transport from the YRC (Tang et al., 1999; Zhou et al., 2003).

The wind-rose distribution of CH$_4$ mole fractions indicated that the

 elevated CH$_4$ mole fractions varied in  early periods (i.e. 1994-1997 and 1998-2002) to recent periods (2003-2007, 2008-2012, and 2013-2017). The elevated CH$_4$ was predominately from the ENE.-SSE sectors in the early years (Fig. 4a-b) but evolved to the NNE-NE-ENE-E sectors in later years (Fig. 4c-e).

Furthermore, the amplitude of enhancements was  also increasing along with the progression of  time. For example, in autumn, the maximum s were from  E in 1994-1997, ENE in 1998-2002, ENE again in 2003-

2007, NE in 2008-2012, and finally ENE in

2013-2017, with the successive increasing  of

8.6, 12.1, 14.7, 16.8, and 19.7 ppb, respectively. Therefore, the local surface wind from the city regions had an increasing effect on the atmospheric $CH_4$ at the WLG.

The CPF was applied  to hourly $CH_4$ and CO data by considering intervals of entire data percentiles including $0$-$20^{th}$, $20^{th}$-$40^{th}$, $40^{th}$-$60^{th}$, $60^{th}$-$80^{th}$, and

$80^{th}$-$100^{th}$  to draw the CPF polar plot (Uria-

Tellaetxe and Carslaw, 2014). It was clear that the different sources only affected the

$CH_4$ mole fractions on different percentile range (Fig. 5), meaning that the specific sources were prominent for specific percentile ranges.  for most wind speed-directions, the CPF probability of the $CH_4$ being greater than the $60^{th}$ percentile was tending to zero , and most sources contributed to the  percentiles  less than the $60^{th}$ for the $CH_4$ mole fraction (e.g.

$40^{th}$-$60^{th}$ ) . These results implied that most areas around the WLG had a small contribution to the $CH_4$ emissions. In addition, the wind from the southwest and the southeast was important for the cases of  the higher percentiles, resulted in the highest $CH_4$ mole fractions of 1849-1872 ppb for the $60^{th}$-$80^{th}$  percentile and

1872-2031 ppb for the $80^{th}$-$100^{th}$ percentile (Fig. 5), which revealed the existence of a southeastern and a southwestern strong source region. The anthropogenic emissions from cities (e.g. Lanzhou or Chengdu) were the only cause for high values in the southeast, and the southwest region that was farther away from the WLG was possibly due to sources from other countries, such as India. It's obvious that the CO sources  gradually shifted  with the increase of the percentile ranges (Fig. 5). The areas where the CPF probabilities were higher  were to the NW-SW sectors when  percentages ranged from 0 to $40^{th}$. Nevertheless, when data exceeded the $60^{th}$ percentile, the high probability areas completely moved to NE-SE sectors . Unlike that of $CH_4$, the high CO mole fractions were consistent from the east regions (urbanized areas) (Fig.

5), indicating strong anthropogenic sources in the city regions (e.g. Xining and

Lanzhou) and different source distributions between $CH_4$ and CO around the WLG

(Zhang et al., 2011).

3. 4  Long-range transport and potential source distributions

3. 4. 1  Air mass transports

Figure 6 illustrates the cluster analysis of the 3-day back trajectories between 2004 and 2017. In the spring, majority of the air masses were from the west and northwest regions, which accounted for about 24% (cluster 3) and 44%

(cluster 5) of the total trajectories (Fig. 6a). These air masses were also accompanied  by higher $CH_4$ mole fractions than those from the east and northeast regions, such as cluster 1 (13.3% of total) and cluster 4 (11.69%) (Table 3).

The largest enhancement was in cluster 3 at ~18 ppb relative to the average

. In the summer, 45% of the air masses were from eastern regions (cluster 1). However, the high $CH_4$ mole fractions were in cluster 2 and cluster

5 from northwest and west regions, respectively. However, low percentages were found in each of those clusters, i.e. 26% in cluster 2 and 7%

cluster 5 (Fig. 6b).

~~cluster 2, with ~9 ppb larger than the average in summer.~~ In the autumn, a large proportion of air masses cluster 2 (49%) and cluster 3 (32%)  originated from the west and southwest station, respectively (Fig. 6c). Cluster 3 showed the highest $CH_4$ mole fraction, with an enhancement of ~4 ppb relative to the seasonal average.
In winter, the air masses  primarily came from northwest and
southwest regions , e.g.  cluster 3 (59%), and
cluster 1 (34%) (Fig. 6d) The cluster 1 brought the highest CH$_4$ mole
fractions with an enhancement of ~7 ppb over the
seasonal average.

The air masses from east and northeast regions passed over the cities of Xining and
Lanzhou (capital of Gansu province), which are the populated centers and industrial
areas (Fig. 6). However, the higher CH$_4$ mole fractions were not observed when the air
masses were from these sectors, the high values were frequently brought by air masses
that came from the northwest to southwest (Table 3). The higher values seen in the
northwest to southwest airflow were because the air masses had passed through the
northwest of the Qinghai province and the central area of the Xinjiang Uygur
Autonomous Region (XUAR). This is where the Ge'ermu urban area (the second
largest city of Qinghai) was located and where there was with rapid industrial
development, natural gas and petroleum resource exploitation, and residue burning of
large crops, hence the CH$_4$ emissions were strong (Fang et al., 2013; Zhang et al., 2013).
This result was also similar to the CPF percentile analysis (Fig. 5), as time went by, the
southwest or northwest regions that were farther away from the site became the
strongest source regions to the WLG.

3. 4. 2  **Spatial distribution of potential source regions**

The potential sources were analyzed over different periods, i.e. 2004-
2007, 2008-2012, and 2013-2017 (Fig. 7). Generally, the strongest  sources
were located  northeast or southeast of the station, especially in summer, but a
large area of sources was identified from the southwest to the northwest regions, which
also contained CH$_4$ emissions from the northwest Gansu province, the northwest

Qinghai province and the southeast of the XUAR. Additionally, tthe source regions differed in various seasons as well as years (Fig. 6). The regions of potential source regions in the spring (Fig. 7Fig. 6a-c) and winter (Fig. 7Fig. 6j-l) wereas obviously larger than thoseat in the summer (Fig. 7Fig. 6d-f) and autumn (Fig. 7Fig. 6g-i). The seasonal difference. was due to the effect of westerlies or the southeast monsoons (Zhou et al., 2004).

There were also trends infor the CH$_4$ source regions correlatedalong with specific years: i) the area of the potential source regions was increasing with the yearsover time, and ii) the location of strong potential sources changed along with the time. For example, in autumn and winter, the strength of CH$_4$ sources were were very strong in the southeast andto northeast regions during 2004-2007 (Fig. 7Fig. 6g & i), and then weakened in 2008-2012 (Fig. 7Fig. 6h & k). Finally,, from and finally (i.e. 2013-2017 the sources ), almost vanished in the eastern regions but moved to the southwest with a very large distribution area (Fig. 7Fig. 6i & l).

More CH$_4$ sources appeared at the WLG along with the progression of time, which could have been attributed to the influence of human expansion. The pattern of strong sources moving indicated that the southwest area relative to the WLG, e.g. Northern India, was gradually becoming a strong CH$_4$ contributor. India with abundant cattle as well as an extensive large-scale coal mining operation possibly contributed large amounts of CH$_4$ to move from northern India to the northeastern Tibetan Plateau (Fig. 7i & l) (Fu et al, 2012). The analysis of air mass transport (Fig. 6d) also supported the conclusions that the air masses from the southwest regions contributed the highest CH$_4$ mole fractions. The studies of atmospheric Hg at the WLG by Fu et al. (2012) also supported this phenomenon, which found the long-range transport of atmospheric Hg from India to the Northeastern Tibetan Plateau.

2. 7  **Extracting the well-mixed ambient methane**

To precisely understand characteristics of atmospheric CH$_4$, e.g., seasonal cycle or

3. 5  **Correlation analysis between CH₄ and CO**

Because parts of the CH₄ and CO in— atmosphere  were from the same anthropogenic sources (e.g. fossil fuel combustion), the long-term trend of

$\Delta CO/\Delta CH_4$ was helpful to understand the variation in the sources/sinks in many studies (Buchholz et al., 2016; Niwa et al., 2014; Tohjima et al., 2014; Wada et al., 2011). In this study, the hourly CO data from 2004 to 2017 was used to further analyze the long- term variations of CH₄ (Fig. S3). T the regression slopes of $\Delta CO/\Delta CH_4$

from 2004 to 2017 were analyzed (Fig. S4). Figure  8 presents the seasonal cycles of the $\Delta CO/\Delta CH_4$ slopes. Generally, the slopes were larger in summer and smaller in winter during the observing period, except for 2004-2007

where the highest slope was in autumn. This was primarily due to the effect of monsoons and air mass transport. Tohjima et al. (2014) found an opposite variation at the Hateruma Island, which showed small slope values in the summer. Wada et al.

(2011) analyzed more than 10-year seasonal variation of the $\Delta CO/\Delta CH_4$ ratios at three monitoring stations, i.e. MNM, Yonagunijima (YON), and Ryori (RYO) in Japan, which also showed an opposite trend to that of the WLG. It was because these sites were considerably affected by the Asian continental source regions, where had enhanced emissions of $CH_4$ in the summer (rice paddies) and CO in the winter (fuels combustion). At the WLG, Regional polarization in the concentration ratios of $CH_4$ and CO were seen (Fig. S5), which implied different strong source distributions between $CH_4$ and CO. This result was also seen in the CPF analysis (Fig. 5). Furthermore, the cluster results (Fig. 6b & d) and the PSCF analysis (Fig. 7f & l) also supported this conclusion. In the summer, the source emissions were mainly from the east-southeast regions (cities) with large amounts of CO but relatively lower $CH_4$ (Table 3), leading to the largest $\Delta CO/\Delta CH_4$. In contrast, the sources mostly from southwest-west regions (Tibetan Plateau) emitted large amounts of $CH_4$ but with relatively lower CO in the winter. Hence, the opposite two air mass transports and source emissions led to a peak in $\Delta CO/\Delta CH_4$ in the summer and a trough in the winter (Fig. 8). These results revealed different local sources/sinks, the special topography conditions and source distributions around the WLG.

Additionally, the regression slopes increased along with the time, which showed the maximum in 2013-2017 and the minimum in 2004-2007. For the year to year variations, tThe $\Delta CO/\Delta CH_4$ slopes showed large fluctuations from 2004 to 2017 at the WLG (Fig. 9Fig. 9). The slopes showed a decreasing trend during 2004-2007 but then increased from 2007- to 2010, and again decreased again after 2010. In spring and summer, increasing trends appeared again after 2014. The slopes in summer were almost the largest but the lowest in winter.

In 2007, a large increase in $\Delta CH_4$ appeared, and from 2010 to 2013, the $\Delta CO$ decreased significantly (Fig. S4). Before 2010, large air masses and potential source regions were identified in eastern regions (cities) with the highest CO emission (Fig. 7 and Table 3). After 2010, the southwest regions had the highest $CH_4$ emissions but relatively low CO emissions. Therefore, the strong seasonal variation of $\Delta CO/\Delta CH_4$ also revealed that the WLG was affected by different anthropogenic sources, e.g.

sources from cities, and sources from the Tibetan Plateau, during a year, especially in the summer and winter. The long-term trend of the slopes implied that the source emission types (CO sources or $CH_4$ sources) around the WLG might have been changing with human activities, like straw burning, in the early years, or coal mining in recent years.

3. 6  **Variation of long-term records**

3. 6. 1  **Seasonal cycles**

Overall,  the seasonal averages of regionally representative $CH_4$ at the WLG

were almost in a cycle starting with summer (1861.7 ± 0.4 ppb), then winter (1847.7 ± 0.3 ppb), autumn (1844.9 ± 0.4 ppb), and spring (1841.1 ± 0.3 ppb), except during 1994-1997 where there was a maximum in the winter and a minimum in the autumn (Fig. 10). Seasonal averages in the

CR were slightly different from that in the TP and even the total regional data. The seasonal averages in the TP were mostly higher than that in the CR from 1994 to 2019, except for in the winter (Table S2). The maximum mole fractions were mostly found in August and the minimum mole fraction appeared in April for the total regional time series, with a seasonal amplitude of 14.4 ppb. The peak to trough amplitude in the CR (~16.7 ppb) was higher than that in the TP (~15.1 ppb) during 1994-2019. The seasonal variations were consistent with the previous short-term studies from 2002-2006 at WLG (Zhang et al., 2013). However, these variations were almost opposite to observations at the adjacent stations of Lin'an, Shangdianzi, and Longfengshan in China (Fang et al., 2013, 2016). For other regional sites in the Asia, Guha et al. (2018) studied seasonal variability at the Sinhagad (SNG) and Cape Rama station (CRI) over India, which also showed an opposite trend to the WLG due to the strong impact of monsoon dynamics. Ahmed et al. (2015) found that the seasonal $CH_4$ showed a maximum in the winter and a minimum in the spring at two urban sites of Guro (GR) and Nowon (NW), in Seoul, Korea over 2004-2013. Kim et al. (2015) investigated the decadal variation (1991-2013) of $CH_4$ at the East Asian sites, e.g. Ulaan Uul (UUM) in Mongolia and Tae-ahn Peninsula (TAP) in Korea, which revealed again an opposite seasonal trend to that of the WLG.

The data were further compared with similar WMO/GAW global stations in the Northern Hemisphere, including MLO (19.54° N, -155.58° E, 3397m a.s.l.) (Dlugokencky et al., 1995, 2019a), JFJ (46.55° N, 7.99° E, 3580m a.s.l.) (Zellweger et al., 2016), MNM (24.29° N, 153.98° E, 7.1m a.s.l.) (Matsueda et al., 2004; Tsutsumi et al., 2006), as well as the marine boundary layer (MBL) from the NOAA/ESRL lab at a similar latitude (Dlugokencky et al., 2019b). It was seen that the stations in the Northern Hemisphere and the MBL showed an opposite trend to the WLG with the minimum being in summer and maximum in the winter or spring (Fig. 11). The peak in the summer at the WLG was attributed to larger grazing, human activities, ruminants, and easterly winds coming from neighboring areas over other seasons. The $CH_4$ emissions from yaks and other ruminants in the Tibetan Plateau (alpine pasture) were very strong in the summer, preceded only by paddy emissions (Fang et al., 2013; Zhang et al.,

2013). Furthermore, the dynamic transport by airflow from the polluted northeast/southeast region was also strong in summer, which all induced high $CH_4$ mole fractions and consequently an opposite trend than other sites (Ma et al., 2002; Xiong et al., 2009).

The seasonal amplitude at the WLG (~14 ppb) was significantly lower than many other sites in the Northern Hemisphere, by about 35-70 ppb. Such sites included MLO in America, BRW in North Pole, UUM in Mongolia, TAP in Korea, Ny-Ålesund in Norway, Bialystok in Poland, Ochsenkopf in Germany, and Beromunster in Switzerland (Dlugokencky et al., 1995; Kim et al., 2015; Morimoto et al., 2017; Thompson et al., 2009; Popa et al., 2010; Satar et al., 2016). MBL also showed a larger amplitude than WLG (Fig. 11). The study at the SNG and CRI over India showed a much larger amplitude close to 200 ppb (Guha et al., 2018). The low amplitude at the WLG was because of the high elevation of the continental mountain sites, where there were relatively fewer effects from local influences than the coastal/island sites (Yuan et al., 2019). Additionally, the ~~Seasonal averages in CR were significantly different to that in TP and also the entire regional data (Total). The seasonal average in TP was mostly higher than that in CR, except for wintertime (Table S2). The atmospheric $CH_4$ in August was mostly the maximum and the April was the minimum for the total regional time series (Total), with the seasonal amplitude of 13.4 ppb. The peak to trough amplitude in CR (~15 ppb) was higher than that in TP (~13 ppb) during 1994-2016. Additionally,swaslydroppedalong withwere(Total)displayeabout99363242016.~~. This revealed that the Tibetan Plateau was intensively affected by strong regional sources (e.g. grazing or emissions from India) over time.

  **Long-term trend**

The fluctuating trend in atmospheric $CH_4$ during 1994-2019 at the WLG (Fig. 12) was similar to the global trend reported by many studies (Bergamaschi et al., 2013; Rigby et al., 2017; Nisbet et al., 2019). In the 1990s, the $CH_4$ growth rates were very  small  or even negative at WLG. Subsequently,  a steady period  with  near-zero growth rates was found during 2002-2006. However,  the atmospheric $CH_4$ increased significantly after 2007 (Fig. 12a). In  1997/1998, 2000/2001, 2007/2008, and 2011/2012, a large fluctuation in the growth rates was found and a strong growth appeared (Fig. 12b). The growth rates fluctuated evenly with both positive and negative values before 2009. However, almost all of the growth rates showed a positive value after 2009. Three  developing stages (i.e. highlighted green, blue, and red blocks) could be seen from the 1990s to 2010s. The $CH_4$ mole fraction slightly decreased during 1998-2000 (green color), and then  went through a relative steady period during 2003-2006 (blue color), finally increased  rapidly after 2007 (red color) (Fig. 12).

The  annual growth rate waere 5.1 ± 0.1 ppb yr$^{-1}$ throughout 1994-2019 at the WLG (Table 4). The periodic annual growth rats were 4.9 ± 0.1, 2.5 ± 0.2, 4.9 ± 0.1, 7.7 ± 0.1, and 5.5 ± 0.1 ppb yr$^{-1}$ during 1994-1997, 1998-2002, 2003-2007, 2008-2012, and 2013-2019, respectively. Similar growth rates were found between the CR and the TP during 1994-2019 (Fig. S6). In 1994-1997 and 2003-2007, the growth rates in the TP were even larger than that in the CR (Table 4). These results indicated that there were also strong $CH_4$ sources from the TP. The previous study by Zhou et al. (2004) showed the annual increase in $CH_4$ by 4.5 ppb yr$^{-1}$ in 1992-2001, which was close to our study in 1994-1997 and 1998-2002. Tohjima et al. (2002) found that the $CH_4$ levels at the Cape Ochi-ishi and Hateruma Island in1995-2000 respectively increased by 4.5 and 4.7 ppb yr$^{-1}$, which were also similar to that of the WLG. Tsutsumi et al. (2006) analyzed the trend of hourly $CH_4$ data from 1998 to 2004 on the YON, which showed a similar increase (~3.0 ppb $yr^{-1}$) to the WLG. The study at the GR and NW in Seoul, Korea, presented almost an identical trend of 2 ppb $yr^{-1}$ between 2004 and 2013 (Ahmed et al., 2015), which was lower than that of the WLG in similar period. In the early 1990s, the $CH_4$ growth rates at the WLG were very low and similar to the global level. The level of •OH radicals might control the decrease or increase of $CH_4$ in the atmosphere during this period (Dlugokencky et al., 1998; Rigby et al., 2017; Turner et al., 2017). However, the growth rates were high in 1998 (Fig. 12b), which may have been due to the high temperatures and a large amount of biomass burning (Cunnold et al., 2002; Lelieveld et al., 2004; Simmonds et al., 2005). The growth rate of $CH_4$ observed at the Ny-Ålesund, Svalbard, increased from $0.3 \pm 0.2$ ppb $yr^{-1}$ during 2000-2005 to $5.5 \pm 0.2$ ppb $yr^{-1}$ during 2005-2014, which had a similar variation but with a little lower growth rates than that of the WLG (Morimoto et al., 2017). The study suggested that the temporal pause in 2000-2005 was ascribed to the reductions of $CH_4$ emissions from the microbial and fossil fuel sectors, while the increase in 2005-2014 was due to an increase in microbial release.

The constantly larger $CH_4$ growth rate after 2007 at the WLG (Fig. 12) (Table 4) was similar to the recent studies by Nisbet et al. (2016) and (2019), which showed that the global $CH_4$ increased by $5.7 \pm 1.2$ ppb $yr^{-1}$ in 2007-2014, and was much higher at $12.7 \pm 0.5$ ppb $yr^{-1}$ in 2014, with $10.1 \pm 0.7$ ppb $yr^{-1}$ in 2015, $7.0 \pm 0.7$ ppb $yr^{-1}$ in 2016, and $7.7 \pm 0.7$ ppb $yr^{-1}$ in 2017. The average growth rate in the Northern Hemisphere was $7.3 \pm 1.3$ ppb in 2007 and $8.1 \pm 1.6$ ppb in 2008 (Dlugokencky et al., 2009), which was also similar to the observation at the WLG (Table 4). After 2007, most sites in the Northern Hemisphere had large $CH_4$ growth rates. Also, the average global growth rate was similar to the WLG at 7.1 ppb $yr^{-1}$ in the most recent ten years (WMO, 2019). Fang et al. (2013) showed that the annual growth rate of $CH_4$ was $9.4 \pm 0.2$ ppb $yr^{-1}$ in 2009-2011 at the WLG, which was a little higher than this study in 2008-2012. The adjacent stations in China also revealed the high $CH_4$ growth rates of $8.0 \pm 1.2$ ppb $yr^{-1}$ at Lin'an in 2009-2011, 7.9 ± 0.9 ppb yr$^{-1}$ at Longfengshan in 2009-2011, and 10 ± 0.1 ppb yr$^{-1}$ at Shangdianzi in 2009-2013 (Fang et al., 2013, 2016), which were all higher than the similar periods of 2008-2012 or 2013-2019 at the WLG (Table 4). The $CH_4$ measurements in other countries, such as the Beromünster tall tower station, also showed a high growth rate of 9.66 ppb yr$^{-1}$ in 2012-2014 (Satar et al., 2016). The warm temperatures, biomass burning, and the climatic anomalies (El Niño or La Niña), likely enhanced the $CH_4$ emissions after 2007 (Dlugokencky et al., 2009). The anomalous years of increasing or decreasing (e.g. 2007/2008) might have a significant influence on the overall $CH_4$ trend (Fig. 12). These frequent anomalies also appeared in most long-term observation stations, e.g. MLO in the USA (Dlugokencky et al., 2009) and Mount Zugspitze in Germany (Yuan et al., 2019), due to climatic forces, such as those exceptions during the El Niño oscillation, forest fires, volcanic eruptions, and extreme weather events (Keeling et al., 1995; Dlugokencky et al., 2009; Keenan et al., 2016; Nisbet et al., 2019).

Many studies have investigated the potential reasons for the anomalous increasing. The study by Satar et al. (2016) at Beromünster, Switzerland explained that the short-term spikes were possibly related to emissions from agricultural activities, while the longer-lasting peaks were because of air mass transport and mixing. The isotopic evidence suggested that the significant increase of biogenic emissions was the dominant factor for the $CH_4$ rise. This was especially true in the tropical wetlands that had strong rainfall anomalies, and agricultural sources such as rice paddies and ruminants were a cause while fossil fuel emissions were not the main cause (Nisbet et al., 2016). The study from thaw ponds at Arctic regions revealed that there had very weak correlation between the amount of $CH_4$ released from ponds and environmental factors, e.g. air temperature and atmospheric pressure (Burke et al., 2019). Sweeney et al. (2016) using 29-year of measurements on North Slope of Alaska (BRW) to investigate the sensitivity of $CH_4$ emissions to the temperature change, which revealed that despite the short-term temperature sensitivity increases $CH_4$ emissions, it would have little impact in the long term. However, up to now, the specific causes of such distinct variability through the years, including the spikes or near-zero $CH_4$ growth rates, have not yet been determined. It is well established that human activities were mainly responsible for the recent rapid $CH_4$ growth rates and anomalies. The analysis from EDGAR showed the $CH_4$ emission per sector in China (Fig. S7) (Crippa et al., 2019). During the observing period, the waste, oil, and natural gas combustion, and open burning continuously emitted large amounts of $CH_4$ into the air. After 2000, the $CH_4$ emissions from solid fuel increased greatly in China. After 2003, the $CH_4$ emitted from rice cultivation also increased continuously (Fig. S7). The increased emissions from these sectors greatly contributed to the $CH_4$ increase at the WLG, as well as the other regions in China. In addition, studies revealed that China's coal sector dominated the positive trend in recent years, which contributed to the highest proportion of anthropogenic $CH_4$ emissions (~33%) (Janssens-Maenhout et al., 2019; Miller et al., 2019). In 2010-2015, China's coal production increased (from 3400 to 4000 million metric tons), but $CH_4$ emissions from rice cultivation, agriculture practices, ruminants, waste, and oil/gas consumption only increased slightly if at all (EIA, USA). Therefore, the control measures of coal mining reduction or limiting natural gas and petroleum exploitation may play an important role in slowing down $CH_4$ emissions in China.

The $CH_4$ growth rate in CR was significantly different from that in TP (Fig. S3). In 1994-1997, 2003-2007 and 2013-2016, the growth rates in TP were obviously larger than that in CR (Table 4). But in 2003-2007 and 2008-2012, the CR showed higher annual growth rates. In addition, for the entire observing period (i.e. 1994-2016), the growth rates in both TP ($5.2 \pm 0.1$ ppb yr$^{-1}$) and CR ($5.0 \pm 0.1$ ppb yr$^{-1}$) were similar to the overall annual growth rates (Fig. S3).

**2. 103. 7   Case study for air mass transport**

As described above, the northeast and southeast city regions might have acted as strong regional sources influencing the atmospheric $CH_4$ at the WLG. Therefore, to analyze the effect of long-distance range transport of emissions from cities, we further excluded the regionally representative data by air mass transport was further excluded by air mass transport, and the the rest ofremaining regional records were were denoted as 'TR'.

First, We applied the monthly cluster analysis was applied to hourly trajectories over

2005-2007, 2008-2012 and 2013-2017. Then, based on the cluster analysis, the clusters were divided into two groups, i.e. from city regions (red clusters in Fig. S8), and other (black clusters in Fig. S8). Finally, the regionally representative data were accordingly classified as two groups based on the cluster results (cities or other). The statistical resultscluster results from city regions were presented presented in detail in Fig.

S4Figure S8 and Table S3 Table S3.

Consequently, tThe proportions of trajectories from cities wereas 40.3%, 32.5%, and 6.8% in 2005-2007, 2008-2012, and 2013-2017, respectively. And about 22.8%,

35.6%, and 38.4% of the regional records were associated with air masses transport from city regions in 2005-2007, 2008-2012, and 2013-2017, respectively (Table 5Table

5). The average $CH_4$ values for thewhen air masses transport from city region air massess (1863.0 ± 0.3 ppb) wereas obviously higher than the other sectors (1850.6 ±

0.2 ppb) (Fig. S9Fig. S5). –The overall growth rates of the TR in the periods of 2005-

2016 or 1994-2016 were similar to the original data series (Fig. S9). However, after excluding the $CH_4$ records for thewhen air trajectoryies transports from city regions, the growth rates of the TR in 2008-2012 (10.1 ± 0.1 ppb yr$^{-1}$) and 2013-2017 (6.3 ± 0.1

ppb yr$^{-1}$) (Table 5Table 5) were higher than the original regional data series (i.e.i.e. 7.6

± 0.2 and 5.7 ± 0.1 ppb yr$^{-1}$) (Table 4Table 4). And the overall growth rate for TR in

2005-2016 or 1994-2016 was still similar to original data series (Total), no significant difference was found (Fig. S5).

These results suggested that there were possibly other strong $CH_4$ sources at the

WLG that were not from cities and the southwest region (Northern India) was the most likely contributor. The PSCF analysis also supported this result (Fig .7). At present,

Northern India and Eastern China were the two largest sources of $CH_4$ at the WLG (Fig.

S10) (Crippa et al., 2019). Since the Tibetan Plateau was coincidently trapped in the middle of them, the atmospheric $CH_4$ at WLG was very likely dominated by long-range transport from these two regions. Although $CH_4$ emissions increased slowly during 1994-2002, a negative trend appeared (Fig. S10), significantly increased emissions were found in both southeast and southwest Asia after 2007. Chen et al. (2013) illustrated that the warming (0.2 °C per decade) in the Tibetan Plateau resulted in substantial emissions of $CH_4$ due to the thawed permafrost and melted glaciers. The rapid increase of $CH_4$ would probably make it difficult to meet the goals of carbon emission reduction in the future. This would be especially true with the scenario of quickly increasing $CH_4$ on the Qinghai-Tibetan Plateau due to the emissions from the two largest source regions of Northern India and Eastern China. The large growth rate of atmospheric $CH_4$ in the TP revealed that i) the atmospheric $CH_4$ at the WLG was not predominantly influenced by eastern cities in recent years and ii) large amounts of $CH_4$ were transported from the Tibetan Plateau to WLG in recent years.

[revised manuscript text omitted]

4  **Conclusion**

Three developing stages of atmospheric $CH_4$ at Mt. Waliguan from the 1990s to 2010s
were found. The $CH_4$
mole fractions slightly decreased during 1998-2000, and then went
through a relatively steady period during 2003-2006 and finally increased rapidly after
2007. Although near-zero and even negative growth appeared in some
periods,  the overall  $CH_4$  increased
rapidly, especially in recent years.
Although most areas around the WLG had small contributions to the $CH_4$
emissions, two strong source regions were found from the northeast and southwest of
the site.
Northern India possibly has ome a strong contributor
than city regions were in the past.
The temporal patterns , the annual
variations, the long-term trends, or the  source distribution of $CH_4$ at are all changed in recent years. We found that the WLG was increasingly affected by local sources such as human activities.

Additionally, the Tibetan

Plateau was intensively affected by strong sources over time

In recent years, which showed a larger growth rate than that  of the city regions in some periods. Tibetan Plateau was with the highest average altitude and~~

the anomalous variation and unprecedented growth rate of the atmospheric

CH4 in this region revealed that it was urgent to control CH4 emissions. Reducing the emissions from strong source sectors like coal mining, natural gas or solid fuel exploitation, and rice cultivation may play an important role on CH4 emissions reduction in China.

*Data availability.* The gridded meteorological data (2004-2017) from NOAA-ARL was available at ftp://arlftp.arlhq.noaa.gov/pub/archives/gdas1/. The data from MLO, JFJ,

MNM station was downloaded from World Data Centre for Greenhouse Gases (WDCGG) at https://gaw.kishou.go.jp/. The MBL data was available at ftp://aftp.cmdl.noaa.gov/data/trace_gases/CH4/flask/surface/. The geographical distribution of annual emission data by Emissions Database for Global Atmospheric

Research (EDGAR) was from website https://edgar.jrc.ec.europa.eu/overview.php?v=50_GHG.

*Author contributions.* SL, SF and ZF designed the research. SL performed the data processing with assistance of SF and MG. The station were monitored, maintained ML

and PL, and they collected, preprocessed, and provided the hourly observational dataset. SL and SF finished the manuscript with contributions from all the co-authors.

*Competing interests.* The authors declare that they have no conflict of interest.

*Acknowledgments.* This study was funded by the National Key Research and

Development Program of China (2017YFC0209700). We also thanks to the staff who have contributed to the system installation and maintenance at the Waliguan in past decades.

**Table 1.** The statistics of the filtered CH$_4$ data series over different periods during 1994-2019 at the

WLG station.

| Year | Regionally representative | | | Locally influenced | | |
|---|---|---|---|---|---|---|
| | Hours | Percentage (%) | Mean (ppb) | Hours | Percentage (%) | Mean (ppb) |
| 1994-1997 | 16122 | 71.3 | 1801.7 ± 0.5 | 6481 | 28.7 | 1806.2 ± 0.3 |
| 1998-2002 | 26347 | 83.2 | 1832.6 ± 0.7 | 5336 | 16.8 | 1837.7 ± 0.3 |
| 2003-2007 | 28181 | 69.4 | 1832.3± 0.3 | 12443 | 30.6 | 1839.2 ± 0.2 |
| 2008-2012 | 19627 | 63.5 | 1856.2 ± 0.4 | 11287 | 36.5 | 1865.2 ± 0.3 |
| 2013-2019 | 21683 | 44.2 | 1906.8 ± 0.3 | 27329 | 55.8 | 1920.4 ± 0.4 |
| 1994-2019 | 111960 | 64.0 | 1865.8 ± 0.4 | 62876 | 36.0 | 1868.2 ± 0.3 |

**Table 2.** Yearly average CH$_4$ mole fractions at the WLG station.

| Year | Mean (ppb) | year | Mean (ppb) |
|---|---|---|---|
| 1994 | 1799.0 ± 0.4 | 2007 | 1837.2 ± 0.5 |
| 1995 | 18035.68 ± 0.1 | 2008 | 1854.8 ± 0.1  |
| 1996 | 18084.86 ± 0.2 | 2009 | 1847.2 ± 0.1  |
| 1997 | 181106.58 ± 0.2 | 2010 | 1856.6 ± 0.2  |
| 1998 | 18267.80 ± 0.1 | 2011 | 1867.4 ± 0.1  |
| 1999 | 181920.72 ± 0.1 | 2012 | 1879.6 ± 0.2  |
| 2000 | 181920.70 ± 0.2 | 2013 | 1895.7 ± 0.4  |
| 2001 | 18479.32 ± 0.4 | 2014 | 1890.2 ± 0.2  |
| 2002 | 18335.75 ± 0.2 | 2015 | 1913.0 ± 0.4  |
| 2003 | 18402.85 ± 0.2 | 2016 | 1914.4 ± 0.2  |
| 2004 | 1836.17 ± 0.2 | 2017 | 1911.6 ± 0.1  |
| 2005 | 18367.74 ± 0.1 | 2018 | 1925.6 ± 0.3  |
| 2006 | 1834.7 ± 0.2 | 2019 | 1932.0 ± 0.1 |

**Table 13.** The statistics for the cluster analysis result for both CH₄ and CO from 2004-2017 at the

WLG station. The clusters from urban areas are highlighted with facein bold.

| | Cluster | Number | Average CH₄ mole fraction | Average CO mole fraction |
|---|---|---|---|---|
| Spring | **1** | **1243** | **1853.4 ± 2.7** | **185.4 ± 3.3** |
| | 2 | 685 | 1852.6 ± 3.4 | 147.8 ± 3.6 |
| | 3 | 2231 | 1877.6 ± 2.5 | 175.6 ± 3.0 |
| | **4** | **1093** | **1850.5 ± 2.2** | **196.0 ± 4.3** |
| | 5 | 4108 | 1860.8 ± 1.5 | 137.8 ± 1.1 |
| Summer | **1** | **3981** | **1869.9 ± 1.2** | **173.2 ± 2.4** |
| | 2 | 2244 | 1878.5 ± 2.6 | 135.0 ± 2.6 |
| | **3** | **1040** | **1866.3 ± 2.6** | **165.3 ± 5.2** |
| | 4 | 916 | 1857.8 ± 2.4 | 152.2 ± 5.1 |
| | 5 | 578 | 1876.3 ± 5.1 | 146.5 ± 5.4 |
| Autumn | **1** | **1133** | **1870.1 ± 3.7** | **159.1 ± 8.0** |
| | 2 | 4235 | 1868.8 ± 1.3 | 110.3 ± 1.3 |
| | 3 | 2745 | 1873.6 ± 1.6 | 135.1 ± 2.6 |
| | 4 | 550 | 1865.2 ± 3.6 | 149.1 ± 6.6 |
| Winter | 1 | 3066 | 1879.8 ± 2.1 | 159.9 ± 3.6 |
| | **2** | **601** | **1872.6 ± 3.4** | **282.2 ± 2.1** |
| | 3 | 5261 | 1865.8 ± 1.0 | 129.6 ± 1.7 |

**Table 4.** Annual growth rates of atmospheric $CH_4$ in the City Regions (CR), the Tibetan Plateau (TP), and  total regional records   from 1994 to 2016 at the WLG station.

| | 1994-1997 | 1998-2002 | 2003-2007 | 2008-2012 | 2013-2019 | 1994-2019 |
|---|---|---|---|---|---|---|
| CR | 3.0 ± 0.1 | 3.6 ± 0.2 | 5.6 ± 0.2 | 7.0 ± 0.2 | 6.2 ± 0.1 | 5.2 ± 0.1 |
| TP | 3.4 ± 0.1 | 3.0 ± 0.2 | 6.8 ± 0.2 | 5.6 ± 0.2 | 5.7 ± 0.1 | 5.1 ± 0.1 |
| Total | 4.6 ± 0.1 | 2.6 ± 0.2 | 4.9 ± 0.1 | 7.6 ± 0.1 | 5.7 ± 0.1 | 5.1 ± 0.1 |

**Table 5.** The statistics of CH$_4$ data without air mass transport from city regions (TR) over different periods during 2005-2016 at the WLG station.

| | Transport regions | Hours | Percentage (%) | Average (ppb) | Updated growth rate (ppb yr-1) |
|---|---|---|---|---|---|
| 2005-2007 | TR | 6922 | 77.2 | 1824.9 ± 0.2 | 2.7 ± 0.2 |
| | City | 2041 | 22.8 | 1835.9 ± 0.5 | - |
| 2008-2012 | TR | 7060 | 64.4 | 1853.7 ± 0.2 | 10.1 ± 0.1 |
| | City | 4254 | 35.6 | 1861.0 ± 0.3 | - |
| 2013-2016 | TR | 4152 | 61.6 | 1888.2 ± 0.3 | 6.3 ± 0.1 |
| | City | 2591 | 38.4 | 1888.5 ± 0.5 | - |
| 2005-2016 | TR | 18134 | 67.1 | 1850.6 ± 0.2 | 7.0 ± 0.1 |
| | City | 8886 | 32.9 | 1863.2 ± 0.3 | - |

[Figure]

Figure 1. The lLocation of Mt. Waliguan WMO/GAW global station as well as the other regional
stations in China. The gradient color indicates altitude.  The digital elevation model (DEM) was
downloaded from Geospatial Data Cloud site, Computer Network Information Center, Chinese
Academy of Sciences (http://www.gscloud.cn), and then processed by ArcGis software. The China
map was derived from © National basic Geomatics Center of China (http://www.ngcc.cn/ngcc/).
The world map was obtained from © OpenStreetMap (https://www.openstreetmap.org/). And the
other shpfile file data and entire map were created by ArcGis software.

[Figure]

**Figure 2.** The filtered hourly CH$_4$ data series from 1994 to 2019 at the WLG station. The transparent blue points are regionally representative data. The black points are locally influenced data. The red lines are smooth values of the regional data obtained by the curve-fitting routine of Thoning et al. (1989).

[Figure]

**Figure 3.** Diurnal CH₄ cycles in different periods from 1994 to 2019 at the WLG station. The lines with different colors represent various seasons. Error bars indicate the 95% confidence intervals.

[Figure]

**Figure 4.** The wind-rose distribution of the average  CH$_4$  mole fractions from 16 horizontal wind directions over different periods during 1994-2017 at the WLG station. The different colors represent the CH$_4$ data in different seasons. Error bars in all directions indicate 95% confidence intervals.

[Figure]

**Figure 5.** The polar plot of the distribution of  CH4 and CO concentration  probabilities in different percentile ranges at  WLG station. The analysis was based on  conditional probability functions (CPF) by Ashbaugh et al. (1985). The top plots show the  analysis of CH4 from 1994 to 2017. The bottom plots show the CO measurements in 2004-2017. 'ws'  refers to the wind speed. The values  at the bottom of each panel show the range of concentrations in  relevant percentile range. Gradient colors represent the levels of CPF probability in different percentile ranges.

[Figure]

**Figure 6.** Cluster analysis  of the 72-h back trajectories in different seasons (spring (a), summer (b), autumn (c), and winter (d)) during 2004-2017  at WLG station.

The colored lines represent different cluster s. The proportion of trajectories on each cluster is also marked.

[Figure]

**Figure 7.** The gGeographical distribution of the weighted potential sources of CH₄ in different periods over 1994-2017 at the WLG station. The gradient color shows the strong levels of potential source regions in different seasons, i.e.i.e. spring (a, b, c), summer (d, e, f), autumn (g, h, i), and winter (j, k, l), and different periods, i.e.i.e. 2004-2007 (a, d, g, j), 2008-2012 (b, e, h, k), and 2013- (c, f, i, l).

[Figure]

**Figure 7.** Filtered hourly CH₄ data series from 1994 to 2017 at WLG station. The blue points with transparency represent regional events. The black points are the filtered local events. The red lines are calculated smooth values to the regional data by curve-fitting routine of Thoning et al. (1989).

[Figure]

**Figure 8.** The average seasonal variation of the ΔCO/ΔCH₄ slopes in different periods  during

2004-2017 at the WLG station. The error bars show the standard deviation of the monthly averages.

The vertical bars are the monthly numbers of data in different periods.

[Figure]

**Figure 9.** The long-term trend of ΔCO/ΔCH$_4$ slopes over 2004-2017 at WLG station.

[Figure]

[Figure]

**Figure 10.** Monthly variations of regional CH$_4$ mole fractions from 1994 to 2019 at the WLG

station. The 'CR', 'TP' and 'Total' represent the measurements from the City Regions, the Tibetan

Plateau and the  total regional records, respectively. The box  shows the 25$^{th}$

percentile, the median and the 75$^{th}$ percentile from bottom to top. The bottom and the top whiskers respectively reach the minimum and 1.5 times the IQR (interquartile range). The black points are identified as outliers. The red squares are the averages. The cyan lines are the smoothed curve of the averages using the method of loess (Local Polynomial Regression Fitting). The gray bands are the 95% confidence interval of smoothed curve.

[Figure]

**Figure 11.** The top panel (a) shows the smooth curve and monthly values of CH$_4$ mole fraction in the City Regions (CR) and the Tibet Plateau (TP) during 1994-2016 at WLG station. The bottom panel (b) is the annual growth rates of atmospheric CH$_4$ records from CR, TP as well as the total regional time series (Total). The growth rates are calculated from the first derivative of trend curves. The smooth curve and the trend is calculated by the method of Thoning et al. (1989).

[Figure]

**Figure 11 12.** The seasonal cycles of atmospheric CH$_4$ observed  at the WMO/GAW

global stations of Mauna Loa (MLO, 1994-2018), Jungfraujoch (JFJ, 2005-2018), Minamitorishima (MNM, 1994-2019), and Mt. Waliguan (WLG, 1994-2019) in the Nnorthern Hhemisphere. The data of other sites (except WLG) weare from WDCGG. The data in the marine boundary layer (MBL,

1994-2019) weare from NOAA / ESRL lab at  a similar latitude to the WLG.

[Figure]

**Figure 12.** The top panel (a) shows the smoothed curves and monthly values of the CH$_4$ mole fractions in the City Regions (CR) and the Tibetan Plateau (TP) during 1994-2019 at the WLG station. The bottom panel (b) is the annual growth rates of the atmospheric CH$_4$ records from the CR and the TP as well as the total regional time series. The growth rates were calculated from the first derivative of the trend curves. The smoothed curves and the trends were calculated using the method of Thoning et al. (1989).